# Additive Manufacturing of Bulk Metallic Glasses—Process, Challenges and Properties: A Review

**Navid Sohrabi *** , **Jamasp Jhabvala and Roland E. Logé**

Thermomechanical Metallurgy Laboratory, PX Group Chair, Ecole Polytechnique Fédérale de Lausanne (EPFL), 2002 Neuchâtel, Switzerland; jamasp.jhabvala@epfl.ch (J.J.); roland.loge@epfl.ch (R.E.L.)
* Correspondence: navid.sohrabi@epfl.ch

**Abstract:** Bulk Metallic Glasses (BMG) are metallic alloys that have the ability to solidify in an amorphous state. BMGs show enhanced properties, for instance, high hardness, strength, and excellent corrosion and wear resistance. BMGs produced by conventional methods are limited in size due to the high cooling rates required to avoid crystallization and the associated detrimental mechanical properties. Additive manufacturing (AM) techniques are a potential solution to this problem as the interaction between the heat source, e.g., laser, and the feedstock, e.g., powder, is short and confined to a small volume. However, producing amorphous parts with AM techniques with mechanical properties comparable to as-cast samples remains a challenge for most BMGs, and a complete understanding of the crystallization mechanisms is missing. This review paper tries to cover recent progress in this field and develop a thorough understanding of the correlation between different aspects of the topic. The following subjects are addressed: (i) AM techniques used for the fabrication of BMGs, (ii) particular BMGs used in AM, (iii) specific challenges in AM of BMGs such as the control of defects and crystallization, (iv) process optimization of mechanical properties, and (v) future trends.

**Keywords:** additive manufacturing; bulk metallic glass; crystallization; microstructure; mechanical properties; 3D printing

## 1. Introduction

Bulk metallic glasses (BMGs) have the ability to solidify in an amorphous structure, which provides them with high strength and elasticity, hardness, and wear and corrosion resistance, due to the absence of structural defects usually found in crystalline materials such as dislocations, grain boundaries, and chemical segregation. However, BMGs are brittle materials, which generally do not sustain significant plastic deformation at room temperature and generally show poor machinability. BMGs are commonly processed by direct casting and melt spinning with rapid quenching from the melt to avoid crystallization [1,2]. Although it is a convenient one-step process where cooling and forming take place simultaneously, it is difficult to achieve intricate shapes in the final products due to the limitations posed by required high cooling rates.

Thermoplastic forming (TPF) is another method that allows processing BMGs in complicated shapes via compression, injection molding, extrusion, hot rolling, blow molding, or wire drawing just like polymers [3]. The lower viscosity of BMGs in the supercooled liquid region (SCLR) is exploited to facilitate atomic mobility for shaping the material in isothermal conditions. TPF involves a series of steps to process the BMGs at temperatures above their glass transition temperature ($T_g$) as shown in Figure 1. Amorphous BMG feedstock (cast ingots, pellets, plates, rods) is reheated just above $T_g$, formed by applying force in the SCLR, and cooled below $T_g$ [3].

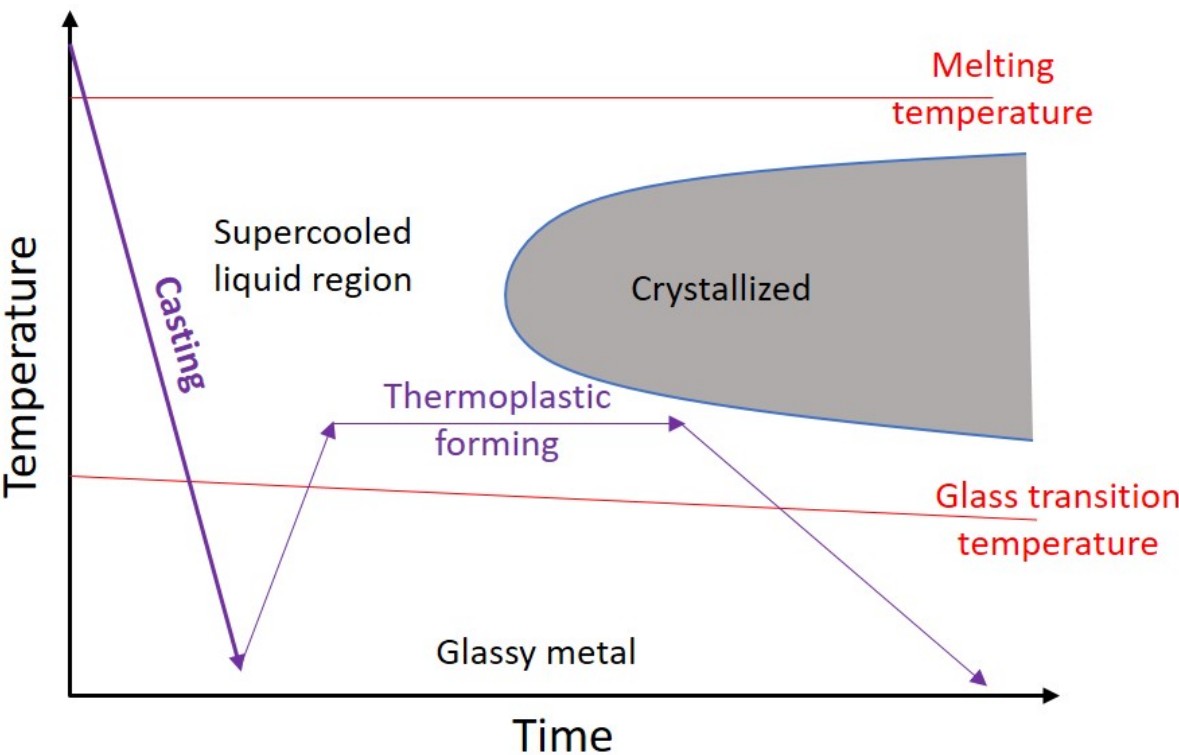

**Figure 1.** Schematic time–temperature–transformation (TTT) diagram showing processing windows for direct casting and thermoplastic forming of BMGs.

BMGs are metastable and tend to crystallize to reach thermodynamic equilibrium. As the temperature is increased in the SCLR, atomic rearrangement favors crystallization. Cardinal et al. [4] showed that crystallization induced brittleness and reduced the toughness of a Au-based BMG. Sohrabi et al. [5] showed that crystallization reduced toughness and led to cracking in a Zr-based BMG. Crystallization also causes solidification shrinkage due to the reduction of volume [6]. Thus, the crystallization of BMGs has to be avoided.

For TPF, the highest possible temperature is desired for low viscosity of the glass, but preventing crystallization then becomes a challenge because the crystallization time ($t_{crys}$) is reduced (Figure 1). Therefore, the suitable temperature window for TPF is a compromise between viscosity and crystallization time [3]. Rapid heating is desirable to leave sufficient time for the complex shaping of BMG before it crystallizes. Johnson et al. [7] used rapid capacitor discharge to heat up Vitreloy 1, which has otherwise low glass-forming abilities (GFA) at rates as fast as $10^6$ K/s. Ma et al. [8] employed resistance welding forming (RWF) to heat a Zr-based BMG. Both of these are millisecond heating methods that make use of the Joule heating principle [9] for instantaneous and volumetric heating of BMGs. These techniques allow for achieving adequate viscosity melt and quick processing without reaching the onset of crystallization. Ultrasonic beating forming (UBF) is another recent practice in which BMG is repeatedly beaten under vibrational loading at high frequency, resulting in heating and forming at the same time [10]. The idea is to minimize the processing steps and time to form the product, thereby avoiding crystallization. The kinetics of heating is crucial to TPF, but there is no need for fast quenching, as is the case in casting, because there is no crystallization nose to bypass, i.e., crystallization time increases monotonically with decreasing temperature. Slow cooling then leads to negligible internal stresses in parts fabricated by TPF [11].

Even though TPF is able to achieve complicated shapes, it faces certain challenges. It is constrained to processing within a short range of time and temperature: BMGs with a narrow SCLR remain not processable by TPF [12]. Due to the short processing time, it is difficult to form bulk parts. Hence, TPF techniques are more popular in BMGs used in micro-electro-mechanical systems [6,13]. As viscosity depends not only on temperature

but also on strain rate, high strain rates often lead to a transition from stable Newtonian to the unstable non-Newtonian flow of the melt, causing furrows and incomplete filling of the mold, ultimately degrading the dimensional accuracy of the product [14].

Apart from the process parameters, the product quality in TPF also depends on the selected tools. As fast heating is required, mold material with high thermal conductivity and low thickness is desirable. For this reason, out of tungsten carbide (WC), K110 steel, alumina, and quartz, WC was found to be the most suitable mold material for uniaxial compression molding [15]. It is also important that deformation should not cause additional heating that would excessively reduce the crystallization time. Physical contact between the viscous BMG and the mold (e.g., press plates in compression molding, Figure 2) may also adversely affect the surface quality of the part. Entrapped air at the interface may induce surface porosity [15], while tool roughness usually agglomerates the viscous melt and increases friction. Lubrication can reduce friction, but compromises the surface finish [16]. The quality of the preform also affects the final product. For example, a good quality end product can be achieved provided the raw material is free from any impurity, porosity, and has a good surface finish [15].

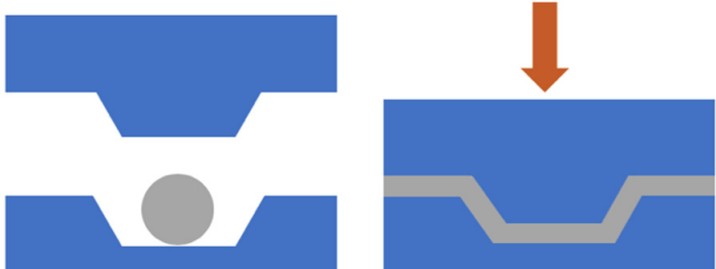

**Figure 2.** Schematic of the compression molding thermoplastic forming method.

As mentioned earlier, the TPF fabrication of large and complex components remains a challenge. In the last decade, the possibility of forming BMGs with additive manufacturing (AM) processes (also known as 3D printing techniques) has been explored. In these processes, parts are fabricated layer by layer, based on a 3D computer-aided design (CAD) model. The known advantages of AM include: lower waste, no need for expensive tooling, mass production of customized parts, low lead time, and the possibility to deal with complex geometries [17]. The complications mentioned above regarding contact with the mold and quality of the preform in TPF are diminished in AM (although the quality of powders remains an issue). Due to extremely high heating and cooling rates and layer by layer deposition, AM manufacturing of large and complex parts is possible [18,19].

The topic of AM of metallic glasses (MGs) has gained momentum since 2015 (see Figure 3) due to the removed constraints on parts geometrical complexity and size.

In laser-based AM methods, since the interaction between the laser and the deposited material is short and confined to a small volume, the local cooling rate ($R_c$) can reach $10^3$–$10^8$ K/s [20], which is typically higher than the critical cooling rate ($CR_c$) of most BMGs [21,22]. This feature allows the material to keep its amorphous structure after solidification and prevents the crystallization in the final part if further heating (after solidification) is limited. However, the $R_c$ in the AM process depends significantly on the process parameters [23].

In the past decade, several reviews have been published on AM of BMGs [18,19,24–27]. Lavery and Williams [19] and Li [24] reported short sections on the topic in 2017, when limited research had been undertaken. Halim et al. [26] included a section about AM of BMGs in a review related to processing methods of BMGs. Liu et al. [18] focused mainly on the crystallization of BMGs fabricated via laser powder-bed fusion (LPBF). Ozden and Morley [25] restricted their review to AM of Fe-based magnetic BMGs. Zhang et al. [27] investigated different techniques used for AM of BMGs mentioned some of the challenges, and focused mainly on the mechanical properties of BMGs and BMG composites.

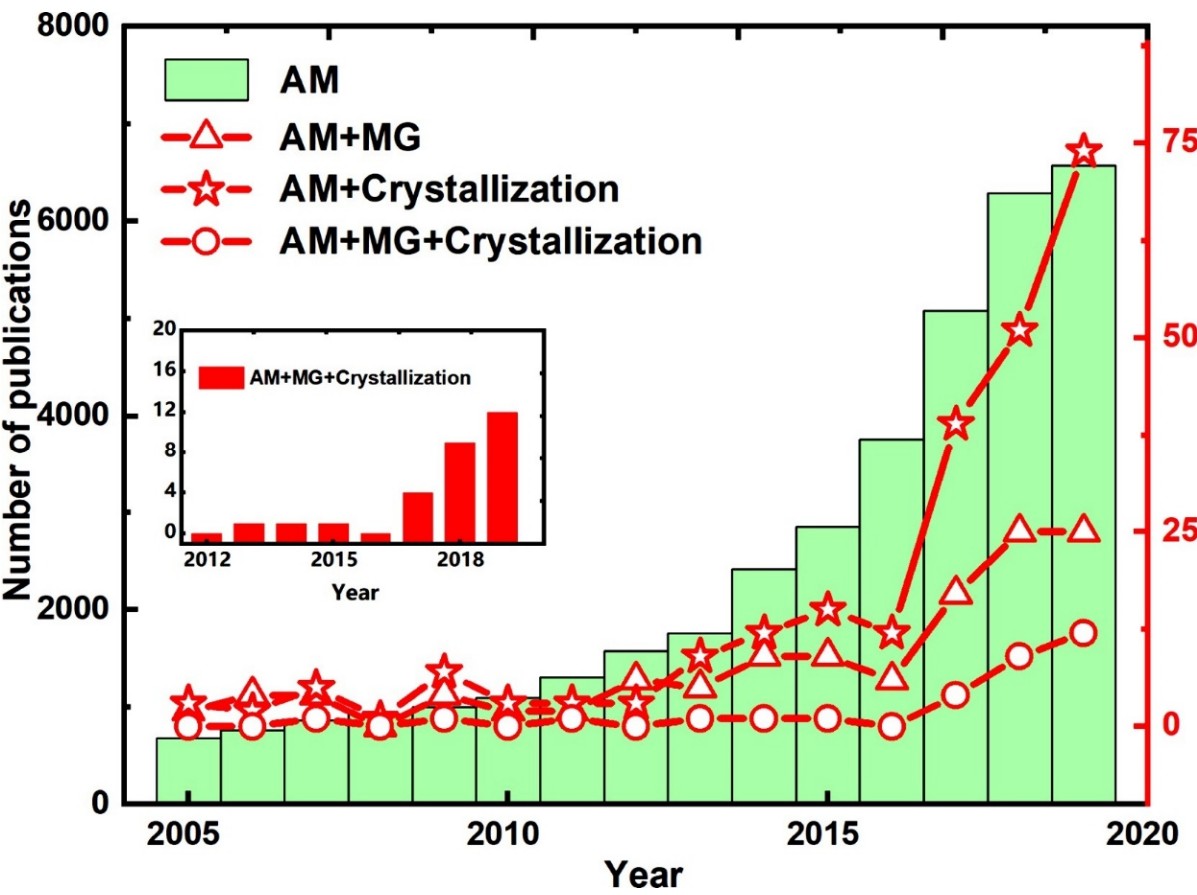

**Figure 3.** Number of publications related to AM, MG and/or crystallization from 2005 to 2019. Reproduced from [18], with permission from Elsevier, 2021.

A full scope review including techniques, materials, microstructure, and properties of BMGs fabricated via AM processes is currently missing. This review therefore targets:

- The materials used in AM of BMGs.
- Challenges in AM of BMGs, such as optimization of processing parameters, defects formation, residual stresses, and low ductility of fabricated parts.
- Crystallization is the main challenge of AM of BMGs, which up to now no comprehensive explanation is presented for the different reasons for the crystallization of BMGs fabricated via AM techniques.
- Mechanical properties of BMGs fabricated via AM processes.

Finally, an outlook is presented in the last section.

## 2. AM Techniques Used for Fabrication of BMGs

AM is a promising alternative to subtractive and replicative manufacturing methods. It allows the creation of complex geometries with high material usage efficiency and without increase of time, material, or difficulty upon increasing geometrical complexity. The model geometry is generally based on a CAD model, which is built layer by layer. After the material deposition and its consolidation, the printing head/build platform rises up/descends by a layer thickness. This repetitive process continues until the model is completely built. Generally, additively manufactured products are produced without the need for tooling or machining [23].

Numerous AM methods and technologies have been developed during the past two decades. The majority of the research on AM of metals and alloys is based on powder feedstock. However, other types of feedstock are also considered, such as wires, metal sheets, ribbons, palettes, etc. (see Figure 4) [28]. The benefits and shortcomings of each of them are indicated in Table 1.

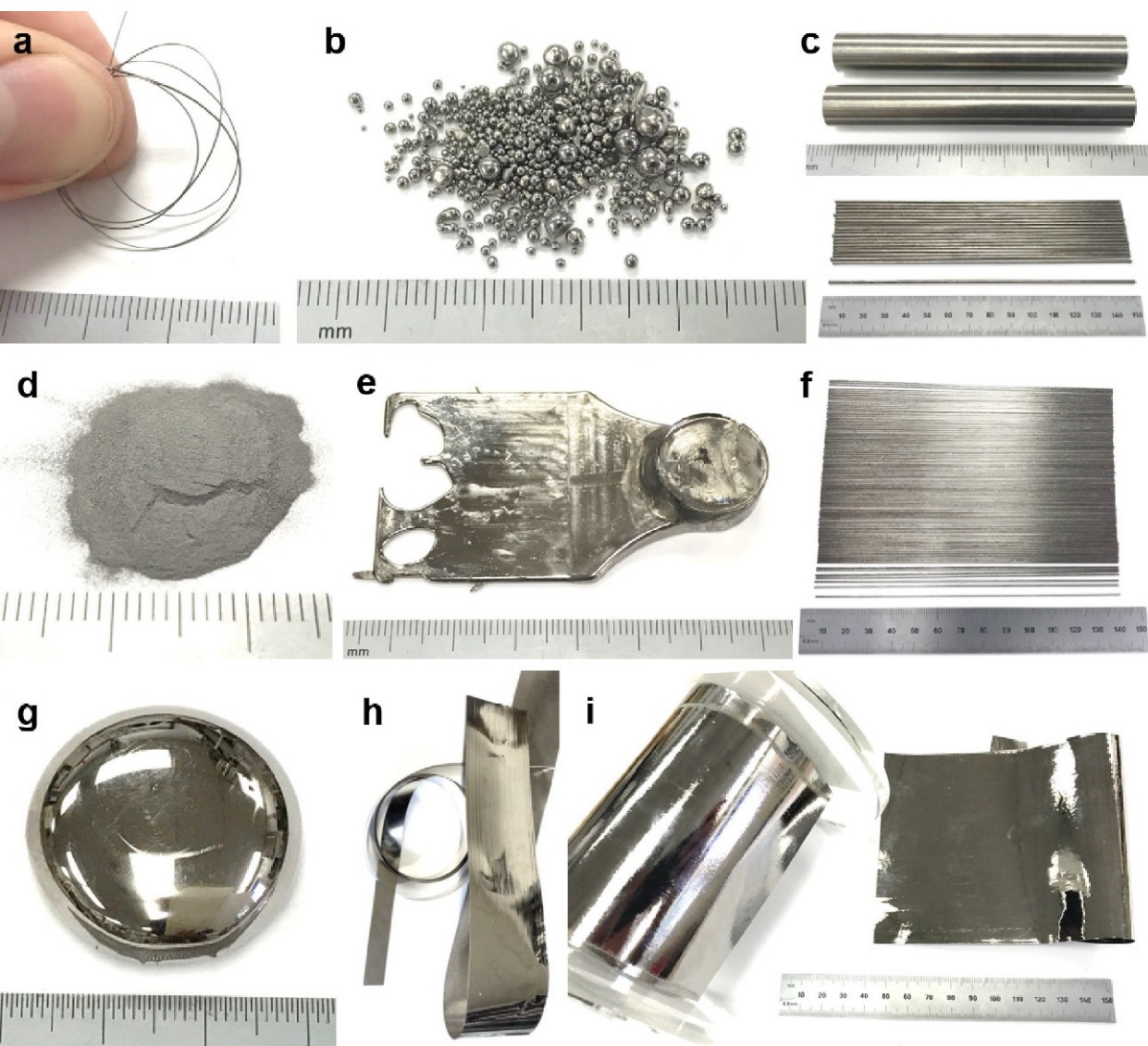

**Figure 4.** Different types of MG feedstock used in AM technologies: (**a**) wire, (**b**) pellets, (**c**) thick cast rods of various diameters, (**d**) powder, (**e**) cast plates, (**f**) thin cast rods, (**g**) ingot, (**h**) melt spun ribbon, and (**i**) melt spun metal sheet. Reproduced from [28], with permission from Elsevier, 2021.

**Table 1.** Benefits and short comings of various types of feedstock for AM technologies. Reproduced from [28], with permission from Elsevier, 2021.

| BMG Feedstock for AM | Typical Feedstock Size (mm) | Scalable Production | Relative Cost | Oxidation and Contamination | Relevance for AM |
|---|---|---|---|---|---|
| Drawn wire | 0.3–1 | Yes | High | Yes | Low |
| Spherical shot | 0.5–5 | Yes | Low | Yes | Low |
| Large cast rod | 2–10 | No | High | No | Low |
| Powder | 0.02–0.08 | Yes | Low | Yes | High |
| Cast plates | 0.75–5 | No | High | No | Low |
| Thin cast rods | 0.5–1 | No | High | No | Low |
| Melt spun ribbon | 0.01–0.05 | Yes | Low | No | Low |
| Metal sheet | 0.1–1 | Yes | Low | No | High |

We identified seven AM techniques for the production of BMGs: laser powder-bed fusion (LPBF), selective laser sintering (SLS), laser solid forming (LSF), direct energy deposition (DED), laser foil printing (LFP), ultrasonic AM (UAM), and fused filament fabrication (FFF). The first three refer to powder bed laser methods. In this section, these technologies are introduced in more detail. Figure 5 shows the related publication activity. The LPBF process is by far the most studied (70%).

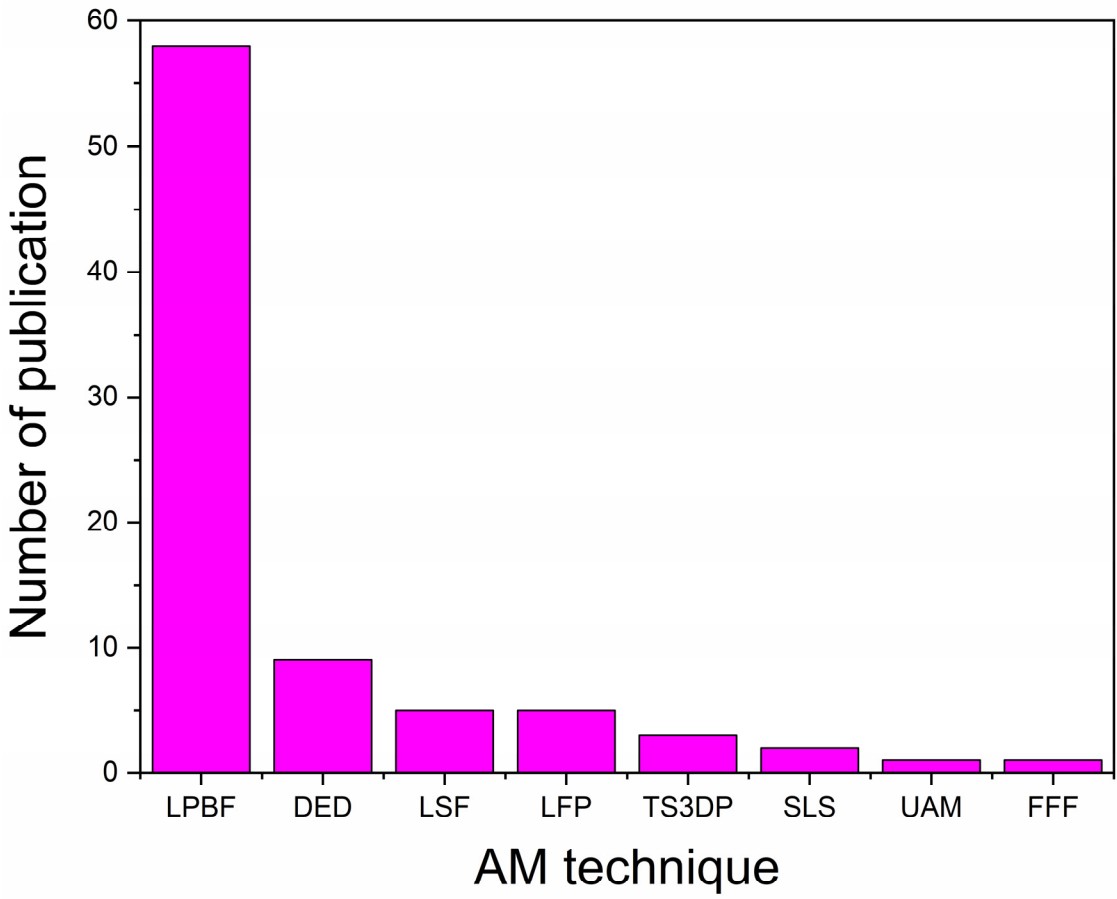

**Figure 5.** Number of publications dedicated to various AM techniques used for the fabrication of BMGs from 2013 to April 2021.

### 2.1. Powder-Bed Laser Methods

#### 2.1.1. Laser Powder-Bed Fusion (LPBF)

Laser powder-bed fusion (LPBF), also called selective laser melting (SLM), is the most widely used and studied metal 3D printing process because of its accuracy and simplicity compared to the other processes [29,30]. As shown in Figure 6, the principle is to "selectively" melt the powder bed, which was previously deposited by a roller or scrapper on a removable substrate, based on a CAD model. The part is then lowered by one layer thickness, and the process is repeated. These actions are continued until the whole part is fabricated [31]. Thanks to the short interaction time between the laser and the powder, in a very small and limited volume, very high heating and cooling rates ($10^4$–$10^8$ K/s) are achieved [30]. This allowed the LPBF technique to process materials that require high cooling rates during solidification, such as BMGs. The typical LPBF cooling rate is indeed higher than the $CR_c$ of most BMGs (<$10^2$ K/s) [32]. Among AM techniques used for fabrication of BMGs, LPBF is the most studied [5,32–86].

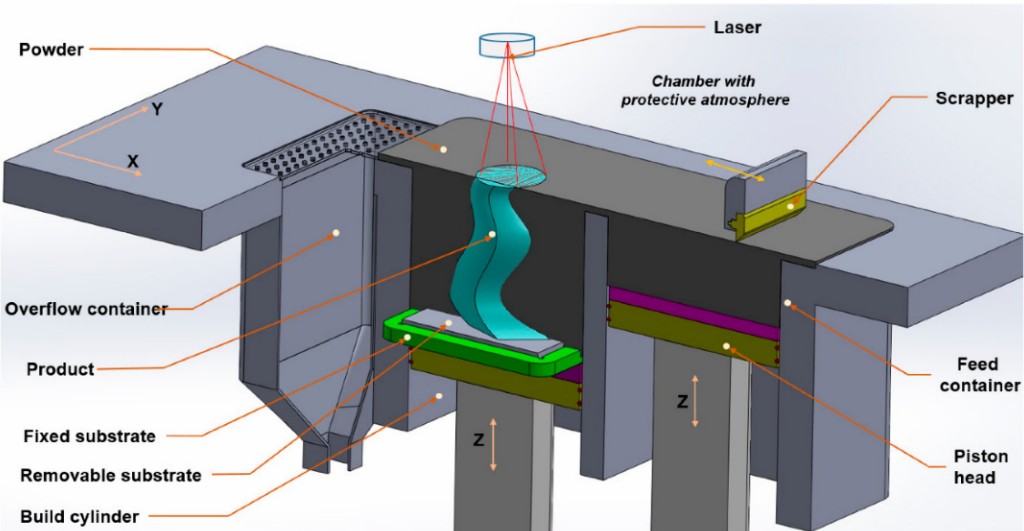

**Figure 6.** Schematic of the LPBF process. Reproduced from [31], with permission from Elsevier, 2021.

Advantages of the LPBF process include good accuracy, ability to produce multiple parts in one build cycle, wide availability of powders for a number of metallic alloys, ability to fabricate complex geometries, and rapid time to market. However, limitations remain concerning the part size (related to the size of the machine), residual stresses, defects such as porosity, and the cost of the powder [29].

LPBF parameters may be categorized into four classes, related to: the laser, the scan strategy, the powder, and the temperature. Each class includes several parameters listed in Figure 7, which can affect the quality of the final part. Some of these parameters will be discussed in the sub-section dedicated to Parameter optimization.

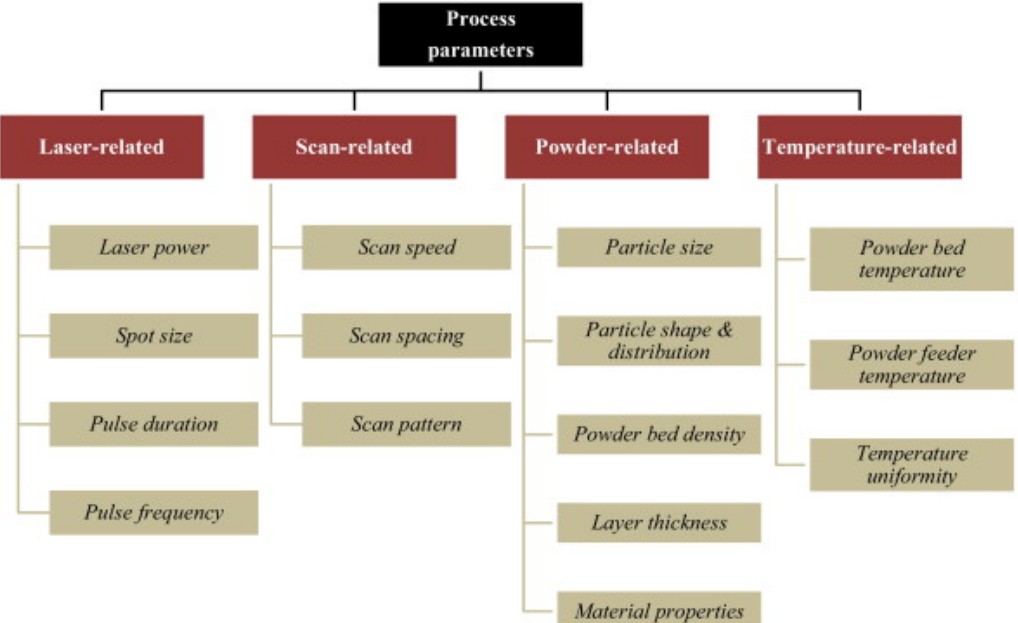

**Figure 7.** Controllable processing parameters in the LPBF process, which influence the final quality of the product [87].

### 2.1.2. Selective Laser Sintering (SLS)

Selective laser Sintering (SLS) is very similar to LPBF (or SLM), the only difference being that the powder particles are only partially melted, i.e., they are sintered [88]. Therefore, the process does not lead to a fully dense part.

SLS was used in two studies for consolidation of MG powders, but it did not result in a fully amorphous and dense parts [89,90].

### 2.1.3. Laser Solid Forming (LSF)

Laser solid forming (LSF) is a powder-bed AM technique with a principle similar to LPBF, but with the following differences:

The laser spot size and powder layer thickness of LSF are almost one order of magnitude larger. Consequently, the size of powder particles used in LSF is larger.

The scanning speed in LSF process is lower than the typical hundreds of millimeters per second used in LPBF.

The combination of larger laser spot size, thicker powder layer, and lower scan speed results in lower cooling and heating rates for the LSF process.

There are several studies in the literature on LSF of BMGs [91–95]. Partial crystallization was systematically observed in the fabricated parts.

### 2.2. Direct Energy Deposition (DED)

Direct energy deposition (DED) is a laser-assisted AM technique, differing from power-bed techniques due to the powder being injected inside the melt pool or a focal zone via a delivery nozzle. Therefore, the flowability and the shape of powder particles are not as important as in the powder-bed processes. Figure 8 illustrates a schematic of the DED process. The main advantages of the DED process are a higher deposition rates, the ability to build large parts, to repair and add features, to be compatible with a wide range of powders, including multiple powders of different materials, with the possibility of in situ alloying. Limitations of DED arise from the lower accuracy, the presence of defects and residual stresses, and the need for a computer numerical control (CNC) robot. The first BMG fabricated via DED was reported in 2017 [96]. Several other Zr- and Fe-based BMGs were produced since then [97–104].

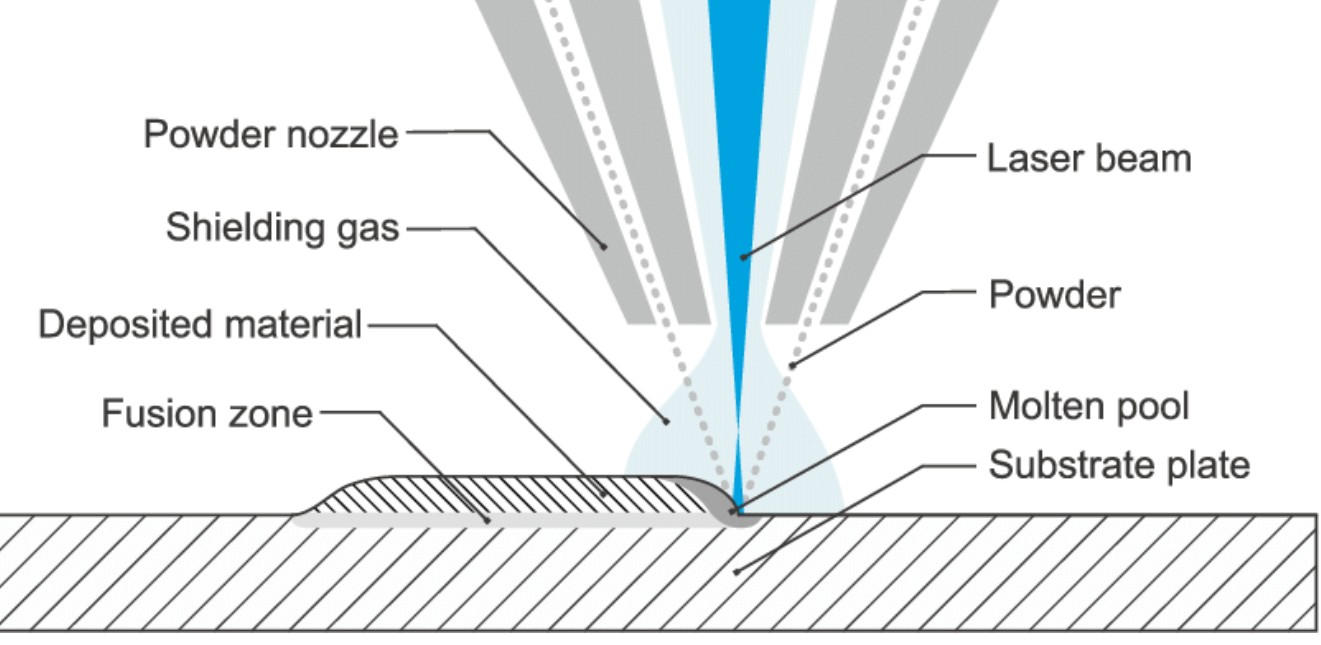

**Figure 8.** Schematic illustration of the DED process [105].

### 2.3. Laser Foil Printing (LFP)

Instead of using powder as in most commercial AM technologies, solid-state material can also be used to feed the process. Laser foil printing (LFP) or laminated object manufacturing uses metallic foils. It combines both additive and subtractive methods [28].

The technology is composed of two smaller processes, welding and cutting. The foil undergoes two types of laser welding. The laser spot-welding will anchor the new foil on top of the substrate or the semi-finished model in order to avoid distortion. The major control parameters in laser spot welding include laser power, pulse duration, spot size, and spot distance. Then, a laser raster-scan welding will fully weld the arriving foil on the previous foil or substrate. After welding a new foil on top of the semi-finished part, a cutting laser will act on the inner and outer regions of the foil to remove the unwanted redundant foil. This mechanism is repeated until the desired 3D model is built. The mechanical properties of the part are usually defined by the laser welding (especially laser raster-scan welding) and the surface properties by the laser cutting process [106]. Schematic illustration of the LFP system and the steps are presented in Figure 9.

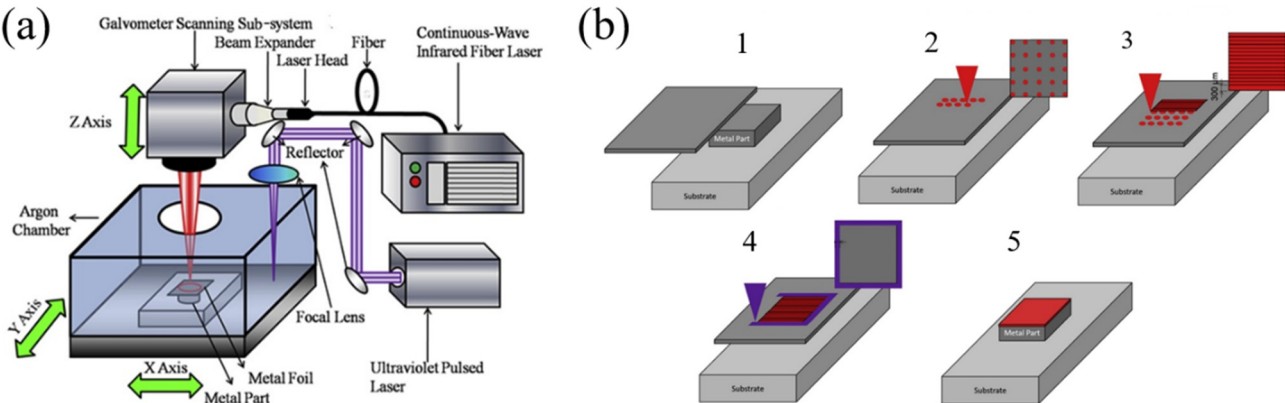

**Figure 9.** Schematic illustration of (**a**) the LFP system, (**b**) LFP steps. Reproduced from [106], with permission from Elsevier, 2021.

The main advantages of this technology compared to classical AM technologies are the low cost and high flexibility. BMGs were produced with the LFP process starting from amorphous foils [28,107–110], as illustrated in Figure 10. The major problem is the need of a crystalline metallic substrate, since a thick amorphous substrate is obviously difficult to obtain. The weld between the first layer and the substrate is then often problematic due to the different chemical compositions and thermal properties. Partial crystallization of the first foil layer may occur as a result of significant diffusion between the melted zones of the substrate and the foil. However, the subsequent foils do not suffer from this issue and can be fully amorphous [107–109].

### 2.4. Thermal Spray 3D Printing (TS3DP)

Thermal spray is a widely used technique for coating parts susceptible to wear and corrosion [42]. A high-velocity oxygen-fuel (HVOF) gun is used to make the material semi-molten, and it is blown away toward a substrate. As a result, the semi-molten material is solidified in a short fraction of time and the high cooling rate allows processing alloys such as MG. If multiple coating passes are applied, bulk parts can be made using the thermal spray technique, and this technique can be called thermal spray 3D printing (TS3DP). Since one of the advantages of AM techniques is the possibility of fabricating complex parts, a make can be placed between the gun and the substrate to selectively deposit materials in the desired locations. The main advantage of TS3DP is lower residual stress compared to the techniques where melting is required. Two Fe-based BMGs have been fabricated via TS3DP so far [42,111,112].

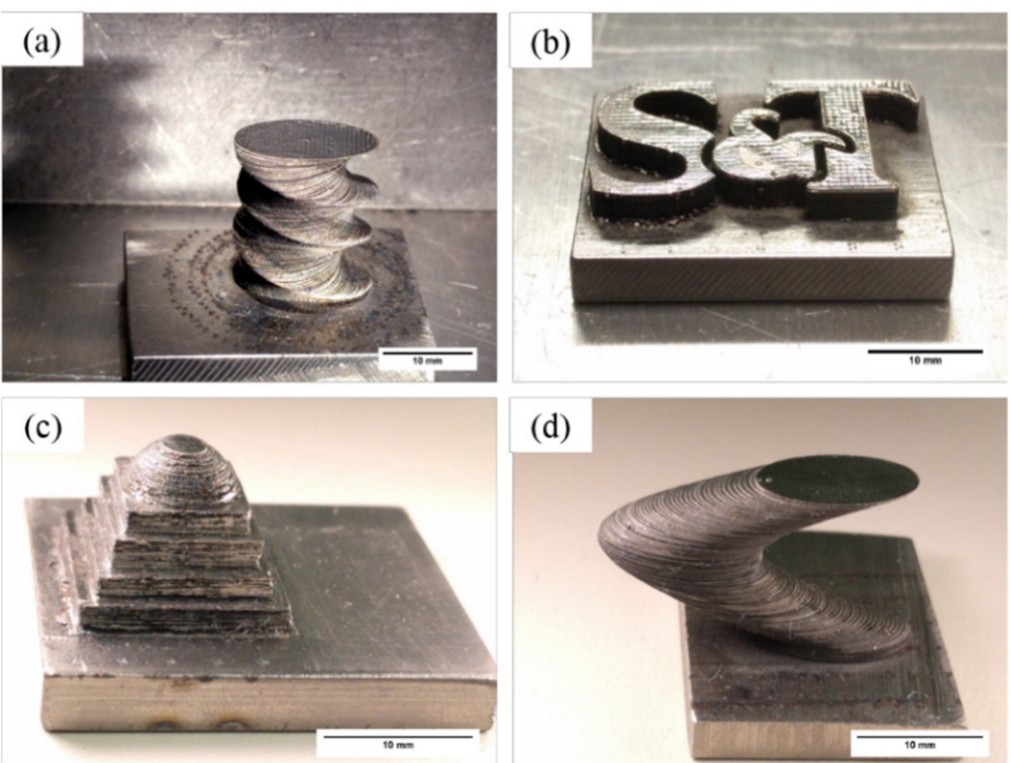

**Figure 10.** BMG parts fabricated via LFP with different geometries. Reproduced from [109], with permission from Elsevier, 2021.

*2.5. Ultrasonic Additive Manufacturing (UAM)*

Ultrasonic additive manufacturing (UAM) is a solid-state AM technology that uses ultrasonic vibration in order to form ultrasonic bonding (welding). The excess material is then removed using CNC machining, to achieve the desired geometry [113]. The schematic illustration of UAM steps is exhibited in Figure 11. The advantages of UAM over other AM methods are: (1) dissimilar materials can be joined, (2) reduced residual stresses due to the absence of melting, and (3) the ability to embed sensitive/functional components in the bulk of samples [114].

Wu et al. [115] used UAM to fabricate a Ni-base BMG, which resulted in an amorphous structure and a hardness value similar to the feedstock (thin Ni-MG strips).

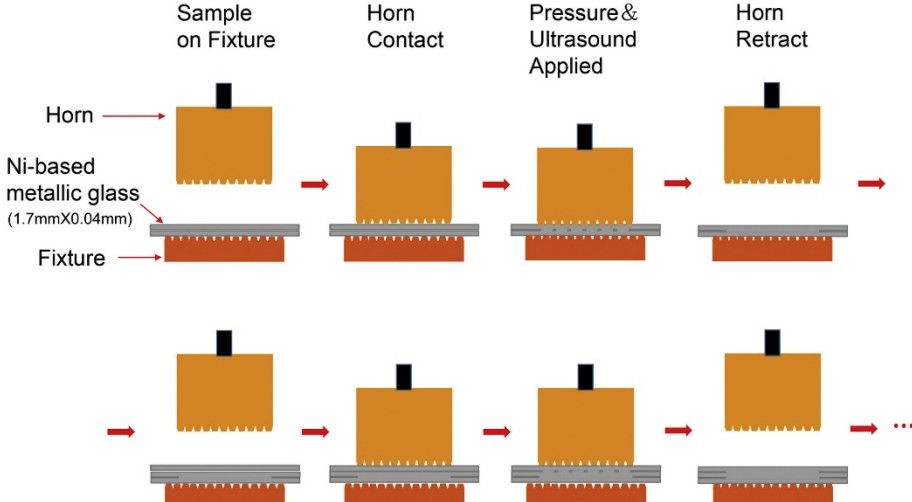

**Figure 11.** Schematic illustration of the UAM process. Reproduced from [115], with permission from Elsevier, 2021.

## 2.6. Fused Filament Fabrication (FFF)

Fused filament fabrication (FFF) is a semi-solid AM technique, which is mainly used for the fabrication of polymeric parts [116]. The schematic of the FFF process is illustrated in Figure 12a. The feedstock is forced into the extruder head and heated (via Joule effect) to a temperature at which the viscosity is reduced. The substrate is heated to form a better bond with the initial layer of the extruded material. To have a good bonding between extruded layers, the temperature of the previously extruded layer should be locally increased. Therefore, a metal brush (blue part in Figure 12a) precedes the extruder by a short distance [117].

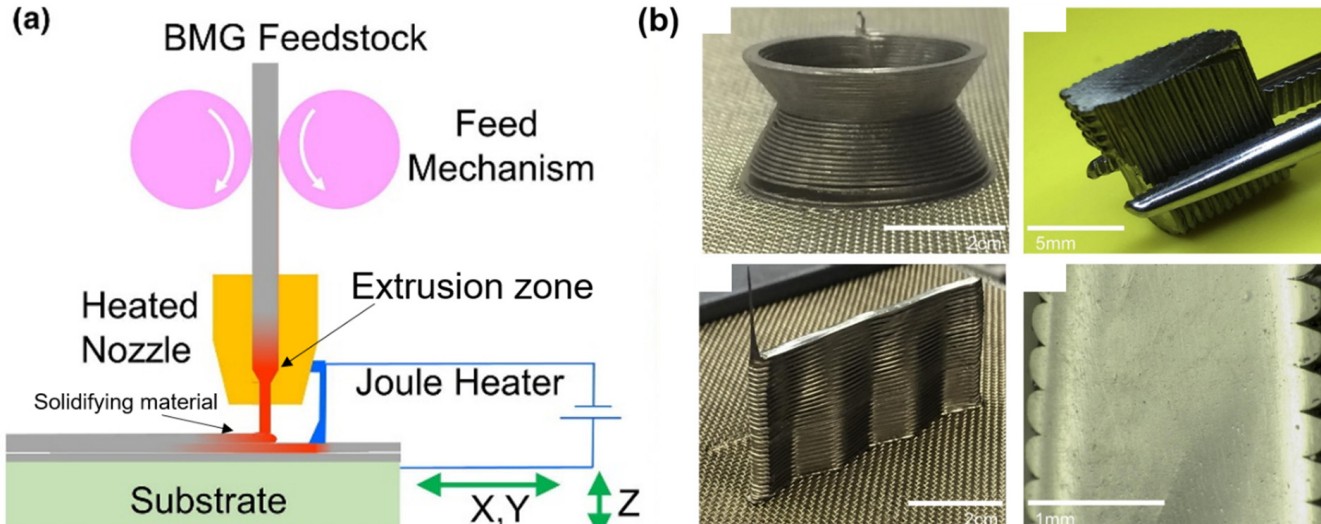

**Figure 12.** (**a**) Schematic illustration of the FFF process and (**b**) large Zr-based BMGs fabricated via FFF. Modified from [117], with permission from Elsevier, 2021.

The advantages of the FFF method over other AM methods are: (1) no protective atmosphere is required, (2) flowability of the feedstock is not an issue, (3) residual stresses are lower, and (4) lower cost compared to laser-based AM techniques.

When BMGs are heated above their $T_g$, their viscosity is significantly reduced, and they can be easily formed. Gibson et al. [117] used FFF for producing large Zr-based BMG parts (see Figure 12b) with acceptable mechanical properties.

To use FFF for manufacturing BMGs, the time spent at high temperatures should be minimized to prevent crystallization. Only BMGs with good GFA can be processed by this method.

## 3. Materials Used in AM of BMGs

AM of BMGs attracted much attention in the last eight years. As shown in Figure 13, among all BMGs, Zr-based ones were the most studied alloys due to their improved GFA compared to Fe, Al, Cu, and Ni-based BMGs, and the lower cost compared to Ti and precious metals.

### 3.1. Based on Zirconium

The first Zr-based BMG with a critical diameter larger than 10 mm was synthesized in 1993 at Caltech, USA [118]. Adding Be allowed Lou et al. [119] to produce a Zr-BMG with a critical diameter of 73 mm, using copper mold casting. They have applications in sporting goods (golf clubs, tennis rackets, baseball bats, etc.), coil-shaped and helical springs, diaphragms for pressure sensors, screws and bolts, medical devices (pacemaker, knee-replacement devices), and surface coating and cladding [120]. In 2016, the first Zr-based BMG was fabricated via AM [85]. A lot of attention has been attracted to the production of Zr-based BMGs in the last five years [5,28,32–34,36,40,41,45–47,50,51,53,55–

58,60,61,63,66,67,69–72,74,77–82,85,86,91–99,101–104,107–110,117]. Thirty-five percent of the published papers in AM of Zr-based BMGs are related to $Zr_{59.3}Cu_{28.8}Al_{10.4}Nb_{1.5}$, whose industrial name is AMZ4. The reason is that AMZ4 is an industrial-grade alloy, which contains more impurities compared to lab-grade alloys, and this makes AMZ4 an economical alloy. The impurities, such as a high amount of oxygen content, can cause significant problems, which will be discussed in Section 4.6. Figure 14 shows some Zr-based BMG parts fabricated via the LPBF process.

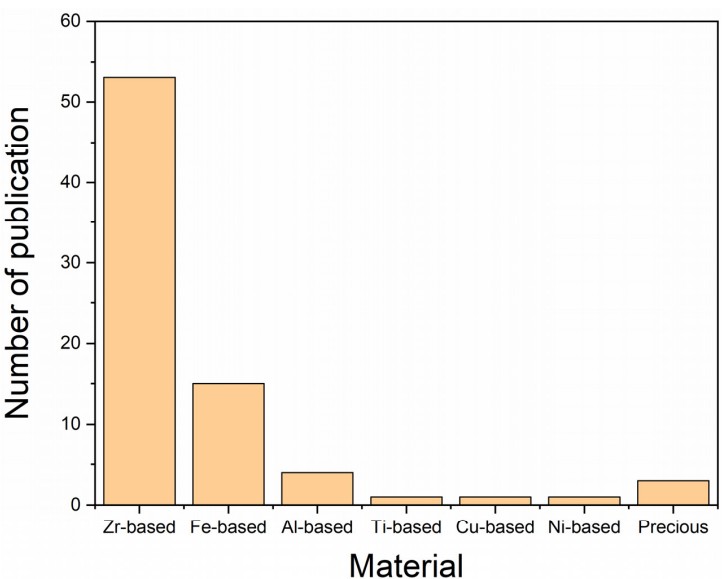

**Figure 13.** Number of papers related to AM of particular BMG materials from 2013 to April 2021.

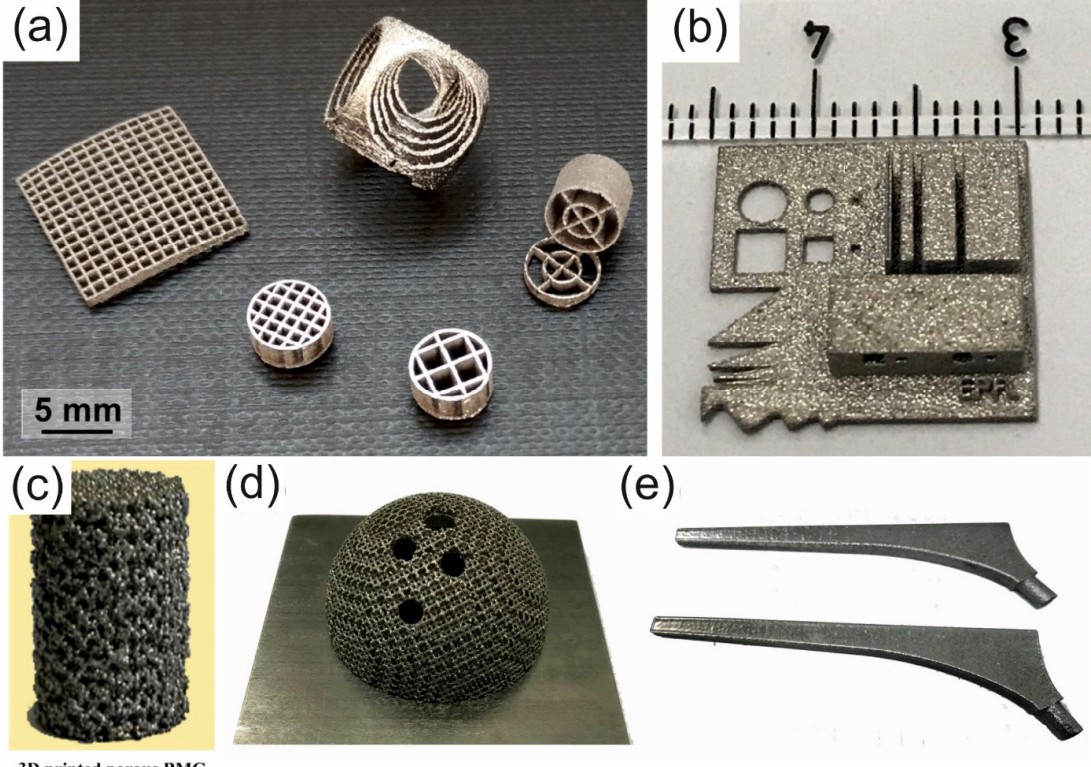

**Figure 14.** Samples with complex geometries in (**a**) Reproduced from [33], with permission from Elsevier, 2021. (**b**) Reproduced from [32], with permission from Elsevier, 2021. (**c**) porous BMG part, and (**d**,**e**) LPBF parts used for biomaterials applications. Reproduced from [50], with permission from Elsevier, 2021.

### 3.2. Based on Iron

The first Fe-based MG (Fe–Al–Ga–P–C–B) was developed in 1995 [121]. C and B were required for glass-forming ability. Cr and/or Mo were added to Fe-based BMGs to improve the corrosion resistance by forming a passive layer [2]. Fe-based MGs have excellent magnetic properties such as low coercivity, high magnetostriction, and magnetic permeability. These properties are attributed to the lack of crystal-related defects, such as dislocations and grain boundaries [25]. Due to the above-mentioned properties, Fe-based MGs are used in sensors, actuators, transformers, and communication equipment [122].

Crystallization decreases the soft magnetic properties of the amorphous structure [123] and conventional fabrication methods, such as casting, cannot provide sufficiently high cooling rates to produce large amorphous parts. This explains that the application of Fe-based MGs is often restricted to small devices (<2 mm). AM technologies provide the opportunity to overcome this limitation.

The first published paper on LPBF of BMGs, in 2013, considered an Fe-based MG (see Figure 15) [43]. Although the fabricated part was partially crystallized, it paved the way for applying AM techniques to BMGs. Several Fe-based BMGs with different chemical composition have been produced using AM methods [35,37,38,42–44,48,49,59,62,64,68,75,76,100]. Mahbooba et al. [37] produced a Fe-based BMG via LPBF with a diameter 15 times the critical casting diameter (<2 mm) and clearly showed the AM method potential for the fabrication of BMGs.

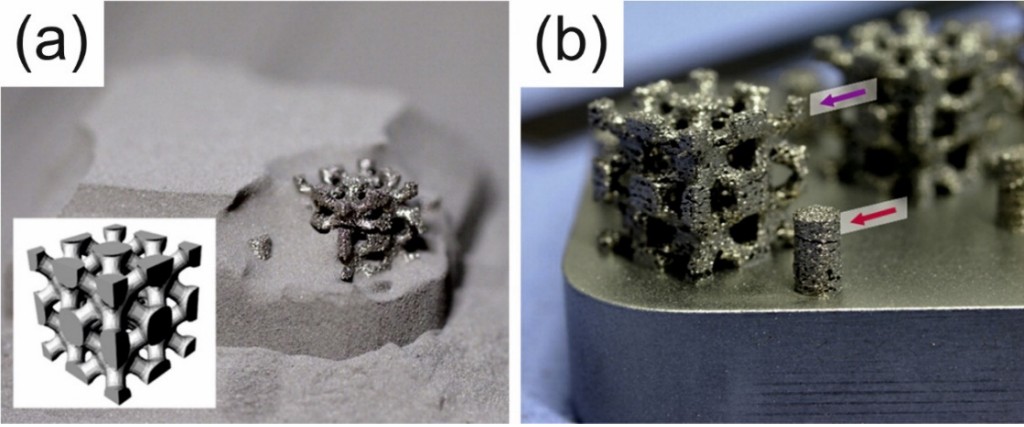

**Figure 15.** Fe-based BMG parts fabricated via LPBF (**a**) inside the LPBF machine and (**b**) after removing the remaining powder. Modified from [43], with permission from Elsevier, 2021.

### 3.3. Based on Aluminum

Al-based alloys have gained much attention in engineering applications due to their high strength to weight ratio and corrosion resistance [124]. Their strength can be improved by forming an amorphous structure. The first Al-based MG (Al-(Fe or Co)-B) was developed in 1981 [125]. Al-based MGs have low GFA, and producing amorphous parts with a diameter larger than 1 mm using copper mold casting took almost three decades ($Al_{86}Ni_6Y_{4.5}Co_2La_{1.5}$) [126]. Additive manufacturing is another production method for Al-based MGs. So far, there are four studies on LPBF of three Al-based B/MGs, $Al_{86}Ni_6Y_{4.5}Co_2La_{1.5}$ [65,83], $Al_{85}Nd_8Ni_5Co_2$ [84], and $Al_{85}Ni_5Y_6Co_2Fe_2$ (Figure 16) [54].

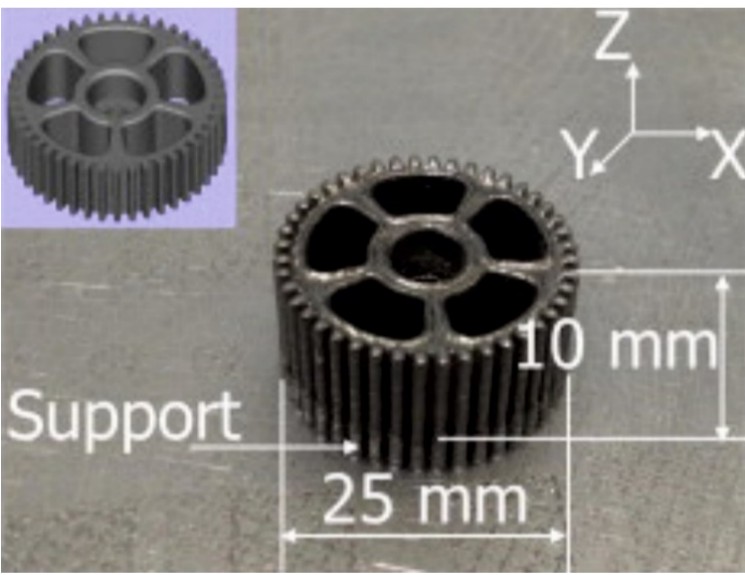

**Figure 16.** Gear made of an Al-based bulk metallic glass composite (BMGC) using the LPBF process. The inset shows the CAD file of the gear. Modified from [43], with permission from Elsevier, 2021.

### 3.4. Based on Copper

Cu-based BMGs are known for their low fabrication cost due to the cheaper base metal compared to precious metals. They have applications in corrosive environment (e.g., marine) [2]. One of the well-known Cu-based BMG systems is Cu-Zr-Al. The critical casting diameter of these alloys is around 3 mm, and they have a compressive strength higher than 2 GPa, together with a plastic strain around 0.2% [127].

In 2019, a ternary Cu-based BMG, $Cu_{50}Zr_{43}Al_7$, was fabricated for the first time via AM [52]. Thanks to the AM technology, in addition to a disk with a diameter of 20 mm, parts with complex shapes and geometries were obtained (Figure 17).

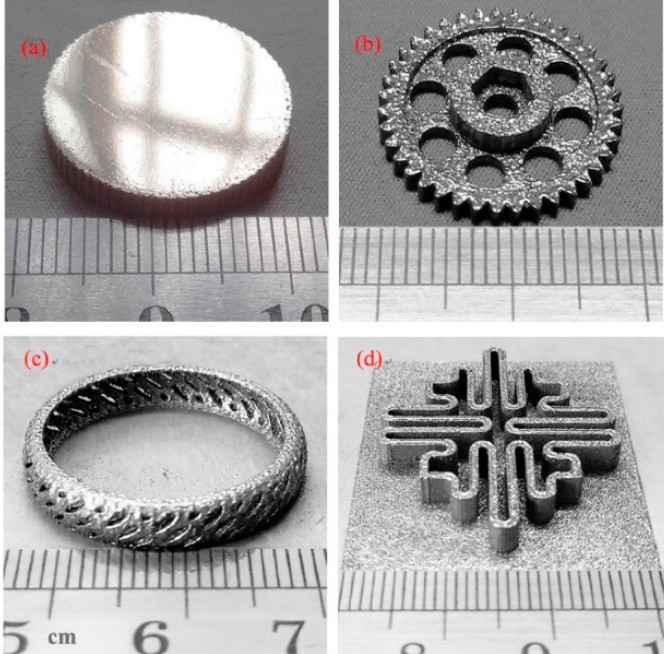

**Figure 17.** Cu-based BMG parts with different geometries fabricated via LPBF. Reproduced from [52], with permission from Elsevier, 2021.

### 3.5. Based on Nickel

Ni-based crystalline alloys have high corrosion resistance. One of the ways to further improve their corrosion resistance is to produce them in an amorphous state [2]. The increased hardness also opens applications where wear resistance is required [128], such as micro gears [129]. One of the well-known Ni-based BMGs is NiCrBSiFe system, which is used as a coating for high-temperature applications [130]. In 2019, the first Ni-based BMG ($Ni_{82.2}Cr_7B_3Si_{4.8}Fe_3$) was fabricated via an AM method [115]. Thin strips of the Ni-based MG prepared by melt spinning were used for UAM. An amorphous structure with a high hardness ($8.55 \pm 0.96$ GPa) was achieved.

### 3.6. Based on Titanium

Ti-based alloys have been widely used in aerospace, orthopedic prostheses, and dental implants, due to their high strength to weight ratio, and biocompatibility. Ti-based BMGs have lower Young's modulus than their crystalline counterparts, closer to the human bones, which prevents stress shielding and consequent damage. Be is one of the elements increasing the GFA of Ti-based BMGs leading to a critical casting diameter beyond 14 mm. However, it is a toxic element, and alloys containing Be are not recommended for implant applications. A new Ti-based BMG ($Ti_{47}Cu_{38}Zr_{7.5}Fe_{2.5}Sn_2Si_1Ag_2$), Be and Ni-free, was developed in 2015 with a critical diameter of 7 mm [131]. In 2018, the first additively manufactured Ti-based BMG ($Ti_{47}Cu_{38}Zr_{7.5}Fe_{2.5}Sn_2Si_1Ag_2$) was fabricated via LPBF (Figure 18) [39]. Complex 3D parts were produced with an XRD amorphous structure.

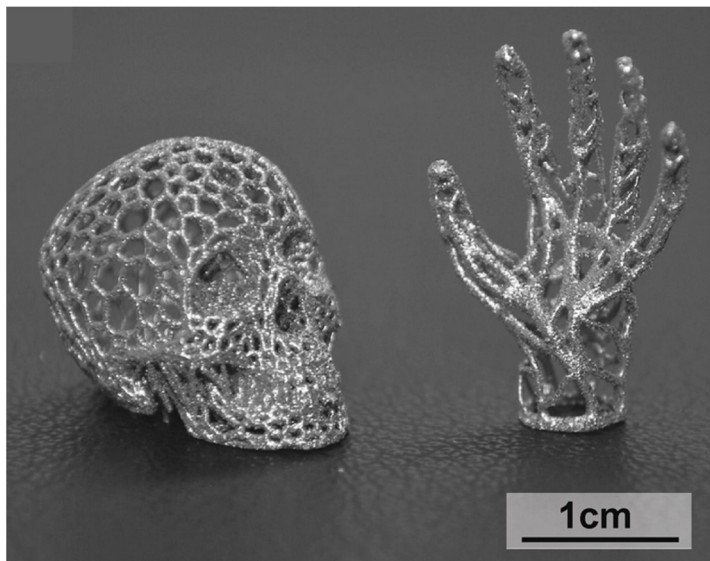

**Figure 18.** Ti-based BMG parts fabricated via LPBF. Modified from [39], with permission from Elsevier, 2021.

### 3.7. Based on Precious Metals

Precious metals have a variety of applications, especially in the watch and jewelry industries, related to their charming appearance, oxidation resistance, and non-allergic nature. Hardness and wear resistance are the two most important mechanical properties for the materials used in these sectors [4,132–134]. For example, hardness higher than 300 HV is needed for watch components where wear and scratch resistance is required [135]. Since pure precious metals are soft and not resistant to wear and scratching, precious-metals-based alloys have been developed to obtain materials with improved (mechanical) properties. However, desirable mechanical properties in precious alloys are not easy to achieve, even after several thermo-mechanical treatments [133,136–140]. In addition, some of the precious alloys are difficult to process. For instance, Pt-based alloys have a melting temperature higher than 1800 °C, and at such a high temperature it becomes challenging

to prevent a reaction with the crucible, oxidation or tarnishing. Moreover, the shrinkage happening due to solidification and solid–solid phase transformations reduces the accuracy of the cast parts [2].

Bulk metallic glasses (BMGs) present a promising alternative to overcome the intrinsic limitations of their crystalline counterparts. In addition, they are lighter and have a higher elastic limit. The first metallic glass was produced in 1960, based on gold [141]. Later, BMGs based on other precious metals such as Pd [142], Pt [143], and Ag [144] were developed. There are three studies in the literature concerning AM of precious BMGs [73,89,90]. A Pt-based MG ($Pt_{57.3}Cu_{14.7}Ni_{5.3}P_{22.7}$) powder was consolidated using pulsed SLS, and a partially crystallized structure was achieved [89,90]. Recently, using the LPBF method, a dense, amorphous, and crack-free Pd-based BMG ($Pd_{43}Cu_{27}Ni_{10}P_{20}$) was fabricated [73], while exhibiting good mechanical and aesthetic properties.

## 4. Challenges

### 4.1. Parameter Optimization

Many of the LPBF processing parameters listed in Figure 7 are shared by several laser-based AM techniques. In this sub-section, the most studied parameters in AM of BMGs and their effect on the microstructure/mechanical properties are discussed. These parameters are laser power ($P$), scanning speed ($v$), hatching distance or scan spacing ($h$), and scanning strategy or scan pattern. A few studies investigated other parameters such as the layer thickness ($t$), the border/contour parameters (power), powder size, inert gas, and pre-annealing of the powder. All these studies are listed in Table 2.

**Table 2.** Studied parameters used in AM of BMGs.

| Material | Ref. | Method | Parameters |
|:---:|:---:|:---:|:---:|
| AMZ4 | [5] | LPBF | $P, h, t$ |
| AMZ4 | [56] | LPBF | $P$ |
| AMZ4 | [69] | LPBF | Inert gas |
| AMZ4 | [78] | LPBF | Border power and distance |
| $Al_{85}Ni_5Y_6Co_2Fe_2$ | [54] | LPBF | Scanning strategy |
| $Fe_{68.3}C_{6.9}Si_{2.5}B_{6.7}P_{8.7}Cr_{2.3}Mo_{2.5}Al_{2.1}$ | [76] | LPBF | $P, v$ |
| $Zr_{52.5}Ti_5Cu_{17.9}Ni_{14.6}Al_{10}$-LM105 | [85] | LPBF | $P, v, h$, Scanning strategy |
| $Zr_{52.5}Cu_{17.9}Ni_{14.6}Al_{10}Ti_5$-LM105 | [33] | LPBF | $P, v, h$ |
| $Fe_{43.7}Co_{7.3}Cr_{14.7}Mo_{12.6}C_{15.5}B_{4.3}Y_{1.9}$ | [35] | LPBF | $P, v$ |
| $Fe_{43.7}Co_{7.3}Cr_{14.7}Mo_{12.6}C_{15.5}B_{4.3}Y_{1.9}$ | [38] | LPBF | $P, v$ |
| FeCrMoBC | [42] | LPBF | $P, v, h$, Scanning strategy |
| FeCrMoBC | [59] | LPBF | Scanning strategy |
| FeCrMoBC | [100] | DED | $P, v$ |
| $Fe_{49.60}Cr_{18.10}Mn_{1.90}Mo_{7.40}W_{1.60}B_{15.80}C_{3.82}Si_{2.40}$ | [44] | LPBF | $P$ |
| $Zr_{57.4}Ni_{8.2}Cu_{16.4}Ta_8Al_{10}$ | [46] | LPBF | $P, v$ |
| $Cu_{50}Zr_{43}Al_7$ | [52] | LPBF | $P, v$ |
| $Fe_{73.7}Si_{11}B_{11}C_2Cr_{2.28}$ | [62] | LPBF | Scanning strategy |
| $Cu_{46}Zr_{47}Al_6Co_1$ | [60] | LPBF | $h$ |
| $Zr_{55}Cu_{30}Al_{10}Ni_5$ | [61] | LPBF | pre-annealing of the powder |
| FeCoBSiNb | [68] | LPBF | $P, v$ |
| $Zr_{55}Cu_{30}Al_{10}Ni_5$ | [93] | LSF | Powder size |
| $Zr_{44}Ti_{11}Cu_{10}Ni_{10}Be_{25}$ | [94] | LSF | $v$ |

**Table 2.** *Cont.*

| Material | Ref. | Method | Parameters |
|---|---|---|---|
| $Zr_{50}Ti_5Cu_{27}Ni_{10}Al_8$ | [97] | DED | $P$, $v$ |
| $Zr_{51}Ti_5Cu_{25}Ni_{10}Al_9$ | [101] | DED | $P$ |
| $Zr_{39.6}Ti_{33.9}Nb_{7.6}Cu_{6.4}Be_{12.5}$, DH3 | [102] | DED | $P$, $v$ |
| Zr51 BMG | [104] | DED | $P$, $v$ |
| $Zr_{52.5}Ti_5Cu_{17.9}Ni_{14.6}Al_{10}$-LM105 | [107] | LFP | $P$, $v$ |
| $Zr_{52.5}Ti_5Cu_{17.9}Ni_{14.6}Al_{10}$-LM105 | [108] | LFP | $v$, $h$ |
| $Zr_{52.5}Ti_5Cu_{17.9}Ni_{14.6}Al_{10}$-LM105 | [109] | LFP | $v$ |
| $Zr_{52.5}Ti_5Cu_{17.9}Ni_{14.6}Al_{10}$-LM105 | [110] | LFP | $v$ |

Sohrabi et al. [5] investigated the effects of $P$, $h$, and $t$ on the amorphous content of an AMZ4 coating. As $P$ was increased, the amorphous content decreased because of the increasing volume energy density (VED), which is

$$VED \left( J/mm^3 \right) = \frac{P}{v.h.t} \tag{1}$$

When the hatching distance increased (VED decreased), there was lower overlap between the adjacent laser tracks, which resulted in lower temperatures and increased amorphous fraction. Increasing the layer thickness from 20 μm to 30 μm (decreased VED) while keeping all other parameters constant increased the amorphous content. One should note that a minimum VED is required to melt the powder, have a high density, and create good bonding with the previous layers and the adjacent laser tracks. Pauly et al. [33] mapped the relative density of a Zr-based BMG fabricated via LPBF as a function of $P$, $v$, $h$, and VED (Figure 19). They showed that as the VED increased, there was more chance for crystallization.

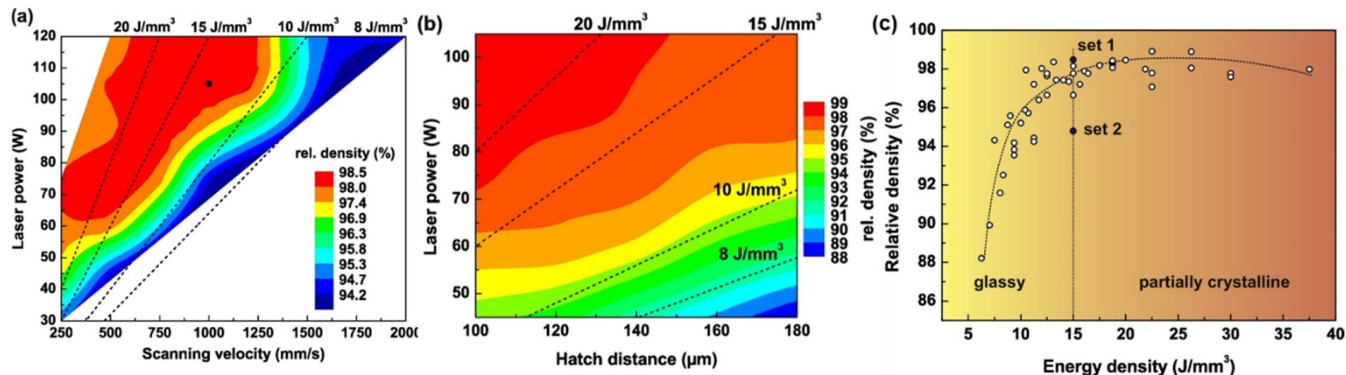

**Figure 19.** (**a**) Contour map of relative density of a Zr-based BMG fabricated via LPBF as a function of $P$ and $v$, (**b**) contour map of relative density as a function of $P$ and $h$, and (**c**) relative density as a function of VED. Modified from [33], with permission from Elsevier, 2021.

When dealing with BMGs, people use more often the VED [33,72,85] than other concepts such as normalized enthalpy, because it gives a global account of the laser energy received by the material, which has direct consequences on the crystallization behavior. The normalized enthalpy, on the other hand, measures the net energy input at the scale of the melt pool [69,73], accounting for material thermo-physical parameters but neglecting subsequent heating coming from adjacent laser tracks.

For crystalline alloys, low VED values (yellow region in Figure 20) result in a porous structure that is generally not desired. The porosity content is due to LoF formation. There is a processing window (black double-sided arrow) where the porosity content is typically lower than one percent. Increasing VED values (white double side arrow) give rise to the

creation of porosity due to keyhole melting [145]. However, the situation for BMGs is more constrained. Apart from porosity, crystallization must also be avoided. Therefore, the process window for BMGs is even smaller than for conventional crystalline alloys, because the VED value should be sufficiently large to produce a dense sample, but not larger than a threshold value that induces non-negligible crystallization (green region). Although increasing the VED beyond the green region (until the vertical blue dashed line) can result in a sample with lower porosity, it also induces crystallization, with consequences on brittleness and cracking of the part. Identifying the narrow processing window requires very careful optimization.

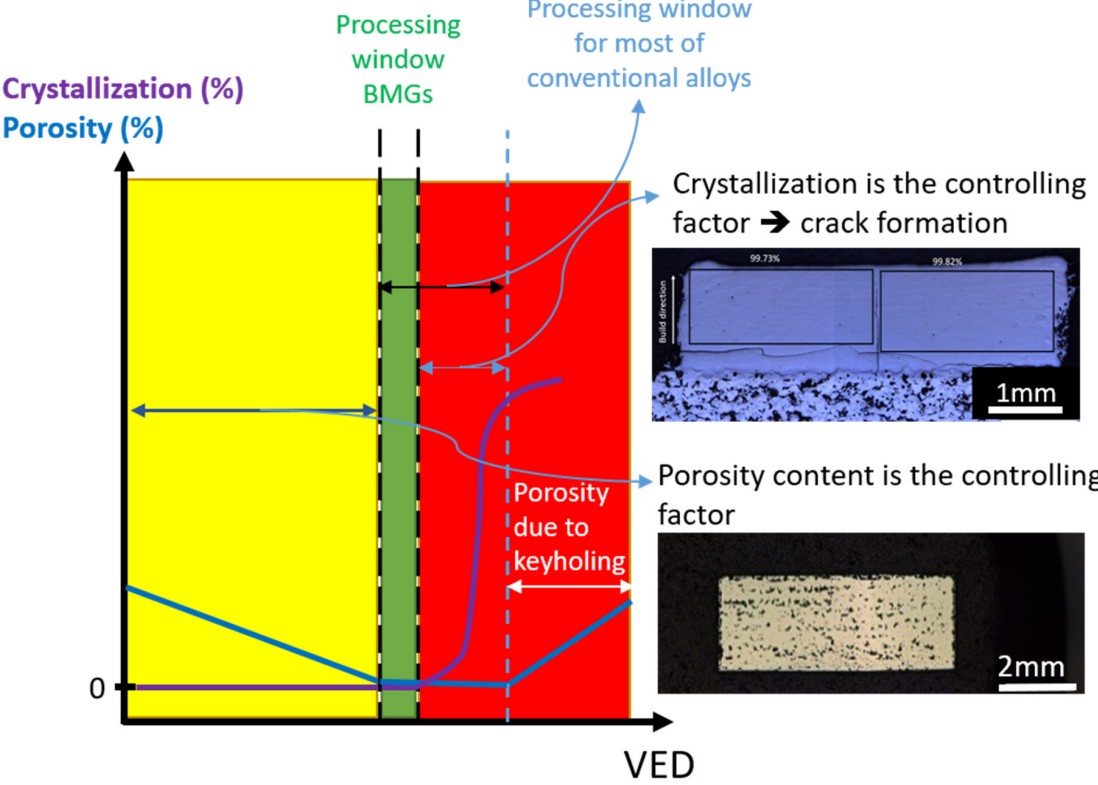

**Figure 20.** Schematic graph contrasting the AM processing windows of BMGs and conventional crystalline alloys.

Li et al. [54] used a remelting scanning strategy for a cracked Al-based BMG fabricated via LPBF. The treatment induced a stress relief, which could stop crack propagation. In another study on a Zr-based BMG [85], the authors demonstrated that remelting could homogenize the elements distribution, leading to an amorphous structure. Wegner et al. [69] investigated the effect of inert gases ($N_2$, Ar, and $Ar_{98}H_2$) on the density and amorphous content of AMZ4 fabricated via LPBF. They showed that, for the same energy input, Ar resulted in a sample with higher density and amorphous fraction. Processing parameters are typically changed near the part edges (or border) such as to further improve the defect content and the surface state. The laser power of the contour zone, if not optimized, can significantly affect mechanical properties, essentially due to the presence of LoFs in the near-surface regions [78].

### *4.2. Defects*

As mentioned earlier, one of the main drawbacks of AM techniques is defect formation in the AM fabricated parts. These defects are categorized into three groups, porosity (spherical), LoF, and crack, which are explained in more details below.

### 4.2.1. Porosity

The main origins of porosity in fabricated parts are (1) existing pores in the feedstock (powder) [146], (2) evaporation of volatile substances or low melting elements in the melt pool [147], and (3) pores induced by keyhole instability [148]. In addition, Hojjatzadeh et al. [146] detected three other mechanisms of pore formation in the LPBF process, for which the reader is referred to Ref [146].

A majority of studies dedicated to AM of metals attempt to maximize the density of the parts and reduce the porosity content. However, as discussed above, for BMGs, the processing window is narrower with respect to crystalline alloys due to the occurrence of crystallization at high energy inputs (Figure 20). Porosities are stress risers and can lead to premature failure, especially under fatigue loading [149]. Taking the example of AMZ4 fabricated via LPBF, a decrease of porosity content from 5.24% to 0.26% led to a 26% increase in tensile strength (reaching 1243 MPa) [77]. Shi et al. [63] characterized the porosity content of AMZ4 fabricated via LPBF and showed that the sample with 0.45% porosity content had a hardness comparable to the as-cast sample, while 8% of porosity decreased the hardness (HV5) by 17%. Sohrabi et al. [73] showed that 0.4% of porosity in a Pd-based BMG sample fabricated via LPBF resulted in a compressive strength 14% lower than the as-cast material (with a slightly different chemical composition). The fracture surface showed crack propagation through porosities, which confirms the idea that porosities are stress risers. However, according to Deng et al. [70], small porosities can act as a second phase and deflect the direction of shear bands, which then becomes beneficial for ductility. Sohrabi et al. [78], distinguished the small beneficial ones from the detrimental ones: larger or open to surface.

In crystalline alloys, the porosity content can be decreased by hot isostatic pressure (HIP) treatment [150]. This is not an option for BMGs due to the need to prevent crystallization. Thanks to the development of micro-X-ray computed tomography (μCT) techniques, volumetric quantification of porosities with good resolution has been demonstrated [32,33,39,46,50,57,63,67,70,73,77,78].

### 4.2.2. Lack of Fusion (LoF)

Lack of fusion (LoF) defects are formed as a result of a low input energy, unable to completely melt the feedstock, e.g., powder. High scanning speed and low laser power promote LoF formation. Due to their large size (hundreds of microns) and irregular shapes (see Figure 21), they can cause stress concentration and act as crack initiation sites. There are several studies on AM of BMGs that blamed LoFs for the premature failure in tensile [78], compression [33], bending [34], impact toughness [78], and fatigue [79] tests, which are discussed more in details in the section related to mechanical properties. To mitigate LoFs in the near-surface regions, the border (or contour) processing parameters, such as the power, must be optimized. This strategy resulted in 28%, and 27% improvements in impact toughness and tensile strength, respectively, for AMZ4 fabricated via LPBF [78].

### 4.2.3. Crack

BMGs are very prone to cracking because they tolerate no or very limited amount of plastic deformation. Besides, the very high and local heating and cooling rates in AM processes induce large residual stresses which can lead to cracking for materials with low ductility. Many studies on AM of BMGs reported cracking [33,35,37,38,42,44,48,54,62,64,65,68,69,76,104,107,108]. Most of these studies relate to Fe-based BMGs, which are intrinsically quasi-brittle (tougher than brittle materials such as ceramics). Cracking typically occurs as a result of crystallization, whose induced stresses combine with the thermal residual stresses. Li et al. [54] could stop crack propagation in an Al-based BMG by remelting, acting as a stress relieve treatment. Li et al. [107,108] detected cracking in the first layer of LFP, in a Zr-based BMG deposited on a Ti-substrate, and attributed it to the intermixing (dilution) at the interface, leading to the formation of several intermetallics. Zou et al. [64] added Cu to a Fe-based MG powder

to prevent crack formation during LPBF processing. Cu could form a ductile phase and reduce stress concentration by changing the distribution of elements.

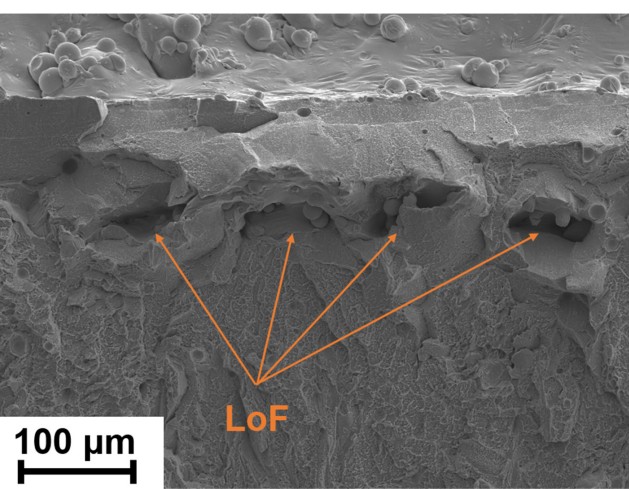

**Figure 21.** Lack of fusion (LoF) defects close to the external surface of an LPBF AMZ4 specimen after Charpy impact test.

### 4.3. Residual Stress

High levels of residual stresses can cause delamination, distortion, and cracking in the additively manufactured parts [151], especially for BMGs with low ductility. Xing et al. [41] investigated the level of residual stresses in an AMZ4 bar fabricated on a comb-shape support structure, using finite element (FEM) simulations (see Figure 22). The scanning strategy used for the fabrication of the bar, the thickness of the bar, and the preheating, affected the level of residual stresses. As the thickness of the bar increased from 0.96 mm to 2.88 mm, the maximum level of residual stresses increased from 592 MPa to 710 MPa, and preheating to 250 °C (sub-$T_g$) reduced the maximum level from 592 MPa to 279 MPa. One should notice that sub-$T_g$ heating can result in the growth of pre-existing nanocrystals in the amorphous matrix, if there are any [79].

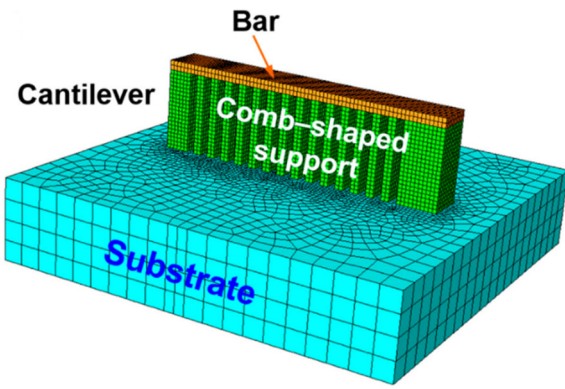

**Figure 22.** Three-dimensional model of a bar with comb-shaped support to measure the level of residual stresses [41].

When BMGs crystallize, they become more brittle, and the presence of tensile residual stresses (TRS) makes them more prone to cracking [5]. Sohrabi et al. [5,32] showed that residual stresses cause distortion in the fabricated sample, while the amorphous structure is able to tolerate a high level of residual stresses without cracking.

Ouyang et al. [86] simulated by FEM the residual stresses at the melt pool scale, in a Zr-based BMG. The maximum compressive residual stresses (CRS) were much lower than the compressive strength of the material. However, when a porosity was introduced in the simulation, the resulting stress concentration increased maximum CRS up to the

compressive strength. These computations illustrate how defects can act as an initiation site for cracking, in agreement with Li et al. [38].

### 4.4. Homogeneity of the Structure

Microstructure homogeneity in parts fabricated via AM methods is always challenging because of the non-homogeneous thermal histories [29,30]. Chemical heterogeneity can be checked using an electron probe microanalyzer (EPMA) [62,68,83,85,92] or the electron dispersive spectroscopy (EDS) [42,64,73,84,107,108], while structural heterogeneity is usually revealed by nanohardness mapping [32,68,70,85]. Using nanoindentation and EPMA, Li et al. [85] could easily detect inhomogeneities in samples containing crystals, while an XRD amorphous sample showed a more uniform structure (see Figure 23). A nanohardness map of 1600 indents performed on AMZ4 fabricated via LPBF is presented in Figure 24a [32]. The hardness is relatively uniform, with however reddish spots attributed to the presence of nanocrystals. A single Gaussian distribution of hardness in Figure 24b indicates the presence of one major phase. The measured average hardness was $5.13 \pm 0.25$ GPa.

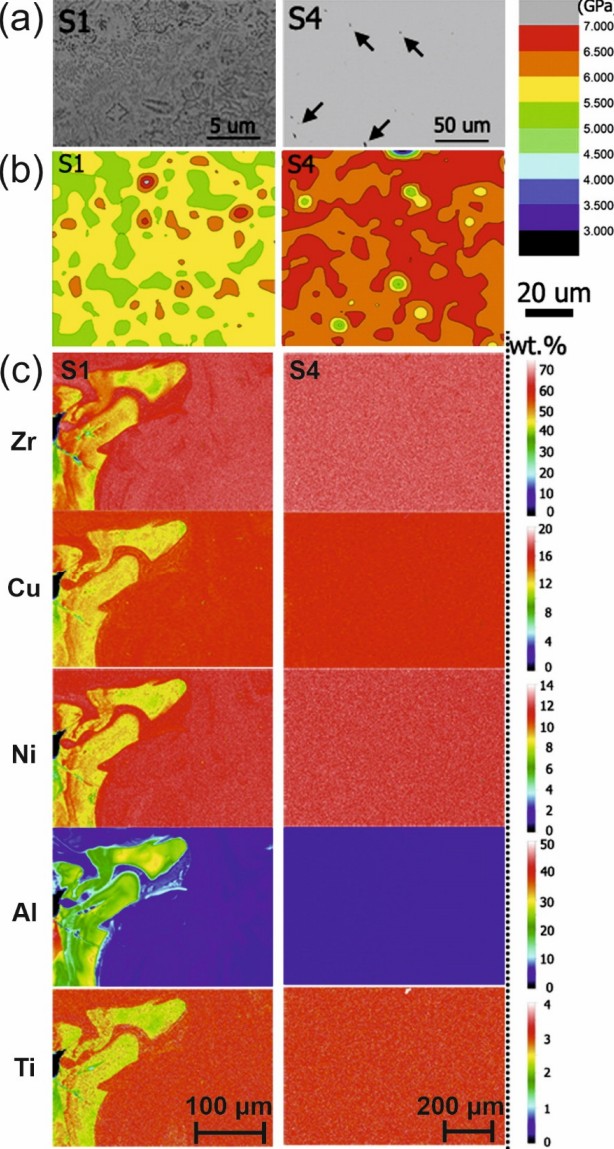

**Figure 23.** (**a**) Back-scattered electron (BSE) images of a Zr-based BMG fabricated with two sets of parameters, S1 and S4, (**b**) nanohardness maps of the two samples, and (**c**) electron probe microanalyzer (EPMA) maps of the two samples. Modified from [85], with permission from Elsevier, 2021.

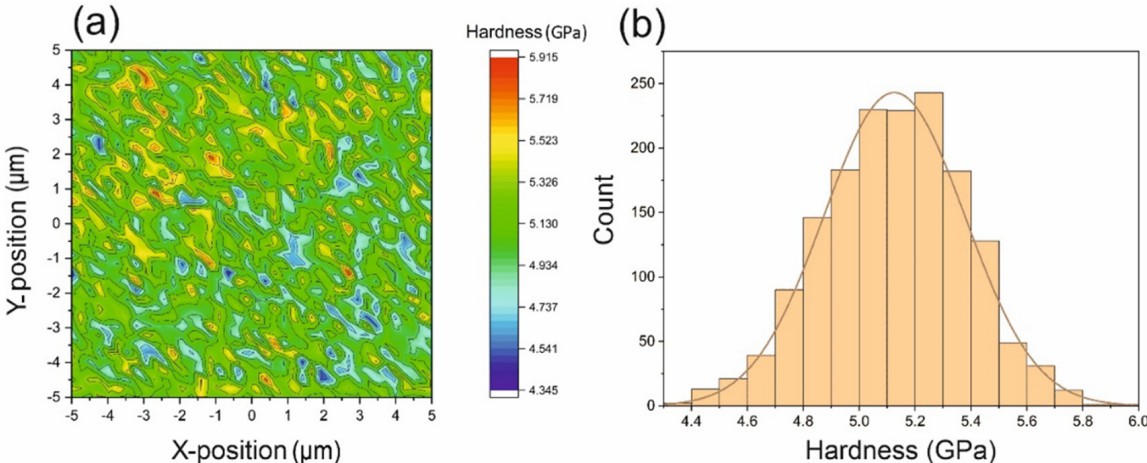

**Figure 24.** (**a**) Nanohardness map of 1600 indents, AMZ4 sample, and (**b**) histogram of the hardness results. Modified from [32], with permission from Elsevier, 2021.

### 4.5. Complex Shapes

One of the benefits of AM techniques over conventional fabrication methods is the ability to manufacture parts with complex geometry, thanks to their layer-wise fabrication. In Figures 14, 15, 17 and 18, BMG parts with complicated shapes were illustrated. Low surface quality is a limitation for AM products. Sohrabi et al. [32] used sandblasting to remove the attached powder to the fabricated part, and could improve the dimensional accuracy (as shown in Figure 25) and the surface roughness. Frey et al. [71] significantly reduced the surface roughness of a Zr-based BMG fabricated via LPBF, by TPF as a post-process treatment. As mentioned before, TPF can be used for those BMGs that have high GFA and can withstand high temperatures during the processing time without undergoing crystallization. However, TPF of parts with complex geometry, such as those in Figures 14 and 18, is almost impossible.

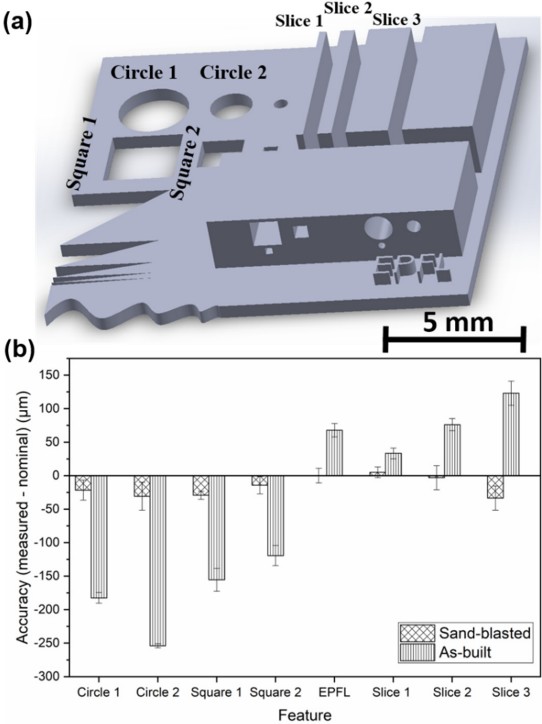

**Figure 25.** (**a**) CAD of a benchmark with complex features, whose printed version is shown in Figure 14b, and (**b**) accuracy of some of the features in the as-built and sandblasted conditions. Modified from [32], with permission from Elsevier, 2021.

Yang et al. [36] showed that one of the challenges in the AM fabrication of BMGs in complex geometries is that the optimized parameters determined from a simple geometry (cubic in Figure 26) are no longer valid, and may result in crystallization. Examples in Figure 26 include hollow shapes and lattices. This issue will be discussed further in the next section, crystallization of BMGs.

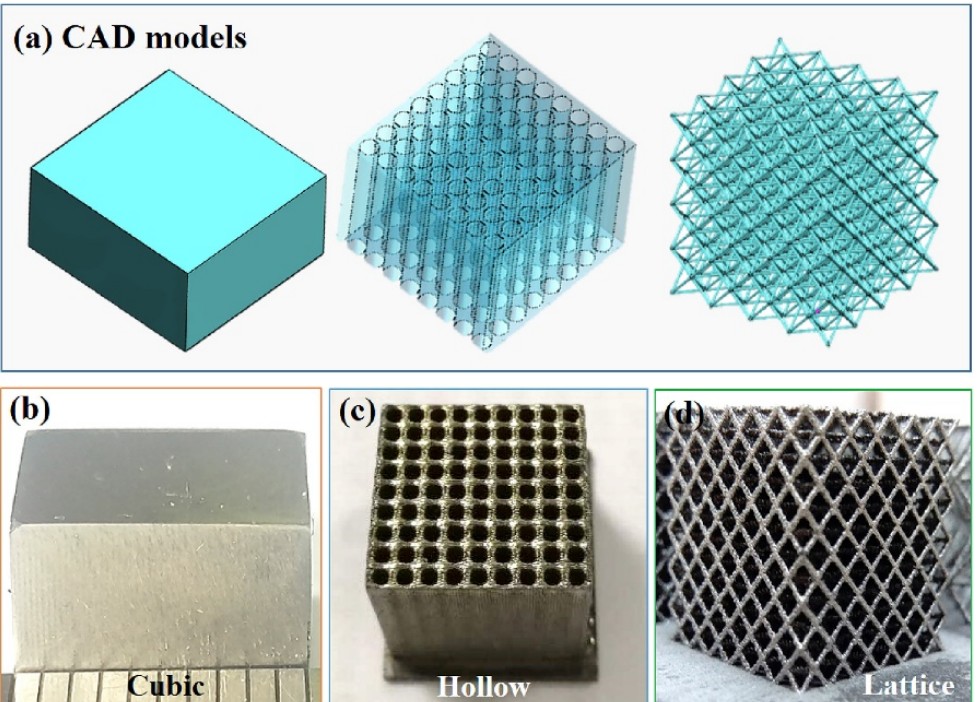

**Figure 26.** (**a**) CAD models and (**b–d**) fabricated parts of a Zr-based BMG via LPBF. Reproduced from [36], with permission from Elsevier, 2021.

### 4.6. Crystallization of BMGs

The most important and studied challenge of BMGs is the occurrence of crystallization and the consequences on properties. Although AM processes provide high heating and cooling rates, uncontrolled crystallization is still a major concern. Almost all studies blame reheating cycles for inducing crystallization in BMGs produced via AM techniques, but there is no unique scenario. Several approaches to crystallization were reported, which are categorized in the following sub-sections: change in chemical composition, structural relaxation, time spent at temperatures higher than $T_x$, shape effect, controlled crystallization and composite formation, global heating, GFA effect, and prediction of crystallization.

#### 4.6.1. Change in the Chemical Composition

Li et al. [107] measured crystallization in the first layer of a Zr-based MG fabricated via LFP, on a Ti-6Al-4V substrate. Due to the intermixing of the foil and the substrate, the chemical composition was locally changed (see Figure 27), shifting the TTT diagram of the MG to the left and therefore increasing the crystallization kinetics. This effect was reduced by changing the substrate to a crystalline Zr-alloy. The intermixing happens as a result of fluid flow in the melt pool.

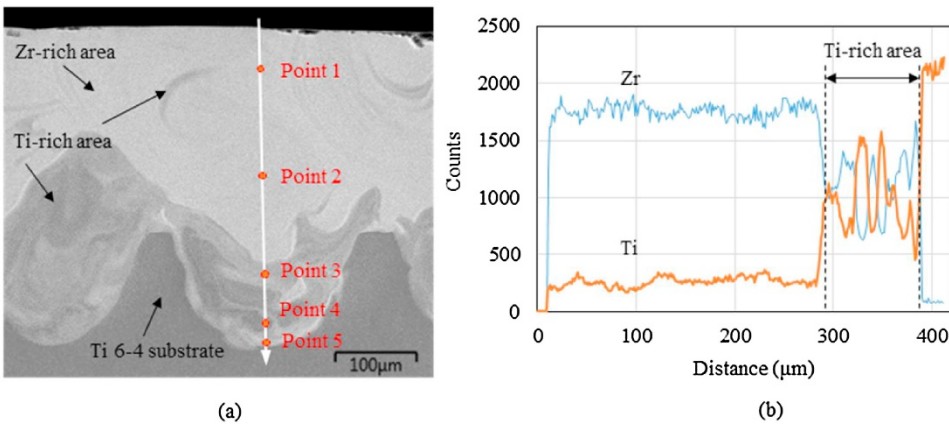

(a)    (b)

**Figure 27.** (**a**) BSE image from the first layer of LFPed Zr-based MG on a Ti-6Al-4V alloy and (**b**) EDS line analysis. Reproduced from [107], with permission from Elsevier, 2021.

Similarly, an intermixing zone was detected for a Pd-based BMG fabricated via LPBF in the near-interface region with the substrate [73]. Figure 28 shows SEM images of formed crystals due to the change of chemical composition; and the distribution of elements was also presented. To remove this region, the sample should be ground after cutting from the substrate, or a support structure should be designed to prevent the intermixing of the substrate with the bulk of the sample.

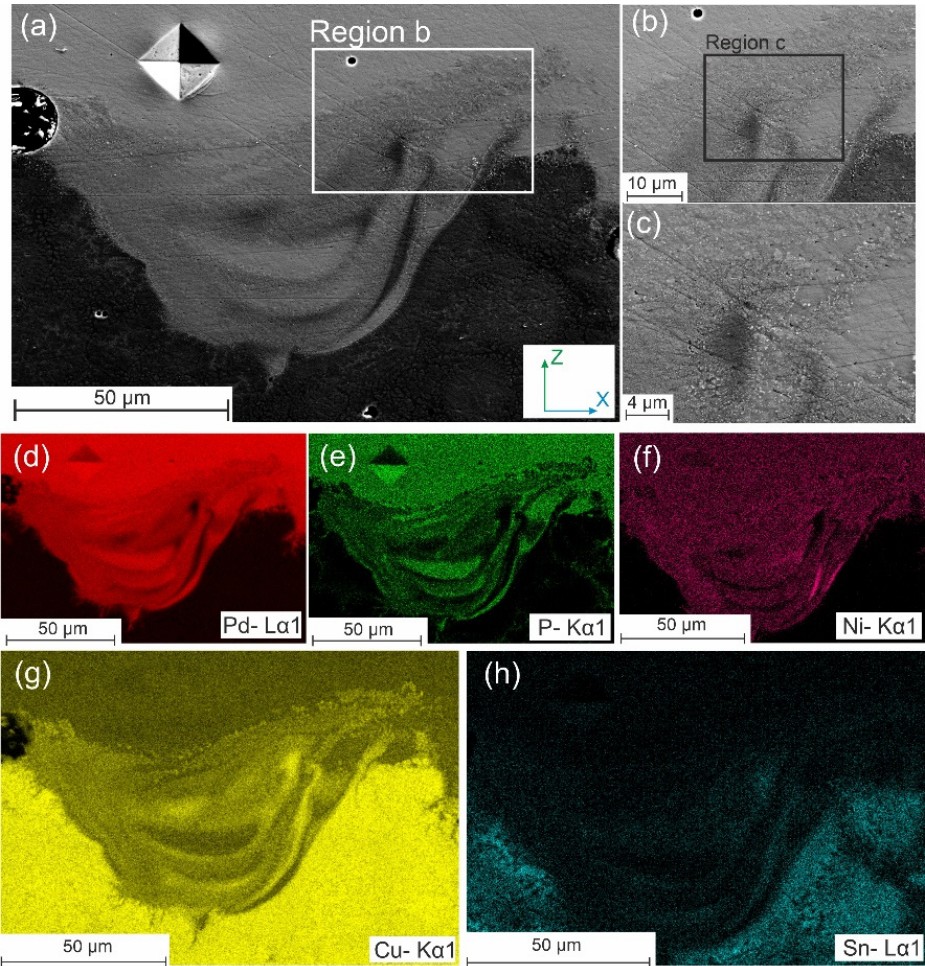

**Figure 28.** Pd-based BMG fabricated via LPBF on a Cu-Sn alloy substrate, (**a**–**c**) BSE images from regions close to the substrate and (**d**–**h**) EDS maps of elements. Reproduced from [73], with permission from Elsevier, 2021.

When MGs are used as a coating/cladding to provide wear and/or corrosion resistance, the substrate is significantly different from the MG. Sohrabi et al. [5] detected crystallization as a result of intermixing and chemical composition change. A high laser power (>60 W) resulted in cracking at the interface between the substrate and the coating, due to the combination of high residual stresses and increased fraction of crystallization. The processing parameters were optimized accordingly, such as to provide a good bonding with the substrate and the lowest possible crystallization in the near interface region.

Apart from the intermixing region close to the interface with the substrate, heterogeneity in the chemical composition in the bulk of the sample can also result in crystallization. Sohrabi et al. [73] reported impurities introduced during LPBF processing of a Pd-based BMG. At the vicinity of those impurities, a CuPd crystalline phase was detected. They mentioned this could happen as a result of a slight change in the local chemical composition of the matrix. According to Li et al. [85], a higher energy input (higher $P$ or lower $v$) affects the flow of the liquid in the melt pool and results in an inhomogeneous distribution of elements (see Figure 23). They optimized the processing parameters such as to reach a high density and low inhomogeneity, and further improved the homogeneity by using a remelting scanning strategy. However, Pauly et al. [33] pointed out that "the chemical heterogeneity observed in [85] might be a consequence of (partial) crystallization rather than its origin."

According to Shen et al. [109], the temperature in the melt pool is high enough to vaporize elements with lower vaporization temperature, such as Al, and cause crystallization due to the local change of chemical composition. However, they did not provide evidence for this hypothesis.

One of the elements present as an impurity in most BMGs is oxygen. Oxygen encourages the crystallization of BMGs by acting as a preferential site for nucleation of metastable quasicrystals, and reduces the GFA [37,152,153]. Industrial-grade BMGs, such as AMZ4, have a high oxygen content. Therefore, the processing window to fabricate parts with high amorphous content is narrower for higher oxygen content. Using fast differential scanning calorimetry (FDSC), the TTT diagram of AMZ4 with a high oxygen content (1200–1400 ppm) was measured. The time to crystallization was on the order of 3 ms in the 750–800 °C temperature range, which effectively prevents the fabrication of fully amorphous AM parts [154]. Several studies on the LPBF of AMZ4 [32,55,56] attributed the present nanocrystals to $Cu_2Zr_4O$ phase.

Wegner et al. [69] showed the effect of the protective gas on the crystallization of AMZ4 during LPBF process. They used $N_2$, Ar, and $Ar_{98}H_2$ as protective gas and reported that $N_2$ resulted in a higher number of cracks and increased crystallized fractions compared to $Ar_{98}H_2$ and Ar. Using a reducing atmosphere, $Ar_{98}H_2$, helped reducing the oxygen content and lowered the crystalline fraction and number of cracks compared to the $N_2$ neutral atmosphere.

### 4.6.2. Structural Relaxation

Structural relaxation causes rearrangement of atoms and annihilation of free volume in BMGs [155]. Even sub-$T_g$ heat treatments can result in structural relaxation [156]. Due to the cyclic heating in AM of BMGs, structural relaxation is inevitable. Its effect on the ductility of the fabricated parts will be discussed in Section 5.5, Yang et al. [91] were the first ones to report crystallization in a Zr-based BMG fabricated via LSF (using a pulsed laser) as a result of the accumulation of structural relaxation in the heat-affected zone (HAZ). No crystallization was detected up to irradiation of 12 pulses (Figure 29). After that, spare clusters of spherulites were formed in the HAZ, which confirmed the accumulation of structural relaxation.

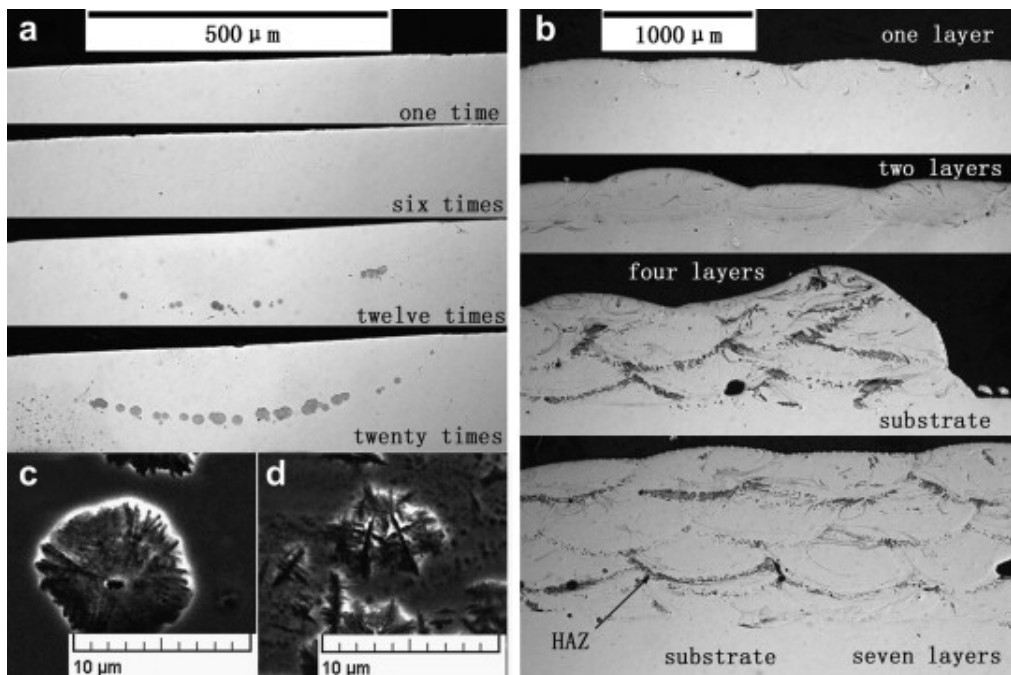

**Figure 29.** (**a**) Cross-sections of a Zr-based BMG as a result of one, six, twelve, and twenty laser pulses, (**b**) cross-section of one, two, four, and seven deposited layers using LSF, (**c**) spherulite crystal in the HAZ of the sample with twenty pulses, and (**d**) crystals in the HAZ of the sample with seven deposited layers. Reproduced from [91], with permission from Elsevier, 2021.

Lu et al. [98] used FEM simulations to extract the thermal history of point A (see inset of Figure 30) during the DED process of a Zr-based BMG. Cycles that led to temperatures higher than $T_g$ were considered as effective thermal cycles. Figure 30 shows several effective thermal cycles, which resulted in the accumulation of structural relaxation and consequently crystallization.

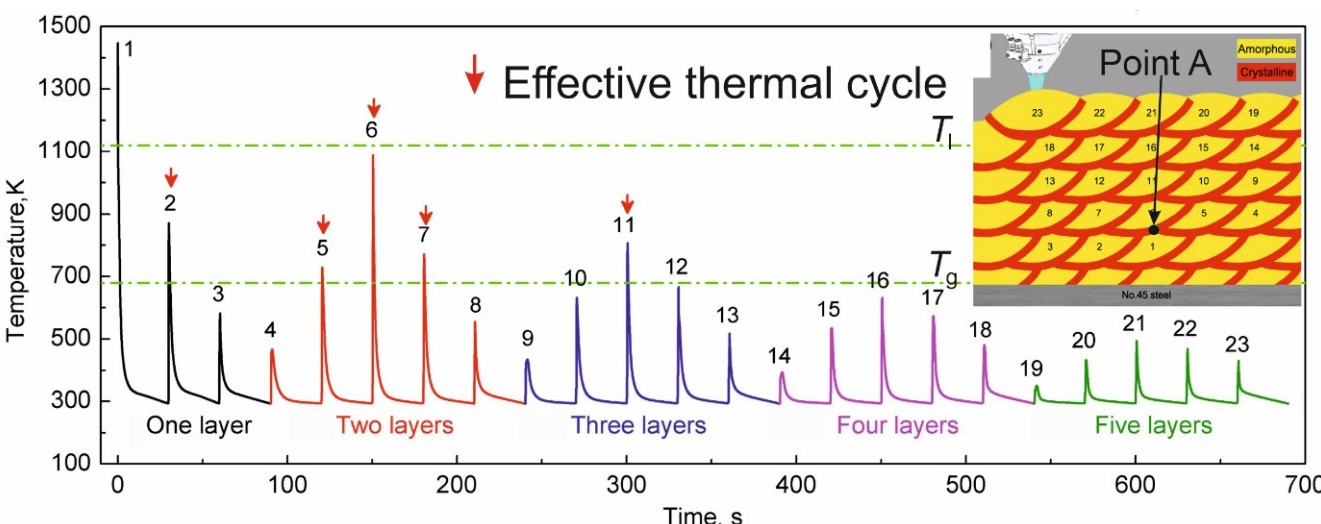

**Figure 30.** Thermal cycles extracted from FEM simulations of the DED process on a Zr-based BMG. The inset shows the schematic of the process and the location at which these thermal cycles were computed. Modified from [98], with permission from Elsevier, 2021.

Figure 31a shows a bright-field transmission electron microscopy (TEM) image of a melt pool boundary of a Zr-based BMG fabricated via LPBF. As can be seen in Figure 31b, amorphous structures show different diffracted intensities in the HAZ and in the melt pool. This could be correlated to (i) structural relaxation that occurred in the HAZ and (ii) local change of chemical composition related to the presence of nanocrystals in the HAZ.

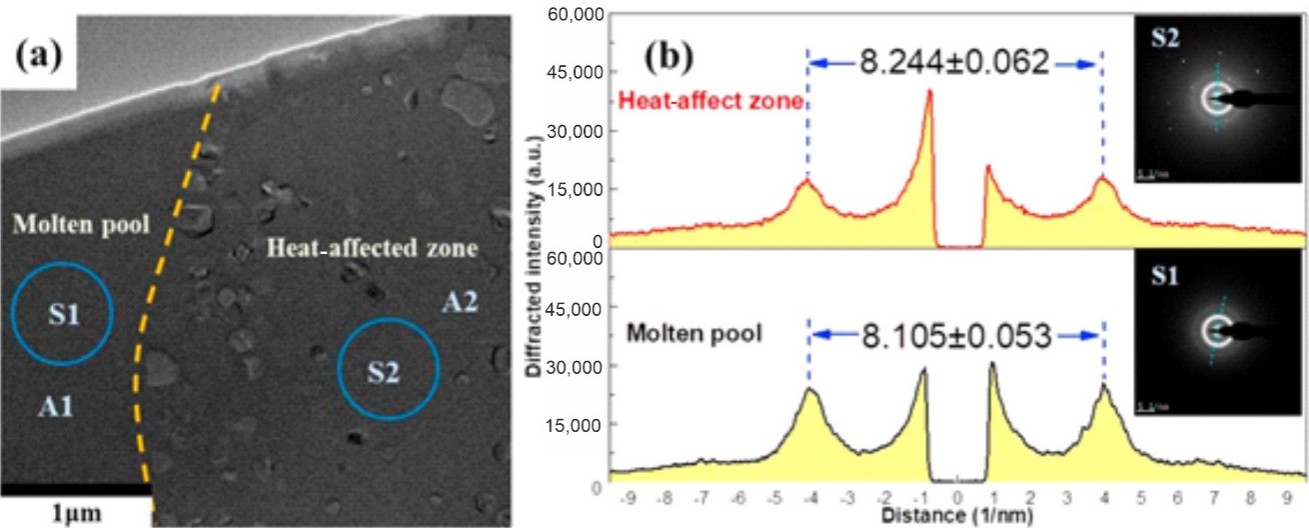

**Figure 31.** (**a**) Bright-field TEM image of the melt pool boundary of a Zr-based BMG fabricated via LPBF and (**b**) diffraction intensity distribution based on the SAED of S1 and S2 in (**a**). Reproduced from [40], with permission from Elsevier, 2021.

Zhang et al. [74] attributed the crystallization of $Zr_{50}Cu_{50}$ BMG during the LPBF process to the structural relaxation in the HAZ. In this case, crystals were beneficial and improved the ductility of the fabricated parts. More information concerning their effects on the mechanical properties will be presented in Section 5. Apart from Zr-based BMGs, structural relaxation as a reason for crystallization in the HAZ was also reported for a Fe-based BMG fabricated via LPBF [68]. In addition to the crystallization, micro and macrocracks were detected in the microstructure (Figure 32). Decreasing *P* and increasing *v*, i.e., decreasing the VED, increased the amorphous fraction. In Figure 32d,e, crystals were observed inside the melt pool, but no explanation was provided. This could be attributed to the cooling rate of the melt pool not being higher than the critical cooling rate, or to an increase of the average temperature as a result of using a high VED. Since the powder feedstock was not fully amorphous, some of the nanocrystals might not have been completely melted in the melt pool, which then increases the critical cooling rate of the alloy. Decreasing the VED increases the cooling rate in the melt pool, decreases the chemical and structural inhomogeneity [85], as well as the average temperature of the sample, which could all be reasons for the reduced amount of crystals in the melt pool of the sample shown in Figure 32c,f,i.

Lin et al. [95] used the notion of annealing effect and Liu et al. [99] used the notions of tempering, instead of structural relaxation, to explain the crystallization mechanism in the HAZ of BMGs during AM processing. These notions are basically the same.

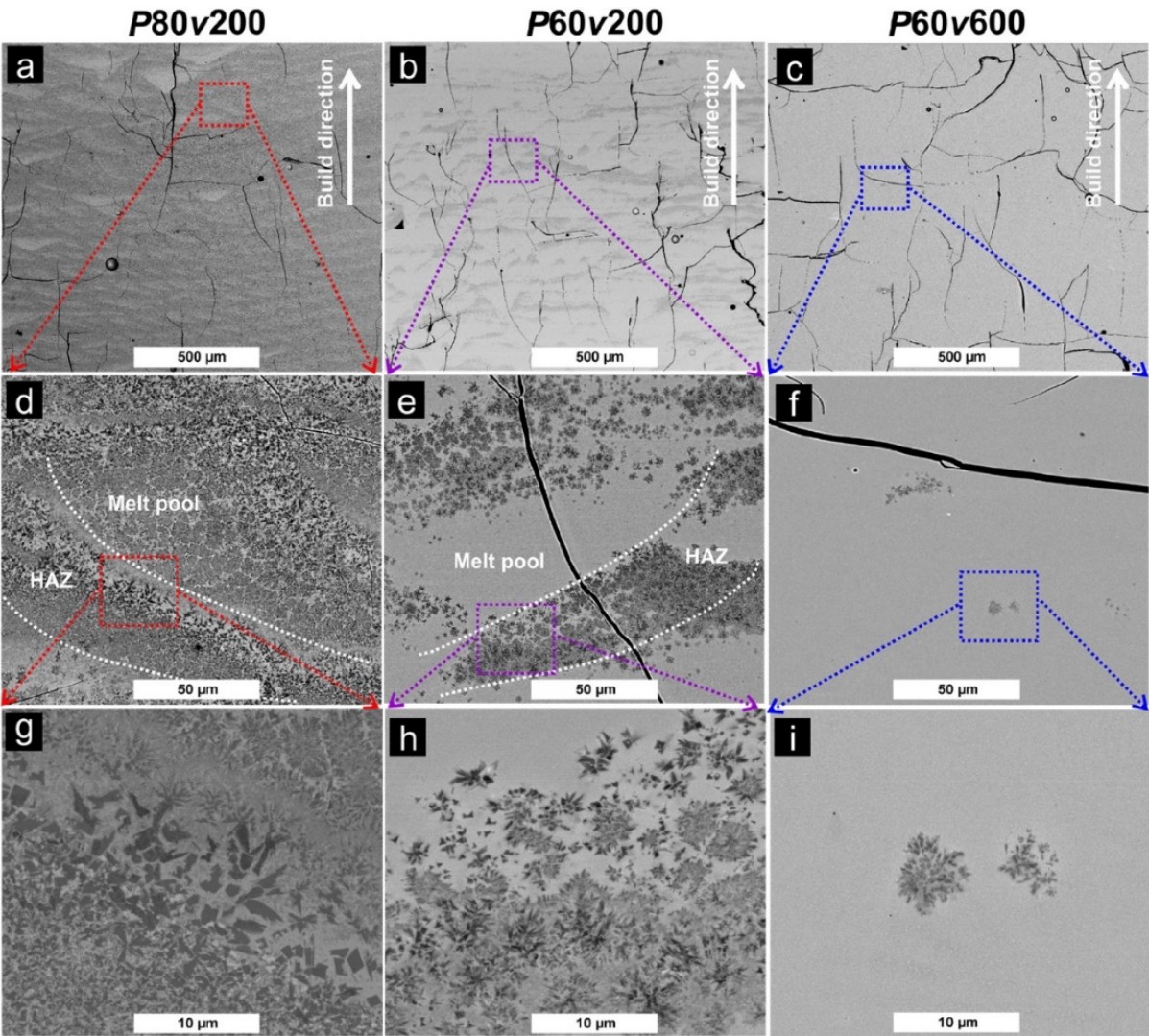

**Figure 32.** Microstructure of a Fe-based BMG fabricated via LPBF using three sets of laser parameters (*P* is the power (W), *v* is the scanning speed (mm/s)). (**a**) Microstructure of the sample fabricated with the power of 80 W and scanning speed of 200 mm/s, (**b**) microstructure of the sample fabricated with the power of 60 W and scanning speed of 200 mm/s, (**c**) microstructure of the sample fabricated with the power of 60 W and scanning speed of 600 mm/s, (**d**) higher magnification of (**a**), (**e**) higher magnification of (**b**), (**f**) higher magnification of (**c**), (**g**) higher magnification of (**d**), (**h**) higher magnification of (**e**), and (**i**) higher magnification of (**f**). Reproduced from [68], with permission from Elsevier, 2021.

### 4.6.3. Time Spent at Temperatures Higher Than $T_x$

Since the heating and cooling rates of the LPBF process are in most of the cases higher than the critical cooling and heating rates of the BMGs, thermal FEM simulations were used to unveil the crystallization mechanisms. Thermal FEM simulations of a Zr-based BMG produced via LPBF are presented in Figure 33 [86]. Thermal histories of three points in Figure 33b are shown in Figure 33c. The point $S_b$ is located in the HAZ and, according to Ouyang et al. [86], when the maximum temperature exceeds $T_x$, it is difficult to prevent crystallization. They used the heating rate of 20 K/s to measure $T_x$ with a conventional DSC system. However, higher heating rates shift $T_x$ to higher temperatures [154]. Sohrabi et al. [32] used FDSC to measure $T_x$ of AMZ4 powder at the heating rate of $10^4$ K/s and used it as a criterion to predict the region more sensitive to crystallization. Based on the thermal history exploited from FEM simulations of the HAZ, crystallization was expected in a region up to 3 μm below the melt pool, which matched well with the experimental results.

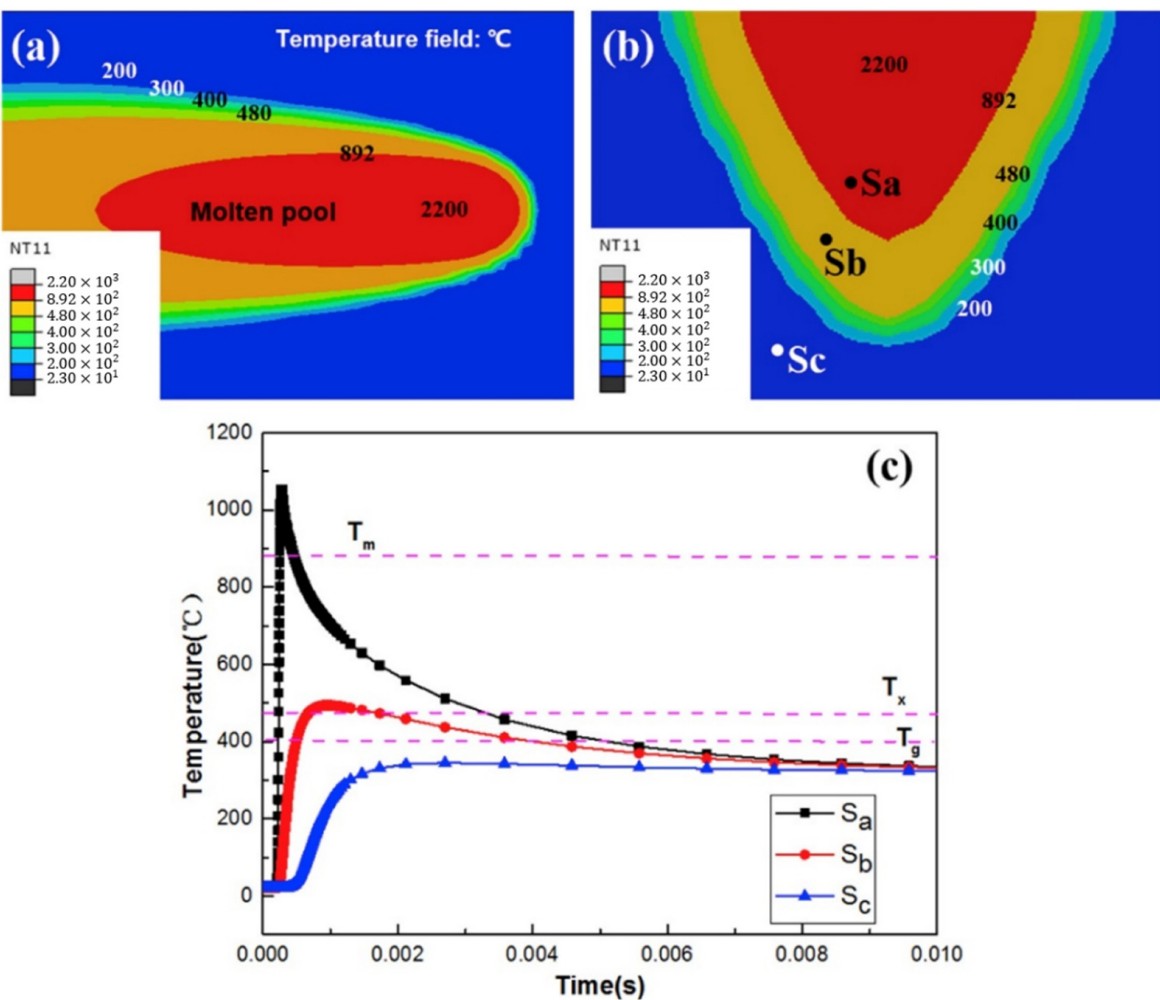

**Figure 33.** Thermal FEM simulation results of LPBF of a Zr-based BMG (**a**) top-view of the melt pool, (**b**) cross-section of the melt pool, and (**c**) thermal history of three points in (**b**). Reproduced from [86], with permission from Elsevier, 2021.

In another study, Ouyang et al. [35] and Wang et al. [49] mentioned that only exceeding $T_x$ in the HAZ is not enough for crystallization, i.e., the time spent at temperatures higher than $T_x$ is also important. If this time is longer than the incubation time for nucleation, and allows for non-negligible growth, crystallization will be measurable. They also used FEM thermal simulations to explain the crystallization mechanism. Figure 34 shows the thermal history of different points in the LPBF simulations. If there is an intersection between the curve of the incubation time and the thermal history, crystallization will happen [35]. Nevertheless, it should be noted that the results of the thermal history did not start from time zero (see time-axis in Figure 34). The graphs show ambient temperature for almost 2 ms in the beginning, which should have been removed and this might have affected the results. Shen et al. [109] used three different sets of parameters for LFP of a Zr-based BMG. They showed that in the HAZ, the time spent above $T_g$ was reduced when shifting from continuous to pulsed laser, which led to a reduced crystallized fraction. The time above $T_g$ is either called "annealing time" or "relaxation time", depending on the studies.

Vora et al. [94] conclude that it is in the regions of the HAZ where the temperature is higher than $T_x$, especially in the temperature range close to the nose of TTT diagram (with lowest incubation times) that crystallization is most likely to occur.

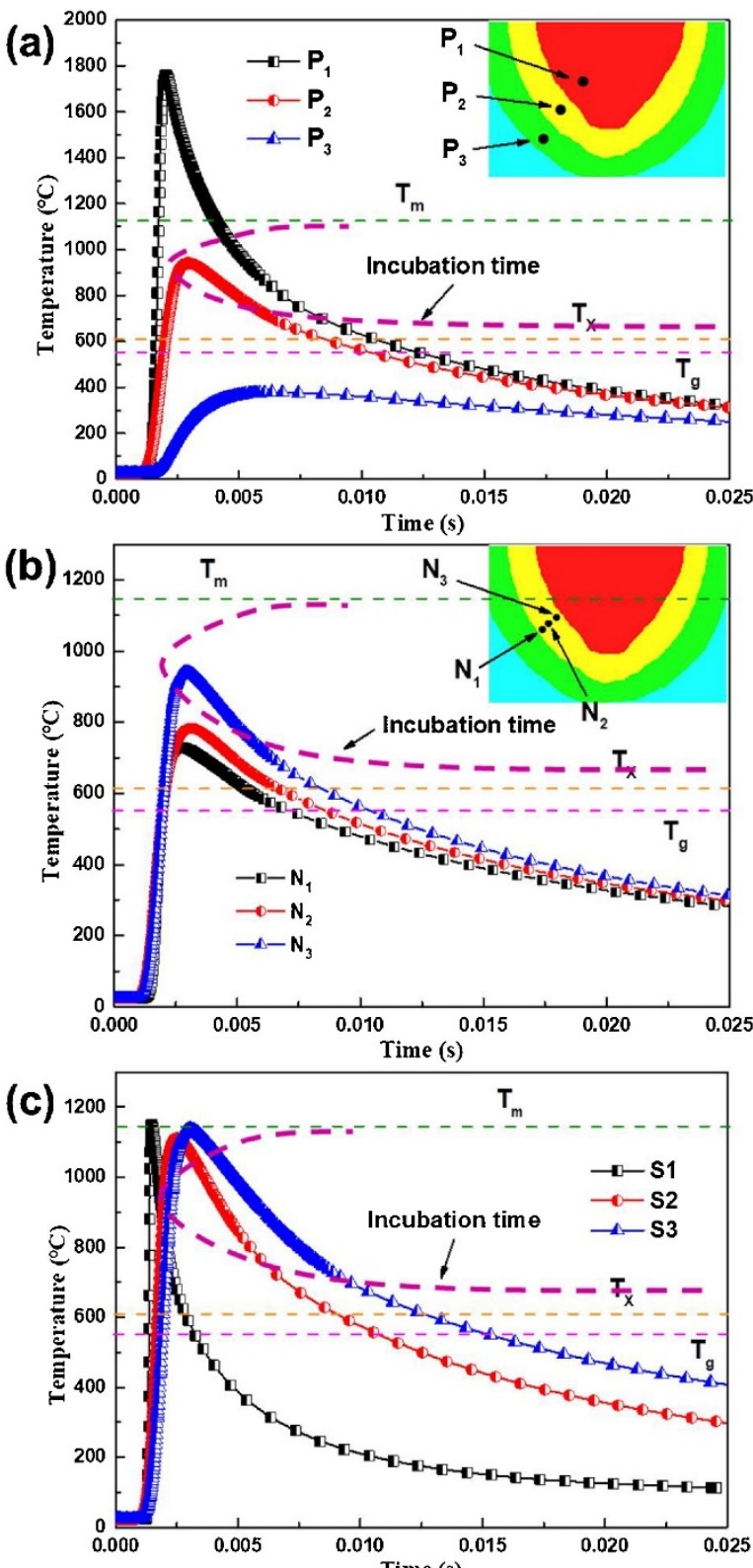

**Figure 34.** FEM thermal simulation of a Zr-based BMG fabricated via LPBF: (**a**) thermal histories of three points in the melt pool, in the HAZ, and below the HAZ; (**b**) thermal histories of three points within the HAZ; and (**c**) thermal histories of the boundary of the melt pool for three samples with different VEDs, S3 > S2 > S1. Reproduced from [35], with permission from Elsevier, 2021.

### 4.6.4. Shape Effect

As mentioned before, one of the advantages of AM techniques over the conventional fabrication methods of BMGs is the ability to produce parts with complex geometries. Yang et al. [36] produced Zr-based BMG samples by LPBF with different geometries, and investigated the microstructure. XRD patterns and DSC curves in Figure 35 show that the shape affects the amount of crystallization because it affects the melt pool size and heat dissipation. Although FEM simulations indicated heating and cooling rates much higher than the critical cooling rate of the alloy, crystallization was detected.

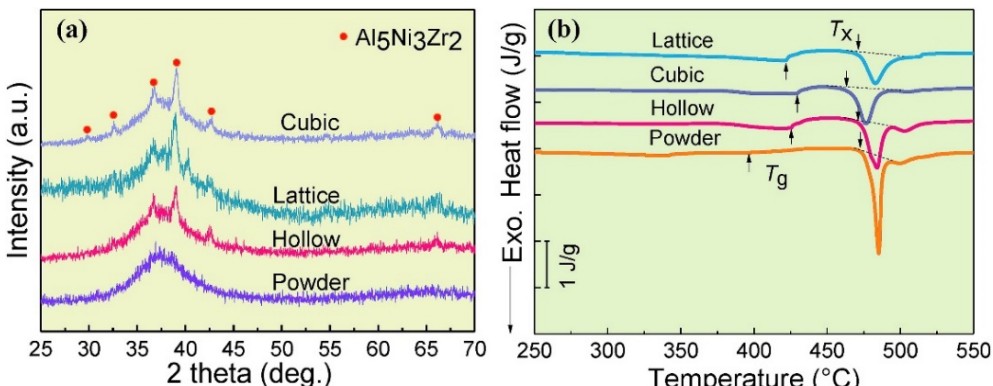

**Figure 35.** (**a**) XRD patterns of samples with different geometries and (**b**) DSC curves of the same samples. Reproduced from [36], with permission from Elsevier, 2021.

Sohrabi et al. [80] fabricated cubes via LPBF with a size of $5 \times 5 \times 4$ mm$^3$, some of them including a cylindrical hole open to the side surface, 300 μm below the top surface, in which a thermocouple was inserted (see Section 4.6.6). A layer of AMZ4 powder was printed 300 μm above the hole, and the thermocouple recorded a temperature higher than the melting point of the BMG (>915 °C). This infers that the melt pool depth was larger than 300 μm. However, for another sample without hole, the melt pool depth was measured as ~160 μm, and FEM simulations demonstrated temperature fields leading to a lower amount of crystallization. This significant difference is related to the low thermal conductivity of BMGs. It can be concluded that AM fabrication of BMGs with complex shapes is challenging and requires adapting of the processing parameters according to each particular geometry.

### 4.6.5. Controlled Crystallization and Composite Formation

As mentioned previously, the crystallization of BMGs, in most cases, is detrimental and unwanted. However, to improve ductility and fracture toughness of BMGs, ductile crystalline phases can be introduced to form BMG composites (BMGCs) [46,64,102]. Lu et al. [102] used DED to fabricate in situ BMGCs using different parameters (changing $P$ and/or $v$) for each layer of $Zr_{39.6}Ti_{33.9}Nb_{7.6}Cu_{6.4}Be_{12.5}$ (DH3). This procedure led to a different crystallized fraction in each layer, and resulted in producing a functionally graded material (FGM) (see Figure 36). FEM simulations allowed assessing the cooling rates associated with each set of processing parameters and relating them to the measured volume fraction of crystallization. The results are presented in Figure 37f. It can be noted that as $P/v$ increased, the cooling rate reduced, and consequently, the volume fraction of crystallization increased. It is noted that the critical cooling rate of DH3 is very high. Even in casting, where no reheating cycles occur, the alloy forms dendritic crystals [157]. Mechanical properties (hardness and tensile strength) of the fabricated samples were also investigated, they will be discussed in Section 5.

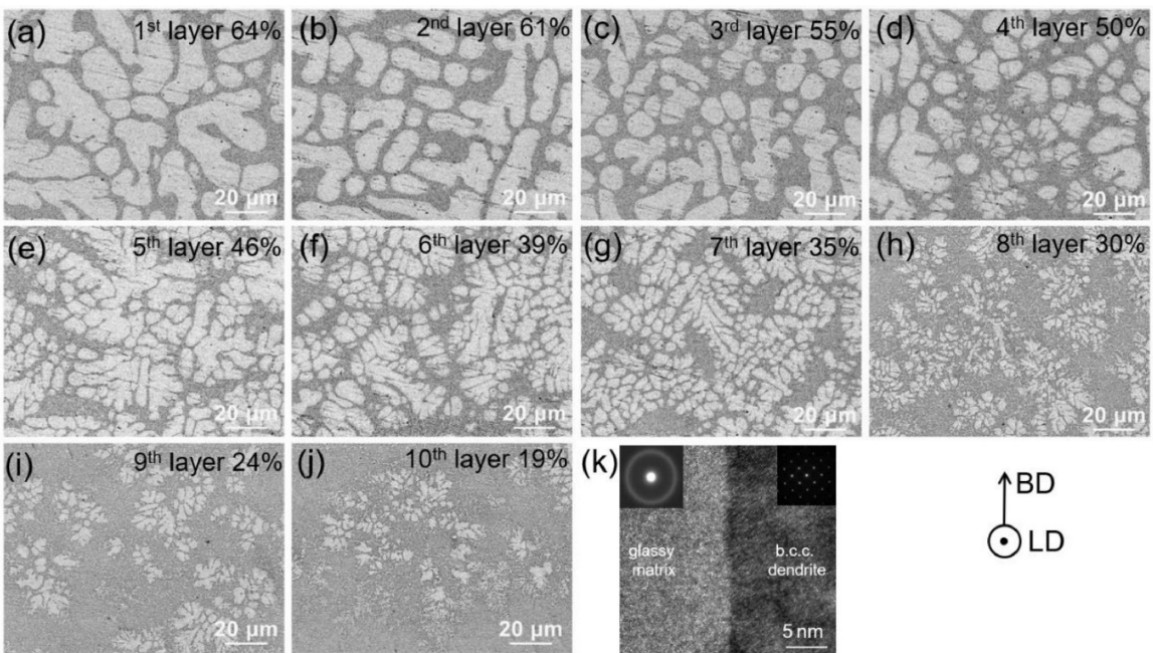

**Figure 36.** (**a**–**j**) BSE images of the microstructure of each layer of DH3 (BMGC) fabricated via DED, the 1st layer was the top layer; (**k**) high-resolution TEM characterization of the interface between amorphous and crystalline phases. Reproduced from [102], with permission from Elsevier, 2021.

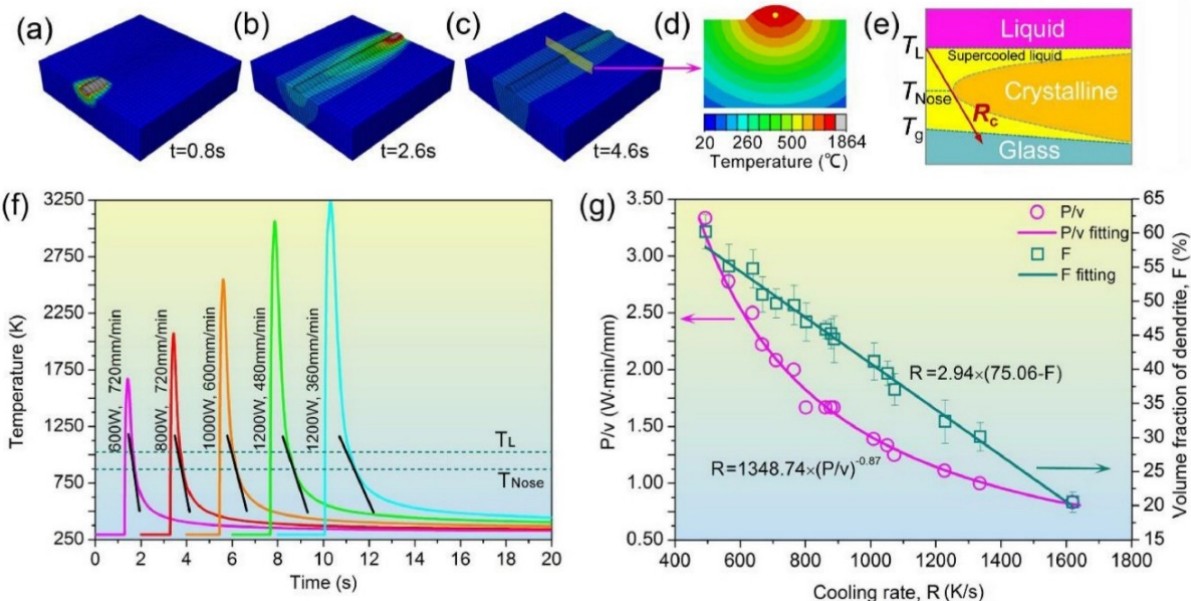

**Figure 37.** (**a**–**c**) FEM simulation of a single laser track, (**d**) illustration of the temperature field of the cross-section of the simulated track, (**e**) schematic of a TTT diagram, (**f**) thermal histories of tracks with different parameters used in the fabrication of the BMGC, and (**g**) relationship between the computed cooling rate, processing parameters, and the measured volume fraction of crystallization. Reproduced from [102], with permission from Elsevier, 2021.

Zhang et al. [46] used Ta in a pre-alloyed $Zr_{57.4}Ni_{8.2}Cu_{16.4}Ta_8Al_{10}$ BMGC to improve ductility in compression and fracture toughness. The Ta particles were clearly detectable in the microstructure. Unlike DH3 [102], it did not result in a dendritic structure. In Section 5.6, it will be shown how Ta particles arrest and/or deflect shear bands and improve mechanical properties.

Zou et al. [64] mixed Cu with a Fe-based MG powder using ball milling, and processed it by LPBF. The Cu/FeCrMoCB composite with 40 vol% and 50 vol% Cu content resulted in more than 10% plastic deformation in compression.

Liu et al. [99] used pure premixed powders of Zr, Cu, Al, and Ni to fabricate BMGCs using DED. Some of the larger powder particles that were not completely melted could act as pre-existing nuclei and form crystals in the melt pool. Besides, the presence of crystals in the HAZ was evident due to the "tempering". The hardness of the HAZ was lower than that of the melt pool. This signifies that the crystals had softer nature than the amorphous matrix and could improve the ductility of the fabricated parts.

Bordeenithikasem et al. [100] created a BMGC based on FeCrMoBC using DED. They calculated the theoretical cooling rate of the process based on the Rosenthal equation [158], which was lower than the critical cooling rate of the alloy, but could still partially vitrify the structure. The increase of *P* resulted in different cooling rates for different samples and the microstructure was highly dependent to the cooling rate. The decrease of the cooling rate made the microstructure harder because of the formation of carbides, borides, and finally, a crystalline eutectic matrix. Figure 38a shows the microstructure of the sample fabricated with a higher cooling rate. Several shear bands are detectable around hardness indents. Arresting shear bands by the crystalline dendritic phase is shown in Figure 38b. However, the sample with a lower cooling rate (see Figure 38c,d) formed a crack at the corner of the hardness indent because of the more brittle structure.

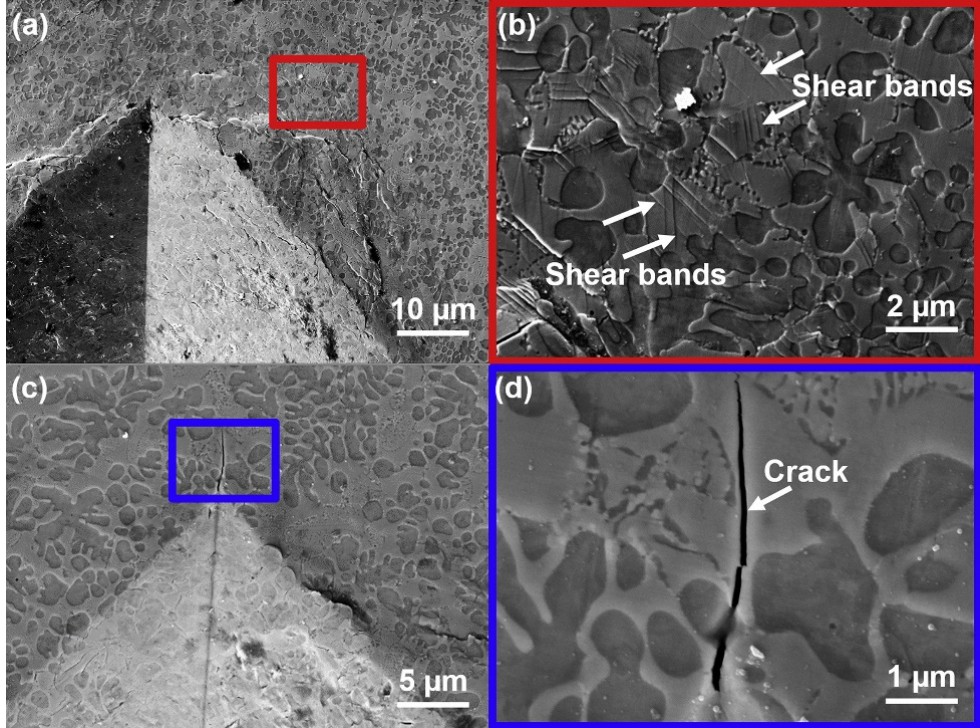

**Figure 38.** FeCrMoBC BMGC by DED, (**a**,**b**) higher cooling rate and (**c**,**d**) lower cooling rate. Reproduced from [100], with permission from Elsevier, 2021.

Since crystallization of BMGs fabricated via AM techniques is almost inevitable, designing BMGs in which crystals improve the mechanical properties, such as $Zr_{50}Cu_{50}$ [74], is of high importance.

### 4.6.6. Global Heating

Sohrabi et al. [80] reported that apart from the *local* thermal effects, which leads to reheating of the laser track neighboring region, i.e., the HAZ, a *global* one also exists in LPBF. The *global* effect results in heat accumulation, an increase of the average temperature of the sample, and reduces the cooling rate, in agreement with the thermal simulations of Lindwall et al. [45]. Sohrabi et al. [80] isolated the *global* effect by using different waiting times after each laser track. To record the temperature, a cube with a hole was fabricated, and thermocouples were inserted (Figure 39a). Then, a layer of AMZ4 (40 μm) was deposited on the cube, and processed by LPBF. The thermal histories of samples with different waiting times are presented in Figure 39b–d. For the sample with 40 ms waiting time, the temperature after each laser scan did not drop and the baseline temperature reached high values. As the waiting time increased, the baseline temperature dropped. For the sample with 1 s waiting time, the temperature drops below 100 °C after each laser track. The crystallized fraction increased as the waiting time decreased. It should be noted that the *global* effect is very dependent to the processing parameters. The higher the VED, the more pronounced the *global* effect. The comparison of the experimental measurements and simulated results are given in Figure 39b–d. Differences in heating and cooling rates between experimental and theoretical results may partially occur due to uncertainty in the position of the thermocouple, and unknown contact conditions of the thermocouple to AMZ4. However, despite these uncertainties, calculated results show reasonable agreement with the measurements.

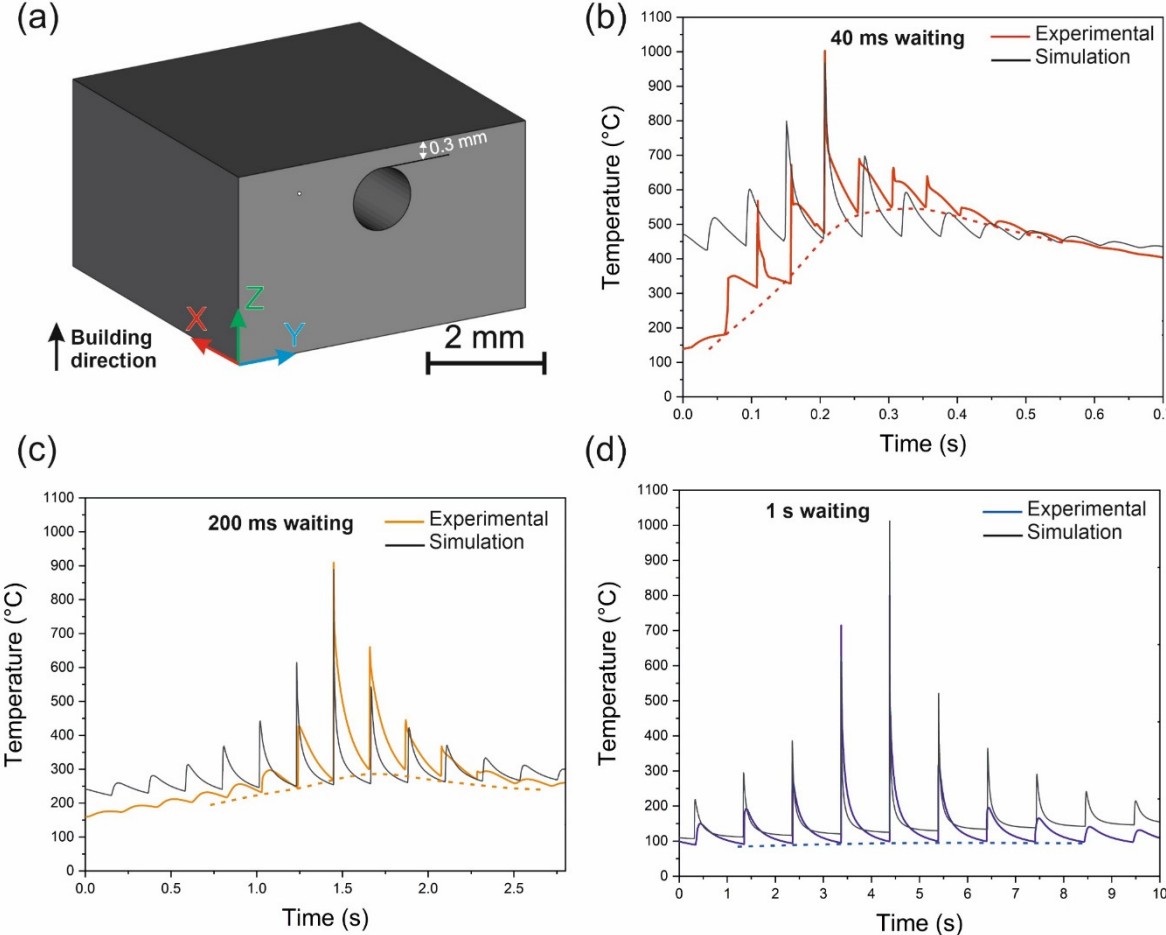

**Figure 39.** (**a**) Schematic of a cube with a size of $5 \times 5 \times 4$ mm$^3$ containing a hole with a diameter of 1 mm and a depth of 2 mm, used to insert a thermocouple. The hole was 300 μm below the top surface. (**b**–**d**) Experimental and simulated temperature evolutions related to the addition of the first layer after inserting the thermocouple, with (**b**) 40 ms waiting time, (**c**) 200 ms waiting time, and (**d**) 1 s waiting time. [80].

Zhang et al. [92] also referred to the *global* effect during LSF of a Zr-based BMG: "to maintain a high-volume fraction of the amorphous content in the deposit, it is important to note that the previously deposited layer should be cooled to a suitable temperature just prior to the subsequent deposition during LSF". However, they did not mention that the time between adjacent laser tracks also matters. Vora et al. [94] showed in Figure 40a that the baseline temperature increased after each laser track, but they did not discuss the *global* effect as one of the reasons for crystallization.

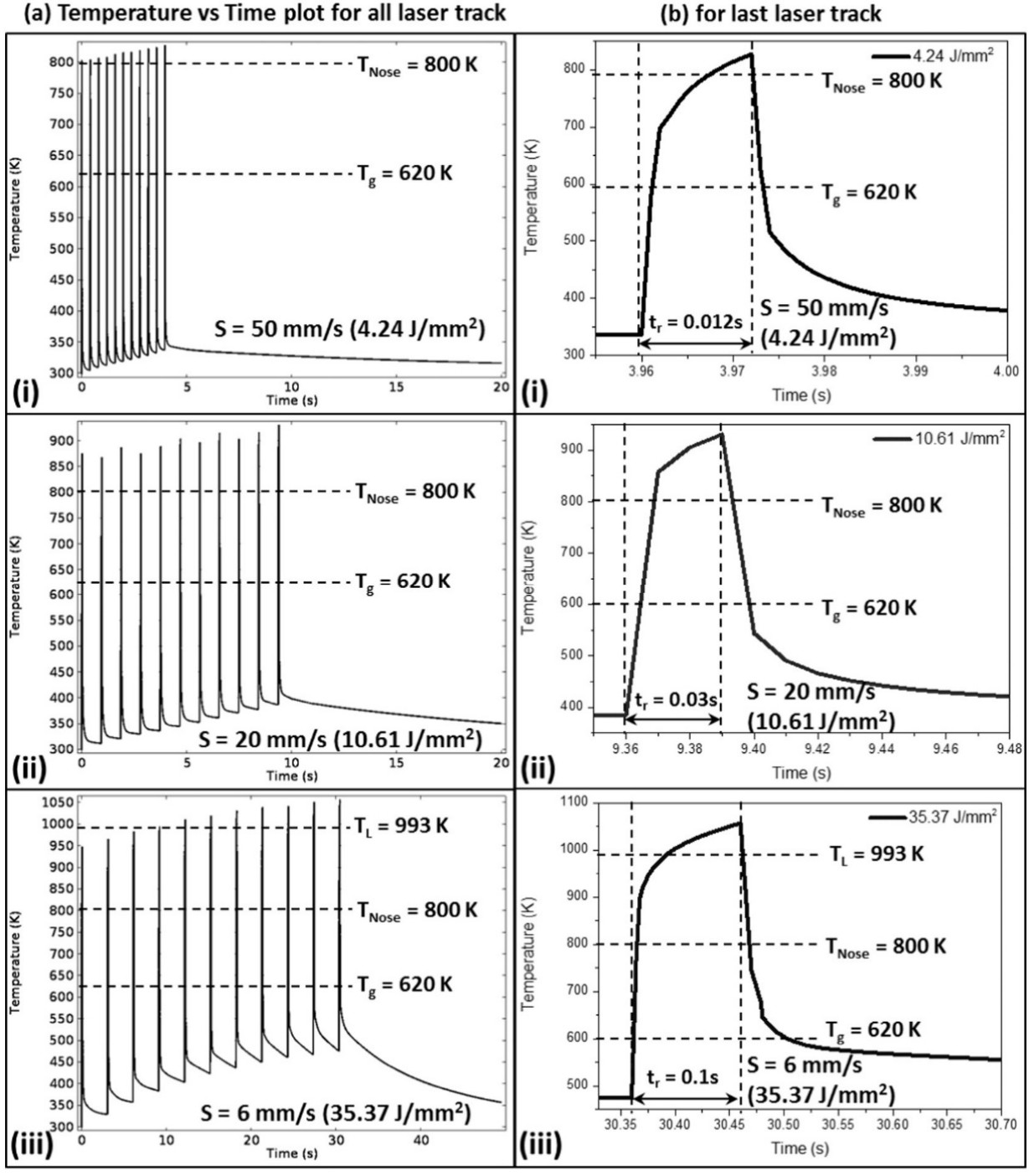

**Figure 40.** Simulated thermal histories with three scanning speeds (**a**) for all 11 laser tracks and (**b**) the last track. Reproduced from [94], with permission from Elsevier, 2021.

### 4.6.7. GFA Effect

Ouyang et al. [72] investigated the crystallization of two Zr-based BMGs (ZrAg and Zr55) with different GFAs fabricated via LPBF. The alloy with a lower GFA, ZrAg, had a higher amorphous content at different VEDs. Isothermal annealing at different temperatures was performed to study the crystallization mechanisms of the two alloys (see Figure 41a). The time to the onset of crystallization was called the incubation time, and the time between onset and endset of crystallization was called the crystallization time. Although the incubation time for ZrAg was lower, it had a longer crystallization time (see Figure 41b).

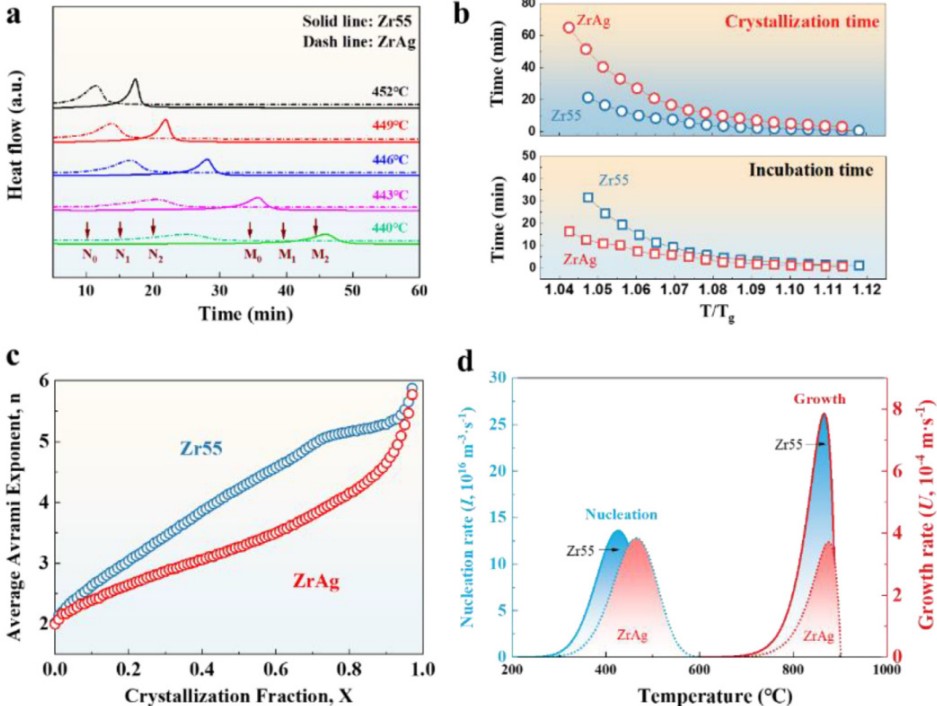

**Figure 41.** (**a**) Isothermal annealing of ZrAg and Zr55 MG ribbons at different temperatures, (**b**) comparison of the crystallization time and incubation time of the two MGs, (**c**) average Avrami exponent for the two MGs at different crystallization fractions, and (**d**) calculated nucleation rates (*I*) and growth rates (*U*) of the two MGs. Reproduced from [72], with permission from Elsevier, 2021.

Ouyang et al. [72] showed that the crystal growth rate was the controlling factor, i.e., ZrAg with the lower growth rate led to a higher amorphous content (see Figure 41d). The difference of growth rates between the two BMGs relates to the number of distinct crystalline phases (several for ZrAg, and only one for Zr55).

### 4.6.8. Prediction of Crystallization

Lindwall et al. [45] used thermal simulations to predict crystallization. To decrease the simulation time, they deposited the energy layer by layer at once. They used a double exposure strategy for each layer and mentioned that the temperature for the second exposure further increased due to the accumulation of heat. The crystalline fraction was the highest five layers below the top surface, and the largest increase occurred after the second exposure. Moreover, they did not provide any experimental results to check whether they match the prediction. In addition, the notion of laser tracks being absent, the reheating from adjacent tracks was neglected.

Lu et al. [96] tried to predict crystallization of a Zr-based BMG ($Zr_{50}Ti_5Cu_{27}Ni_{10}Al_8$) in DED using FEM simulations and an estimated TTT diagram. They estimated the critical cooling and heating rates of the alloy using conventional DSC tests, with low cooling and

heating rates (1 K/s), far from those experienced in DED, which is an issue as the heating and cooling rates do affect the crystallization mechanism [32].

Ericsson et al. [66] used FDSC upon cooling (cooled down from the melt to the desired temperature at which isothermal treatment was performed) to measure TTT diagrams, and the experimental results fitted well with the Mondal–Kumar–Gupta–Murty (MKGM) model [159], used for the nucleation theory and the prediction of crystallization. Although a fast-cooling rate ($2 \times 10^4$ K/s) was used to measure the time to crystallization, in LPBF the phenomenon actually happens upon reheating. Therefore, the TTT diagram upon heating (heated up from the room temperature to the desired temperature at which isothermal treatment was performed) should be used instead. It has been shown that the TTT diagrams upon heating and cooling differ significantly (see Figure 42) [154]. Continuous heating transformation (CHT) and continuous cooling transformation (CCT) curves were measured based on the time to the onset temperature of the crystallization upon heating and cooling, respectively. The TTT diagrams and the CHT and CCT curves are useful in understanding the isothermal and isochronal crystallization behaviors of BMGs. During casting and TPF where there are not several thermal cycles, CHT and CCT could be used. However, in AM processes multiple heating and cooling occur during production with different rates. To predict the crystallization, the Johnson–Mehl–Avrami–Kolmogorov's (JMAK) kinetic model [66,160] is required, which only applies for isothermal kinetics in solid-state transformation or chemical reactions [161]. Extending the JMAK model to anisothermal conditions is done using the additivity principle [80,162].

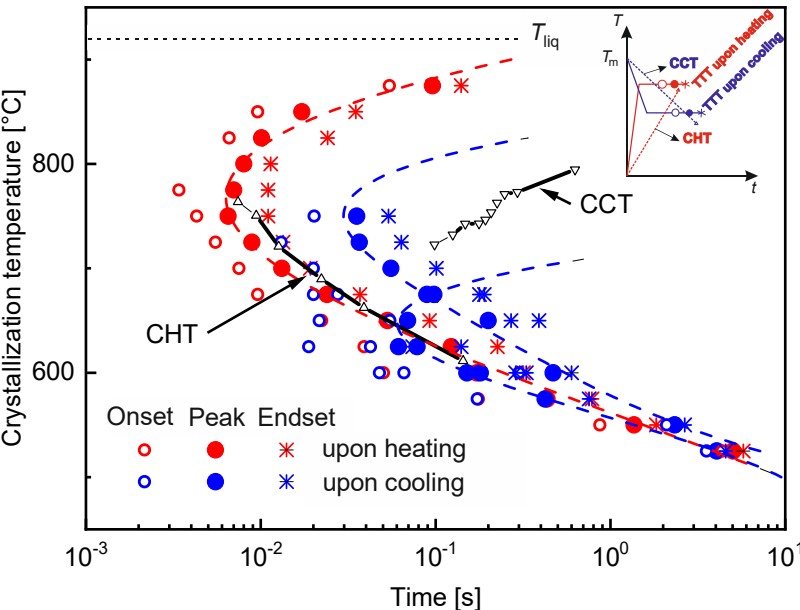

**Figure 42.** TTT diagrams measured upon heating and cooling in addition to continuous heating transformation (CHT) and continuous cooling transformation (CCT) for AMZ4 powder. Reproduced from [154], with permission from Elsevier, 2021.

## 5. Mechanical Properties

To use BMGs fabricated via AM methods for different applications, the minimum requirements for mechanical properties should be met. That is why investigating the mechanical properties of the fabricated BMG parts is of high importance. For crystalline materials, e.g., Al-alloys [163], the relationship between microstructure and mechanical properties is very important. However, for BMGs, this relationship is more difficult to study, as structure and deformation mechanisms of BMGs are not fully understood. However, it has been proven that the presence of defects related to AM methods influences the mechanical properties, both in crystalline [163] and in amorphous [78] alloys.

BMGs have higher tensile and compressive yield strength in addition to a lower Young's modulus compared to their crystalline counterparts, which stems from different deformation mechanisms. However, BMGs do not show good ductility during tensile, compression, and bending tests at room temperature, which is due to the inability to undergo homogeneous plastic deformation [164]. The deformation mechanism in MGs relates to localized shear bands, which results in calamitous failure.

In this section, the mechanical properties of AM fabricated BMGs, such as hardness, compressive strength, tensile strength, bending strength, toughness, wear resistance, and fatigue, are discussed.

### 5.1. Hardness

Hardness is the easiest, fastest, and most commonly used mechanical testing performed on the BMGs fabricated via AM processes. The hardness of BMGs is much higher than that of their crystalline counterparts. The harder the material, the higher the yield strength and wear resistance [164].

### 5.1.1. Microhardness

Defects related to the manufacturing process can affect the microhardness results. For instance, Shi et al. [63] showed that a decrease of porosity content resulted in a hardness increase. The hardness of BMGs is also affected by the degree of crystallization in the amorphous structure. In most cases, unwanted crystallization causes the formation of intermetallics and/or brittle phases. Marattukalam et al. [56] confirmed this fact by producing samples with different amounts of crystallization and reported that crystallization made AMZ4 more brittle.

Microhardness results of BMGs produced via AM methods are presented in Table 3. Since different loads lead to different hardness values, it was preferred to present them in a table rather than with a graph. Although Sohrabi et al. [32] reported the presence of nanocrystals in AMZ4 LPBF parts, the hardness results were lower than those claimed in the literature (see Table 3); nevertheless, a higher density was achieved. This may come from the fact that the resolution of XRD and DSC techniques are often limited to detection of a few percent of crystallization, and advanced characterization methods, such as TEM and synchrotron XRD should then be utilized [32,73].

**Table 3.** Microhardness results of BMGs fabricated via AM methods.

| Alloy | Process Method | Hardness (HV) | Load (kgf) | Structure | Ref. |
|---|---|---|---|---|---|
| AMZ4 | LPBF | 438 | 2 | Amorphous | [34] |
| AMZ4 | LPBF | 466 | 1 | Amorphous | [53] |
| AMZ4 | LPBF | 484 469 466 | 5 0.5 0.05 | Amorphous | [63] |
| AMZ4 | LPBF | 465 455 446 | 1 2 5 | Highly amorphous | [32] |
| $Zr_{60.14}Cu_{22.31}Fe_{4.85}Al_{9.7}Ag_3$ | LPBF | 425 | 0.5 | Amorphous | [50] |
| $Zr_{50}Cu_{50}$ | LPBF | 593 462 | 0.05 | Amorphous Crystallized | [74] |
| FeSiBCrC | LPBF | 900 | 0.1 | Crystallized | [75] |
| FeCrMoBC | DED | 640 1400 | 2 | Amorphous Crystallized | [100] |
| $Pd_{43}Cu_{27}Ni_{10}P_{20}$ | LPBF | 498 | 1 | Amorphous | [73] |
| $Zr_{65}Cu_{17.5}Ni_{10}Al_{7.5}$ | LFP | 418 430 398 | 0.025 0.1 2 | Amorphous | [28] |
| $Zr_{52.5}Ti_5Al_{10}Ni_{14.6}Cu_{17.9}$-LM105 | LFP | 563 | 0.05 | Amorphous | [107] |

Some BMGs fabricated via LPBF are very brittle and cannot tolerate the deformation caused by microhardness indentation (see Figure 43a) [42]. Hofmann et al. [42] used the length of the generated cracks to calculate the fracture toughness. Figure 43b shows shear band formation as a result of the deformation caused by a microhardness indent on a Pd-based BMG fabricated via LPBF [73].

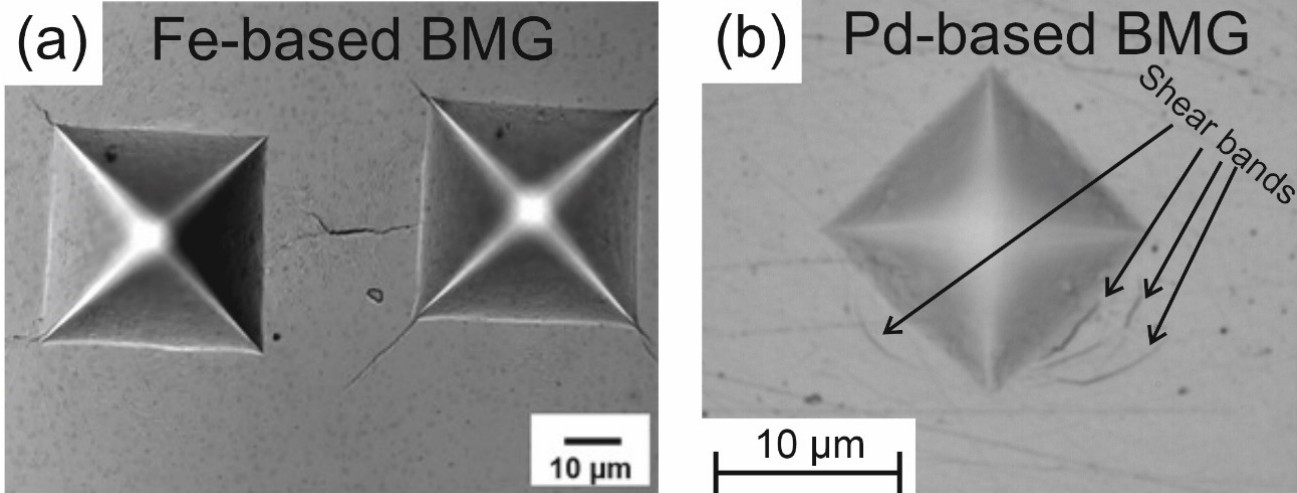

**Figure 43.** (**a**) Cracking at the corners of two hardness indents on a Fe-based BMG [42]; (**b**) shear band formation due to the deformation induced by a hardness indent on a Pd-based BMG [73].

### 5.1.2. Nanohardness

One of the applications of using nanohardness test for BMGs fabricated via AM is to use mapping and check if the microstructure is homogeneous [32,68,70,85] (see Figures 23b and 24). This was explained in details previously in Section 4.4, homogeneity of the structure.

Another application of nanoindentation is to measure the hardness very locally, where the region is smaller than the microhardness indent size. For instance, Ouyang et al. [40] measured the hardness of the melt pool and HAZ as 6.09 and 6.27 GPa, respectively, using nanoindentation. There are several other studies that used nanoindentation for the same purpose [44,49,52,56,58,102,115].

### 5.2. Compressive Behavior

The most studied mechanical property among BMGs fabricated via AM is compressive strength. A number of reported values from the literature are summarized in Table 4.

Figure 44 shows the fracture surface of AMZ4 fabricated via LPBF, after compression test. In Figure 44a, the fracture angle is very close to the maximum shear angle (45°), which is an indication of failure as a result of shear band propagation [32]. However, if the fracture happens as a result of the presence of defects in the AM parts, the fracture angle is 90° with respect to the direction of applied load [86]. Vein-like patterns were detected on the fracture surface, indicating significant softening or reduced viscosity during fracture [165]. Since BMGs, in most cases, do not undergo plastic deformation and have a high elastic limit, they store a high level of elastic energy. The stored energy is suddenly released in the final stage of fracture, which leads to very local temperature increase, and liquid-like features (Figure 44d). This is supported by the findings of Bruck et al. [166] and Liu et al. [167].

**Table 4.** Compression test results of BMGs fabricated via AM methods.

| Alloy | Ref. | Fabrication Method | $\sigma_y$ (MPa) | $\sigma_{max}$ (MPa) | $\varepsilon_p$ (%) | Size (mm³) | Comment |
|---|---|---|---|---|---|---|---|
| $Zr_{52.5}Ti_5Cu_{17.9}Ni_{14.6}Al_{10}$-LM105 | [85] | LPBF | 1500 | 1500 | 0 | Ø3 × 6 | |
| $Zr_{55}Cu_{30}Al_{10}Ni_5$ | [86] | LPBF | 1504 ± 103 | 1504 ± 103 | 0 | Ø3 × 6 | |
| $Zr_{52.5}Ti_5Cu_{17.9}Ni_{14.6}Al_{10}$-LM105 | [33] | LPBF | 1623 ± 52 | 1623 ± 52 | <0.5 | Ø3 × 6 | |
| $Zr_{50}Ti_5Cu_{27}Ni_{10}Al_8$ | [97] | DED | 2550 ± 180 | 2700–2840 | <1 | Ø0.003 × 0.006 | |
| $Zr_{57.4}Ni_{8.2}Cu_{16.4}Ta_8Al_{10}$ | [46] | LPBF | 1710 ± 39 | 1932 ± 37 | 2.15 ± 0.25 | Ø1.5 × 3 | Including ductile phase, Ta |
| $Zr_{55}Cu_{30}Al_{10}Ni_5$ | [95] | LSF | 1266–1452 | 1266–1452 | 0 | Ø3 × 6 | |
| AMZ4 | [53] | LPBF | 1820 ± 50 | 1860 ± 50 | 2.03 ± 0.04 | Ø0.002 × 0.005 | |
| AMZ4 | [32] | LPBF | 1368 ± 41 | 1368 ± 41 | 0 | Ø6 × 9 | |
| $Zr_{52.5}Cu_{17.9}Ni_{14.6}Al_{10}Ti_5$ | [57] | LPBF | 1710 ± 40 | 1710 ± 40 | 0.5 | Ø3 × 6 | |
| | | | 1420 ± 20 | 1540 ± 10 | <0.2 | 3 × 3 × 6 | |
| $Zr_{55}Cu_{30}Al_{10}Ni_5$ | [58] | LPBF | 1499 | 1499 | 0 | Ø3 × 6 | As-built |
| | | | 491 | 491 | | | Heat treated |
| $Ti_{47}Cu_{38}Zr_{7.5}Fe_{2.5}Sn_2Si_1Ag_2$ | [39] | LPBF | 1690 ± 50 | 1690 ± 50 | 0 | Ø2 × 4 | |
| $Pd_{47}Cu_{23}Ni_{10}P_{20}$ | [73] | LPBF | 1138 ± 78 | 1138 ± 78 | 0 | Ø4 × 6 | |
| $Fe_{55}Cr_{25}Mo_{16}B_2C_2$ | [49] | LPBF | 4500 | 6000 | <2 | Ø0.0015 × 0.003 | |
| $Al_{85}Nd_8Ni_5Co_2$ | [84] | LPBF | 940 | 1080 | 2.45 | N.A | |
| $Cu_{46}Zr_{47}Al_6Co_1$ | [60] | LPBF | 940 | 940 | 0 | N.A | Composite |
| $Cu_{50}Zr_{43}Al_7$ | [52] | LPBF | 1044 | 1044 | 0 | Ø4 × 8 | |
| Cu/FeCrMoCB | [64] | LPBF | 780 ± 10 | 885 ± 2 | 2.77 | Ø3 × 6 | 40% Cu |
| | | | 777 ± 4 | 857 ± 2 | 3.88 | | 50% Cu |
| $Zr_{50}Cu_{50}$ | [74] | LPBF | 957 | 1841 | 3.17 | Ø2 × 4 | Composite |
| $Zr_{60.14}Cu_{22.31}Fe_{4.85}Al_{9.7}Ag_3$ | [50] | LPBF | 1607 ± 14 | 1734 ± 30 | 1.43 ± 0.17 | Ø1 × 2 | |
| | | | 1661 ± 34 | 1758 ± 18 | 0.51 ± 0.08 | Ø2 × 4 | |
| | | | 1670 ± 36 | 1770 ± 17 | 0.46 ± 0.03 | Ø3 × 6 | |

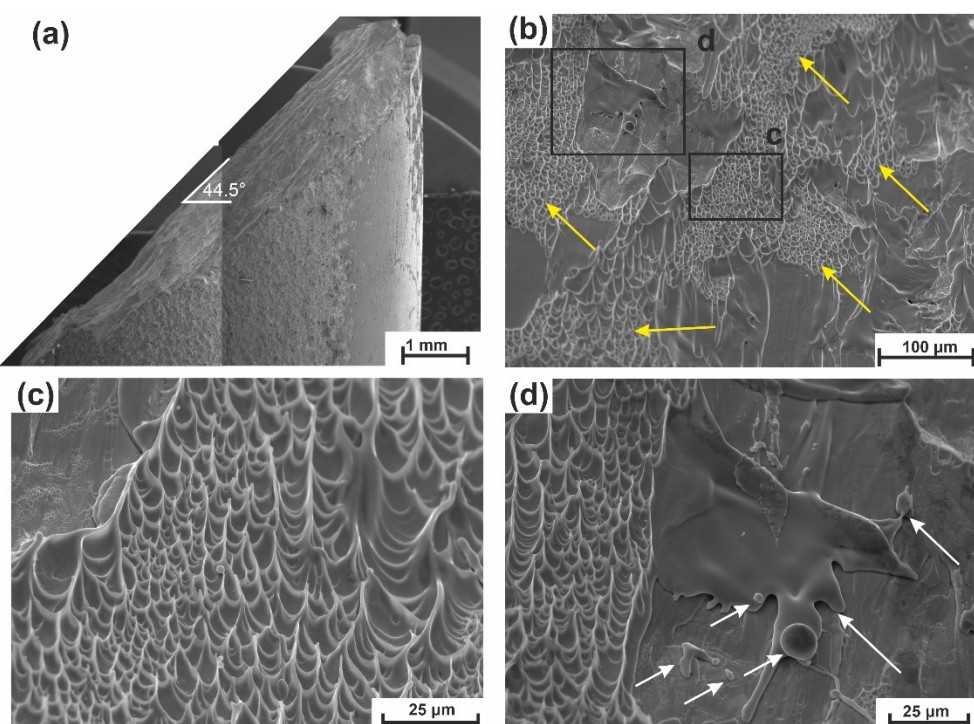

**Figure 44.** (**a**) AMZ4 compression test specimen fabricated via LPBF, after failure, (**b**) vein-like patterns on the fracture surface indicated by yellow arrows, (**c**) higher magnification image of vein-like patterns, and (**d**) presence of liquid-like features indicated by white arrows [32].

Zou et al. [64], who added Cu to a Fe-based BMG to improve ductility. On the other hand, Zhang et al. [74] reported that LPBF of $Zr_{50}Cu_{50}$ resulted in crystallization of the austenitic B2 phase, which transformed to a martensite phase (B19′) due to deformation. This mechanism led to a strong strain hardening, and 3.17% plastic deformation.

As the size of MGs reduces, they become more ductile [168]. The size effect is also observed in BMGs fabricated via AM techniques [49,50,53,97]. In compression tests on micropillars [49,53,97], the effects of defects inherent to the AM processes, and the possible presence of crystals, are not taken into account.

### 5.3. Tensile Behavior

BMGs generally show quasi-brittle behavior without macroscopic plastic deformation under uniaxial tensile loading. As mentioned before, in BMGs, plastic deformation is confined in shear bands with a nanoscale width, which is then followed by rapid propagation, and finally abrupt rupture [169]. Three studies on the tensile properties of a Zr-based BMG fabricated via LPBF are reported [67,77,78]. Best et al. [67] correlated the catastrophic failure of specimens to defects such as LoF. This finding is supported by Refs [77,78]. Shi et al. [77] fabricated three sets of samples with different amounts of porosity content and showed that the ultimate tensile strength (UTS) is more sensitive to the porosity content compared to the Young's modulus. Sohrabi et al. [78] produced three sets of samples to check the effect of defects in the near-surface region and those present in the core of the samples. Improving the parameters related to the near-surface (border) region resulted in 27% of increase in the UTS, while increasing the core power only improved the UTS by 5%. Although improvement of border parameters (reducing border distance and increasing border power) and core power led to the formation of nanocrystals, removing defects such as LoF was more effective in improving the UTS.

The macroscopic image of the fracture surface (X-Z plane) of a sample fabricated with a Set A of processing parameters in Ref [78] (before improving the border and core parameters) is presented in Figure 45a. It is evident that failure starts from the upper-left corner and propagates diagonally. Five LoFs defects are detectable on the fracture surface (shown by yellow-dashed arrows). Figure 45b shows the magnified region b of Figure 45a, where failure started from a LoF close to the surface. Another LoF is shown in Figure 45c. It looks like a spatter attached to the surface, which prevented complete melting. Figure 45d (higher magnification image of region d in Figure 45c) shows dimple-like features. According to Lin et al. [95], dimples are created when nanoparticles are peeled off due to the tensile loading. A vein-like pattern is presented in Figure 45e (higher magnification image of region e in Figure 45b). Figure 45f shows that a crack is arrested when it reaches a porosity, with shear bands multiplication inside the porosity. Although porosity is a detrimental defect, it here locally improves plasticity and delays crack propagation.

There are several studies on the tensile properties of BMGs, and BMGCs fabricated with other laser-based AM process, i.e., DED [95,102–104] and FFF [117]. Although Su et al. [103] worked on a Zr-based BMGC, which contained a ductile crystalline phase, less than 0.5% ductility in tension was measured. This low ductility was attributed to the presence of defects in the fabricated part. Lin et al. [95] hold the presence of defects and crystalline phases responsible for the low tensile strength of the Zr-based BMG. Lu et al. [102] could improve the plasticity of a Zr-based BMG fabricated via DED by introducing a ductile crystalline phase, and they could achieve a tensile yield strength higher than 1.3 GPa, with 13% plasticity. Gibson et al. [117] tested the tensile strength of the fabricated BMG in two different directions. When the loading direction was perpendicular and parallel to the build direction, tensile strengths of 1220 and 790 MPa, respectively, were achieved. Neither of the samples showed plastic deformation.

A comparison of tensile yield strengths of BMGs fabricated via AM is presented in Figure 46.

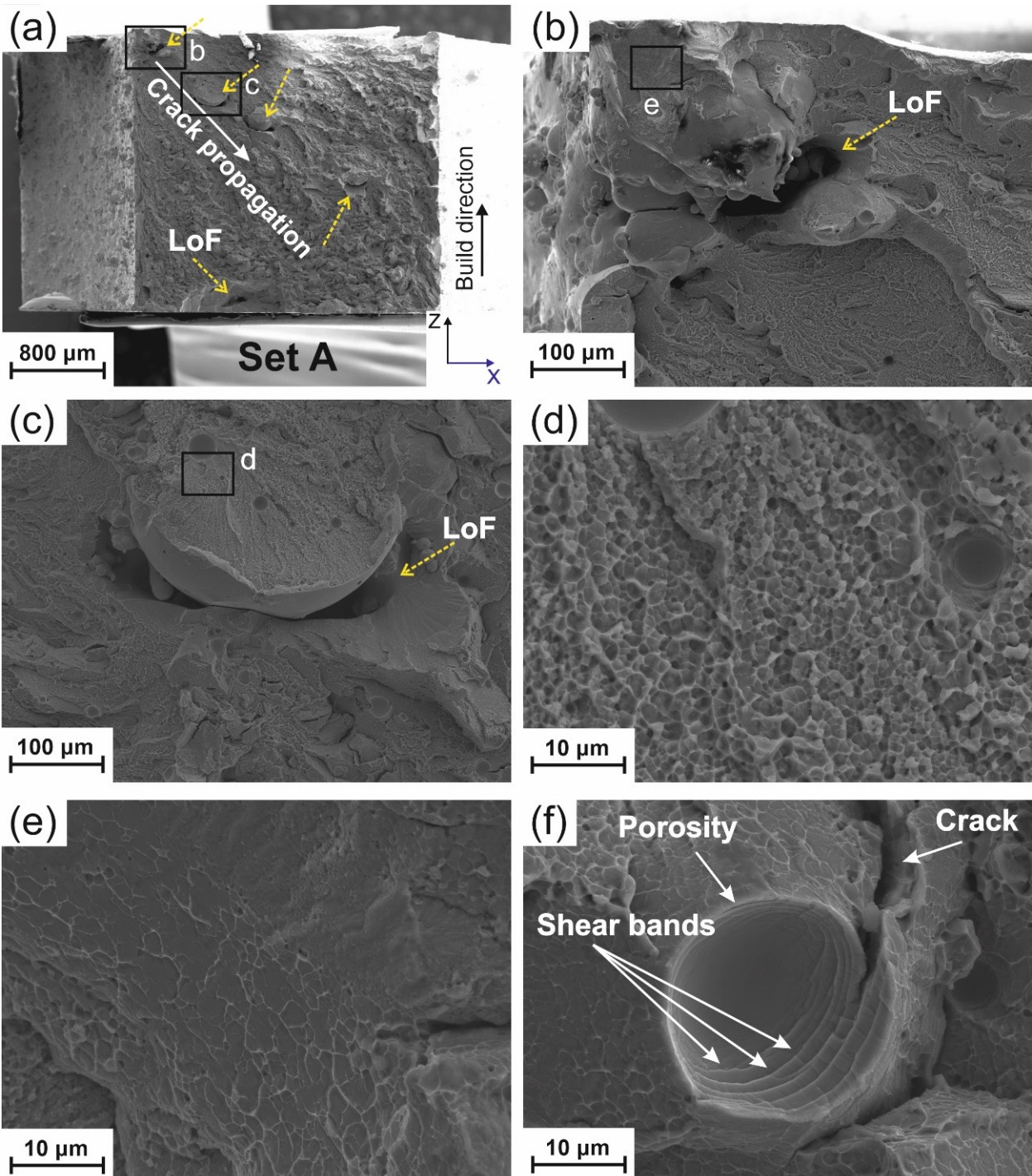

**Figure 45.** (**a**) Macroscopic fracture surface of a sample fabricated by a chosen Set A of processing parameters in Ref. [78], (**b**) magnified region b in (**a**) showing the location where failure started from a LoF close to the surface, (**c**) magnified region c in (**a**) showing another LoF in the core of the sample, (**d**) magnified region d in (**d**) showing a dimple-like structure, (**e**) magnified region e in (**b**) showing vein-like pattern, which is typical of BMGs, and (**f**) porosity arrests a crack and multiple shear bands are formed.

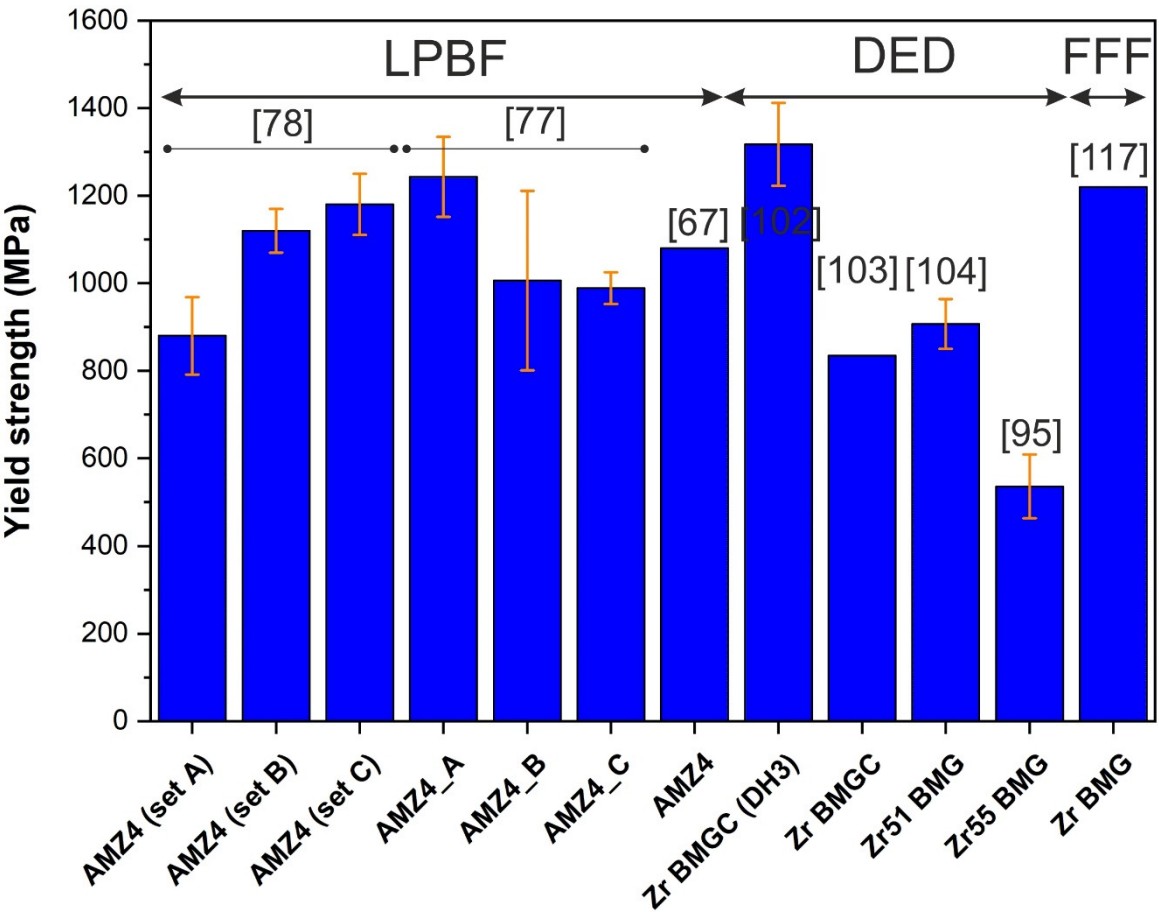

**Figure 46.** Comparison of reported tensile yield strengths of BMGs fabricated via AM. [67,77,78,95,102–104,117].

### 5.4. Bending Behavior

Bending test results of BMGs fabricated via AM methods are presented in Table 5. Bordeenithikasem et al. [28] performed four-point bending (4PB) tests on an industrial-grade Zr-based BMG ($Zr_{65}Cu_{17.5}Ni_{10}Al_{7.5}$) fabricated via LFP. They achieved a high flexural yield strength of 1880 MPa, with a subsequent strain hardening up to 2250 MPa. The reason for such a high flexural yield strength is that, unlike LPBF processed samples, LFP samples do not contain porosity and LoF. The yield and maximum flexural strength of the LFP samples were lower than lab-grade as-cast samples due to a higher impurity content in the industrial-grade feedstock. In another study, Bordeenithikasem et al. [34] fabricated samples from AMZ4 via LPBF and evidenced a low flexural strength compared to lab-grade as-cast samples, which was related to LPBF defects.

**Table 5.** Bending test results of BMGs fabricated via AM methods tested with four-point bending (4PB) and three-point bending (3PB).

| Alloy | Fabrication Method | Ref. | Flexural Yield Strength (MPa) | Maximum Flexural Strength (MPa) | Testing Method | Plasticity | Protective Gas |
|---|---|---|---|---|---|---|---|
| $Zr_{65}Cu_{17.5}Ni_{10}Al_{7.5}$ | LFP | [28] | 1880 | 2250 | 4PB | Yes | Ar |
| AMZ4 | LPBF | [34] | 1300 | 1300 | 3PB | No | N.A |
| AMZ4 | LPBF | [69] | 1167±108 | 1167 ± 108 | 3PB | No | N$_2$ |
| | | | 1684 ± 116 | 1684 ± 116 | | | Ar |
| | | | 1693 ± 50 | 1693 ± 50 | | | Ar$_{98}$H$_2$ |
| AMZ4 | LPBF | [71] | 2100 | 2100 | 3PB | No | Ar |
| AMZ4 | LPBF | [32] | 1666 ± 33 | 1666 ± 33 | 3PB | No | N$_2$ |

Wegner et al. [69] investigated the effect of the protective gas ($N_2$, Ar, and $Ar_{98}H_2$) during LPBF process of AMZ4. The maximum average value of flexural strength was for the samples fabricated in the $Ar_{98}H_2$ atmosphere, and the lowest corresponded to the use of $N_2$. The porosity content and the size of microcracks in the sample fabricated in the $N_2$ environment was higher than the other two. However, no plastic deformation was detected in neither of the three conditions. Sohrabi et al. [32] could fabricate a crack-free, dense (>99.8%) AMZ4 via LPBF in a $N_2$ environment, that resulted in a flexural strength of 1666 MPa. Frey et al. [71] could fabricate AMZ4 using LPBF in an Ar environment with a flexural strength of 2100 MPa, which is the highest result reported so far. The as-cast sample yielded at the same stress level, with more than 2% plasticity, and the maximum flexural strength was between 2500–2750 MPa.

### 5.5. Low Ductility

BMGs fabricated via AM techniques suffer from low ductility because of the presence of defects, annealing cycles during the process, and uncontrolled crystallization. Previously noted that, there are some studies that could detect plastic deformation during compression [46,49,50,57,64,74,97] and tension [102,103], mostly by introducing a ductile crystalline phase or performing the test on miniaturized samples, e.g., compression tests on micropillars.

Defects, such as LoFs and porosities, are considered as stress risers that can impair mechanical properties [78]. Best et al. [67] attributed the premature uniaxial tensile failure of LPBF manufactured AMZ4, before yielding, to the presence of LoFs, which were detected on the fracture surface, in agreement with Refs [34,78]. Sohrabi et al. [73] observed crack propagation through porosities inside a Pd-based BMG cylinder tested under uniaxial compression. These defects facilitated crack propagation and led to fast failure without undergoing plastic deformation.

Lin et al. [95] detected no plasticity in tensile and compression tests of a Zr-based BMG fabricated via LSF, and attributed this behavior to the annealing that occurs as a result of the heating cycles of the process. The annealing causes structural relaxation, rearrangement of atoms [34] and annihilation of free volume [170]. Consequently, there is less chance for the multiplication of shear bands, and failure happens as a result of the propagation of a single shear band [168], which is supported by Refs [40,50,52,58].

The reasons for the crystallization of BMGs fabricated via AM processes were discussed in Section 4.6. Uncontrolled crystallization in most of the cases form brittle phases, such as intermetallics, which act as stress risers, increase brittleness, and do not tolerate plastic deformation [95,107,108,171].

### 5.6. Fracture and Impact Toughness

Fracture toughness indicates the resistance to crack propagation. As already mentioned before, Zhang et al. [46] introduced a ductile crystalline phase (Ta) in a Zr-based BMG fabricated via LPBF to increase both the ductility and the fracture toughness. Figure 47a shows that the notch toughness, $K_q$, of the LPBF and an as-cast samples are comparable. Three distinct regions were detected on the fracture surface (see Figure 47b). The smooth region corresponded to the formation of shear bands. The vein-like region was attributed to the softening phenomenon resulting from the propagation of the shear bands, and the dimple region related to the fast crack propagation. Figure 47c clearly shows how the ductile crystalline phase (Ta) helps arresting the crack and prevents its propagation, thereby improving the toughness. Along the same line, Li et al. [38] added ductile phases, Cu and CuNi, to a Fe-based BMG fabricated via LPBF. The sample without any ductile phase was very brittle and contained microcracks, with a fracture toughness ($K_J$) of 2.2 MPa$\sqrt{m}$. The fracture toughness of the BMGC containing 50% Cu increased 21.4 times compared to the monolithic Fe-based BMG because the ductile phase impeded the crack propagation. Zhang et al. [74] fabricated a binary $Zr_{50}Cu_{50}$ BMGC using LPBF. $Zr_{50}Cu_{50}$ formed a B2 austenite phase in the HAZ, which can transform to B19′ martensite as a result of

plastic deformation. This transformation increased the ductility (3.17% plastic strain in compression) and fracture toughness ($33.4 \pm 1.6$ MPa$\sqrt{m}$).

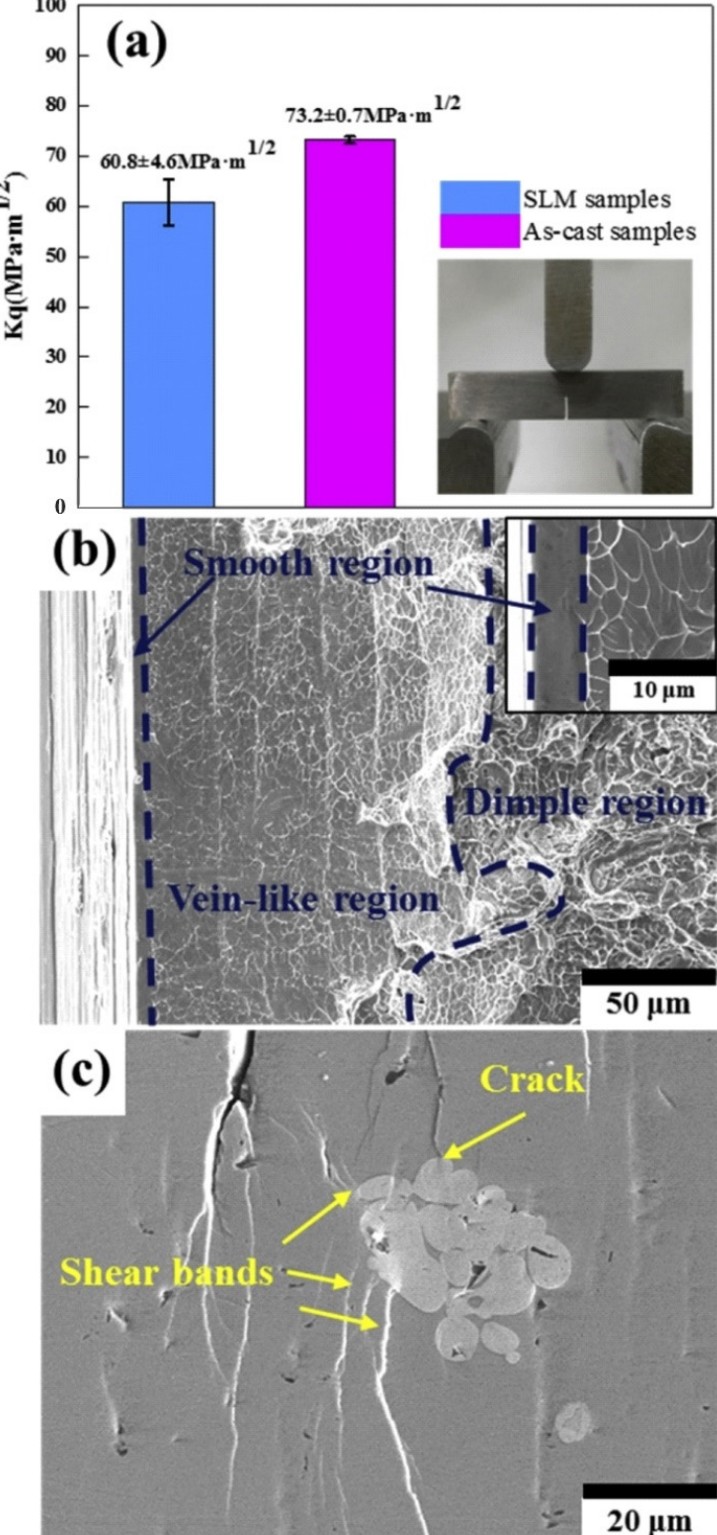

**Figure 47.** (**a**) Comparison of the notch toughness, $K_q$, of LPBF and an as-cast samples, (**b**) fracture surface of the LPBF sample after failure, and (**c**) crack-arrest by the ductile Ta phase. Reproduced from [46], with permission from Elsevier, 2021.

Best et al. [53,67] compared the $K_q$ of AMZ4 with a high amount of oxygen content (1271 ppm) fabricated via LPBF, and of a lab-grade as-cast AMZ4, which had 168 ppm oxygen content. The value for the lab-grade as-cast material ($138 \pm 13.1$ MPa$\sqrt{}$m) was almost five times that of the industrial-grade LPBF sample ($28.7 \pm 3.7$ MPa$\sqrt{}$m). This difference was attributed to the amount of oxygen content, because dissolved oxygen impedes the movement of atoms.

Although BMGs have a fracture toughness comparable to that of crystalline alloys [172], they have a very low (<2 J) impact toughness [156,157,173–177]. Nagendra et al. [173] showed that partial crystallization of an as-cast La-based BMG could drastically decrease the impact toughness by 90%.

There is only one study in the literature, Ref [78], related to the impact toughness of a BMG fabricated via AM. Charpy impact tests (CIT) indicated a very low value, 0.12 J, for AMZ4 fabricated via LPBF. The fracture surface showed the presence of LoFs in the near-surface and core regions of the samples. Two strategies were implemented to improve the CIT results and investigate the importance of the location of defects. Strategy 1 was only related to the border (contour) parameters, such as reducing the border distance and increasing locally the laser power. The strategy reduced the number of LoFs in the near-surface region and resulted in a 28% improvement in CIT. Strategy 2 was, apart from the border parameters, increasing the laser power in the core of the sample. A 3% improvement was measured compared to strategy 1. It was therefore confirmed that the mitigation of defects close to the surface is more important than those in the core of the sample. Besides, as a result of the increase of power in the border and/or core, the amount of crystallization increased. Although crystallization reduces the impact toughness, the effect of defect mitigation prevailed.

Figure 48a shows the macroscopic fracture surface of an AMZ4 sample in LPBF as-built condition, after the impact test. The fracture surface is relatively smooth, indicating low ductility. LoF defects are shown by yellow-dashed arrows. The majority of the LoFs are close to the surface. Figure 48b shows the magnified region of b in Figure 48a. Unmelted powder particles are still detectable. Such defects, especially close to the surface, with an irregular shape (red-dashed lines in Figure 48b), could cause stress concentration and reduce the strength and toughness of the specimen. The sub-surface LoFs are aligned in a region between the border and the core. It seems that the border laser power and/or the border distance were not optimized to fully melt the powder in that region, and led to the formation of LoFs. In addition, a few LoFs inside the core (bulk) of the sample are also detected. The fracture surface of a sample produced with Strategy 1 is shown in Figure 48c, where several LoFs were detected. Since the core power is the same as for the as-built condition, the presence of LoFs in the core of the sample was expected. Using Strategy 1 was effective in reducing the sub-surface LoFs. Figure 48d shows a higher magnification image of region d illustrated in Figure 48c. It is evident that the size of the LoFs is reduced compared to those present in Figure 48b, for the as-built condition. Figure 48e shows the fracture surface of a sample produced using Strategy 2. No LoF was detected, neither in the sub-surface region nor in the core of the specimen. This means that the increase of core laser power effectively prevented the formation of LoFs in the core region. Figure 48f (a higher magnification image of region f in Figure 48e) shows the initiation of a crack close to a porosity open to the external surface. One should not neglect the effect of size, shape, location, and distribution of porosities in determining if they are beneficial or detrimental to the mechanical properties. The fracture surfaces of the samples of Strategy 1 and 2, Figure 48c,e, respectively, show jagged morphology (higher roughness) close to the notch root and then shift to a smoother morphology. The area fraction of the jagged region is higher for samples of Strategy 1 and 2, compared to the as-built condition (Figure 48a), which can be correlated with the higher toughness for Strategies 1 and 2.

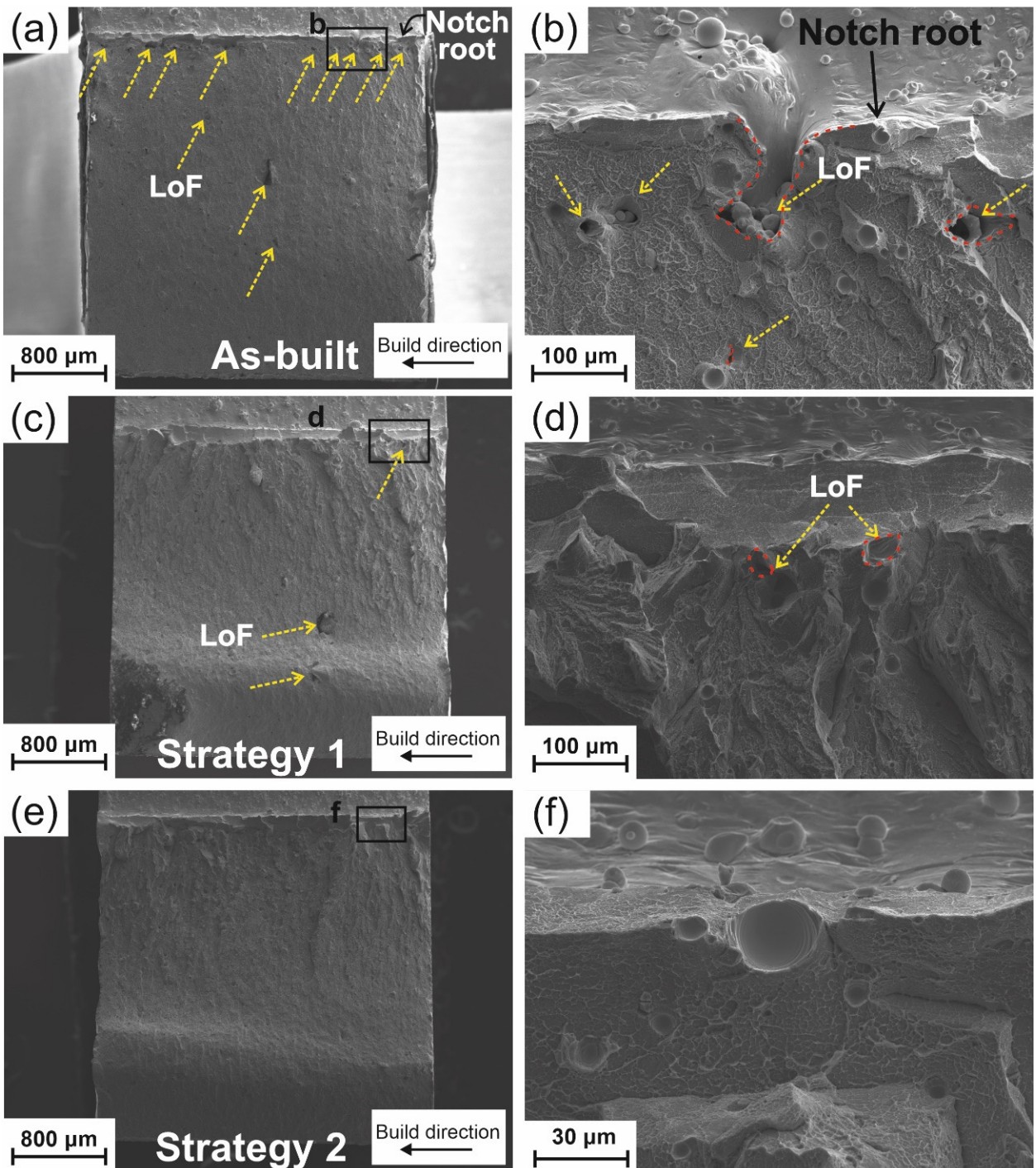

**Figure 48.** Fracture surface of the CIT samples in (**a**,**b**) as-built condition, (**c**,**d**) Strategy 1, by only decreasing the border distance and increasing border laser power, and (**e**,**f**) Strategy 2, increasing the core power in addition to changing the border parameters [78].

### 5.7. Wear Resistance

The high hardness of BMGs makes them good candidates for parts such as gears, bearings, and coating, experiencing friction and requiring a high wear resistance. Therefore, investigating the tribological behavior of BMGs fabricated via AM methods is of high importance.

Sohrabi et al. [5] used a laser-based AM method for cladding AMZ4 on an Al-alloy substrate. They used two different sets of parameters to have a good bonding with the substrate as well as a high amorphous content. They showed that a higher crystallization fraction

led to a harder clad, more resistance to wear, but more brittle. Bordeenithikasem [34] compared the weight loss of as-cast and LPBF fabricated AMZ4 during a 'pin on desk' test, and concluded that the results were similar. A Zr-based BMG ($Zr_{60.14}Cu_{22.31}Fe_{4.85}Al_{9.7}Ag_3$) fabricated via LPBF for potential biomedical applications had lower wear resistance and a higher friction coefficient compared to Ti-6Al-4V [50]. Wear-rate [32] of different BMGs, mostly fabricate via AM, compared to three crystalline alloys, are presented in Figure 49. The wear-rate ($\frac{volume\ loss}{(distance \times applied\ load)}$) of BMGs is generally lower than those of crystalline alloys. The highest wear resistance among BMGs fabricated via AM is the one of Vit105 [57]. SEM images of the wear groove in addition to an EDS map of oxygen are presented in Figure 50 [32]. Wear products are shown by black arrows, and are rich in oxygen, which indicates that the alloy was oxidized during the wear test. Since oxide films are brittle, they are prone to fragmentation. The wear mechanisms were identified as abrasive and oxidation wear, which is supported by Ref [57].

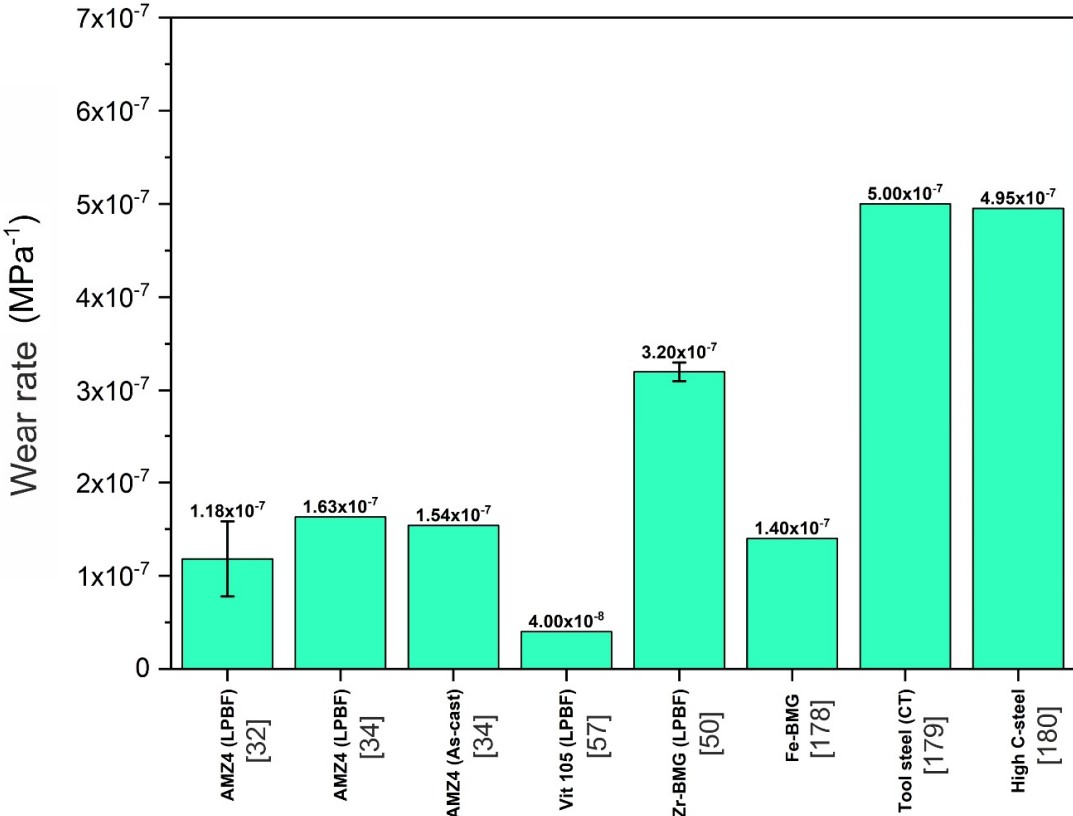

**Figure 49.** Comparison of wear rate for AM fabricated and as-cast BMGs and two crystalline alloys, [32,34,50,57,178–180].

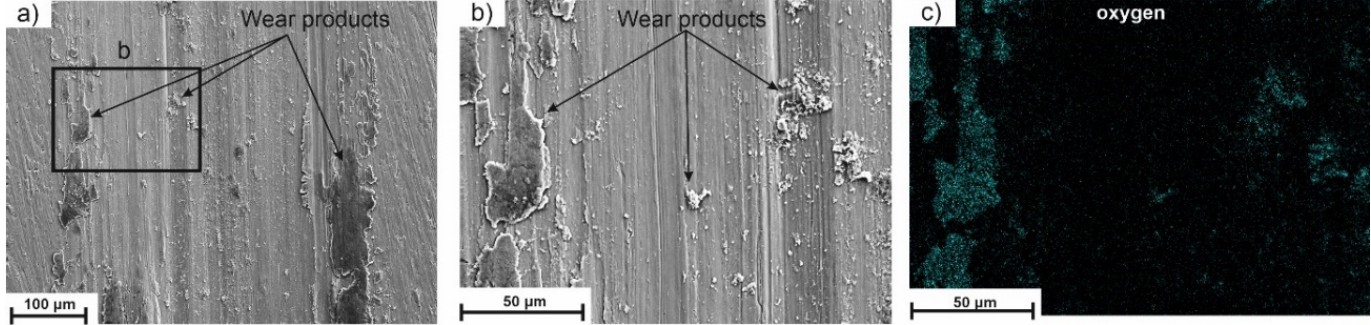

**Figure 50.** (**a**) SEM image of a wear track on AMZ4 fabricated via LPBF, (**b**) higher magnification of region b, and (**c**) EDS map of oxygen content in (**b**) [32].

### 5.8. Fatigue Performance

Fatigue is the most common cause of failure of mechanical parts in engineering applications. The typical fatigue ratio (fatigue limit/ultimate tensile strength, UTS) of BMGs is often below 20%, which is lower than values encountered with crystalline materials such as low carbon steel or aluminum alloys (30 to 40%) [181]. Under very similar chemical compositions, an extreme variability is reported in the achieved fatigue limits of BMGs, ranging from 150 MPa to 1050MPa, which correspond to fatigue ratios between 8% and 50% [168,182,183]. These properties have been shown to depend on multiple factors such as the loading mode [183–185], the sample geometry [168,186], the testing environment [187,188], and the thermal history [156,170,189].

Best et al. [67] measured fatigue crack-growth diagrams of AMZ4 fabricated via LPBF on a pre-notched sample (see Figure 51). The test was performed in tensile loading with a stress ratio, $R$, of 0.1. They reported a threshold stress intensity range, $\Delta K_{th}$, and a maximum stress intensity range, $\Delta K_{max}$, of 1.33 MPa$\sqrt{m}$ and 14 MPa$\sqrt{m}$, respectively. They correlated the low performance of AMZ4 to the high oxygen content, which prevents the mobility of atoms.

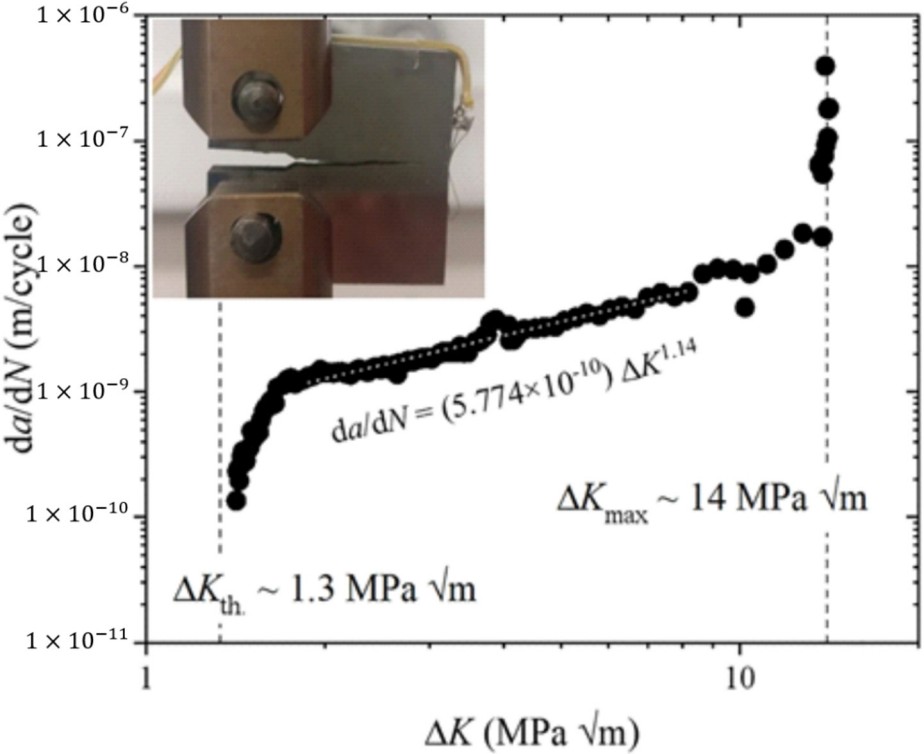

**Figure 51.** Fatigue crack-growth diagram of AMZ4 fabricated via LPBF. Reproduced from [67], with permission from Elsevier, 2021.

The fatigue performance (fatigue life and limit) of AMZ4 fabricated via LPBF was investigated by Sohrabi et al. [79]. The absolute value of the fatigue limit (175 MPa) of AMZ4 was ranked among the lowest reported ones for Zr-based BMGs, and was similar to the partially crystallized BMGs [181]. This low value was attributed to the combination of several reasons: (1) the presence of porosity defects, such as LoFs, in the fabricated part, (2) the presence of a high level of oxygen content, (3) partial crystallization, and (4) the presence of TRS. The fatigue ratio was however, comparable to that of stress-relieved $Zr_{44}Ti_{11}Ni_{10}Cu_{10}Be_{25}$ [170] and $Zr_{56.2}Cu_{6.9}Ni_{5.6}Ti_{13.8}Nb_{5.0}Be_{12.5}$ [190], which contained a ductile crystalline phase, and was higher than that of Vitreloy 1 [185], for which the size of the specimens was closer to the one used in [79].

Three strategies for improving the fatigue behavior were considered: (1) relaxation heat treatment (RHT), giving a slight fatigue life improvement at high loading conditions

($\geq$250 MPa); (2) laser shock peening (LSP); and (3) changing the build orientation. Except for the third one, these strategies are shown to provide little improvement, from which it was concluded that fatigue life in BMGs fabricated by LPBF is primarily influenced by powder quality (oxygen content in particular) and process-induced defects, which cannot be changed with post-treatments. In particular, LoF porosities should be further reduced by an appropriate increase in laser power during LPBF, either in the near-surface regions or in the core of the specimens, while paying attention to the associated (detrimental) increase of crystallized fraction [79].

## 6. Outlook

BMGs attracted much attention after the emergence of AM techniques, as these opened new avenues for producing BMGs larger than their critical casting diameter, and with complex geometries. Based on the reviewed papers, AM showed great potential for accelerating the development of BMGs with high mechanical properties. BMGs fabricated via AM have a potential for use in sporting, bio-medical, aerospace, and electronic industries. However, there are still some challenges that require in-depth investigations.

The first challenge is the presence of defects inherent to AM processes, which impairs the part integrity. Defects can be detected during processing with monitoring techniques, including optical and acoustic ones [191]. It has been shown that different types of defects, such as porosities, LoFs, and cracks, have specific signatures, which can be recognized and analyzed. Acoustic signals can in particular be acquired with simple microphones, and further processed with machine learning (ML) algorithms. If the defect location can be identified, corrective actions can be implemented, for instance remelting operations to remove defects.

Another challenge in AM of BMGs is crystallization. The detection of such defects is not as straightforward, since crystallization results from a diffusion-based phase transformation with a specific kinetics. As noted earlier, it was shown that AM of BMGs with complex shapes lead to variable fractions of crystallization at different locations of the part, due to variable thermal histories. Artificial intelligence (AI) based algorithms currently develop such as to change the processing parameters during the fabrication, using information gathered from monitoring techniques. Another approach uses beam shaping for laser-based methods, such as to dynamically change the spatial distribution of laser energy and engineer the local microstructures. Another use of AI and ML methods consists of designing new BMGs, with properties tuned for AM processes.

Similar to the complex shapes issue discussed above, scalability is another concern, i.e., optimized parameters for small samples do not necessarily lead to the same quality in larger samples. Scalability issues and their consequence on microstructure and mechanical properties of AM fabricated BMGs have not been investigated to this day.

Little attention has been paid to the fatigue performance of BMGs fabricated via AM, which is the most frequent reason for failure in engineering applications of metals. Since fatigue performance of metals is highly affected by defects, the challenge consists of fabricating BMGs with the lowest possible defects content, without inducing excessive crystallization. This would require BMGs with high GFA and low oxygen content. Optimization methods might here again benefit from the ongoing developments in AI and ML.

Full scale thermal simulations of AM processes are a well-known challenging multiscale problem, which is especially relevant when fabricating BMGs. Considering the fluid flow inside the melt pool, the temperature field, vaporization and plasma effects, and identifying the proper thermo-physical (temperature depend) parameters, are all necessary to be able to predict crystallization of BMGs. A number of numerical strategies have been devised to limit computational cost, e.g., layer-by-layer heat input, followed by line-by-line heat input, followed by normal scanning, should allow accounting for both local and global thermal effects.

AM helped the development of multi-materials and functionally graded material (FGM). So far, there are few (less than five) studies on AM of BMGs focused on FGM, based on progressive changing of the processing parameters. In contrast, there is no study on multi-BMGs, or on the combinations BMG-crystalline alloys, or BMG-ceramic parts. This can be of high interest for new engineering applications with demanding material properties.

The low ductility issue in BMGs is being addressed by the introduction of ductile crystalline phases. Post-processing methods such as shot peening (SP), LSP, and deep cryogenic treatment, could help in this regard. New BMGs should be designed with the ability to form soft crystals, as exemplified by the case of $Zr_{50}Cu_{50}$.

## 7. Summary

In the last two decades, additive manufacturing (AM) of metallic alloys has been widely used in different industries, and gained much attention in the academic world as well. The reasons for such a high interest in AM are the lower waste (higher buy-to-fly ratio), and the fabrication of value-added parts, which cannot be obtained by conventional processing methods. Thanks to the high cooling and heating rates ($10^2$–$10^6$ K/s), especially in laser-based AM, and considering the layer-wise fabrication in AM processes, it is possible to produce alloys such as bulk metallic glasses (BMGs).

This review focused on AM of BMGs, and reported AM techniques used for fabrication of BMGs, which are mostly laser-based, but also include ultrasonic AM (UAM) and fused filament fabrication (FFF). The advantages and shortcomings of each technique were discussed. The majority (more than 70%) of the research in this field, so far, is dedicated to laser powder-bed fusion (LPBF), because this technique provides the highest heating and cooling rates. BMGs fabricated via AM techniques were then categorized based on their principle element. Almost 70% of the published studies relate to Zr-based BMGs, among which an industrial-grade BMG called AMZ4 ($Zr_{59.3}Cu_{28.8}Al_{10.4}Nb_{1.5}$), that received the highest attention because it is cheaper and more easily available compared to lab-grade metallic glass powders.

The main challenges in AM of BMGs were identified as related to parameter optimization, presence of defects, crystallization, and low ductility, all of which being interconnected. The processing window for producing BMGs is limited by the need to avoid both defects and crystallization. A critical compromise must be found between these two effects, as it is usually impossible to avoid both of them completely.

Several crystallization mechanisms were reported for BMGs fabricated via AM, involving heterogeneity of chemical composition, structural relaxation in the HAZ, time spent at high temperatures in the HAZ, cooling rates lower than the critical cooling rate, and heat accumulation at the part scale (*global* effect). It is now established that crystallization in the HAZ occurs due to multiple reheating cycles, inherent to AM techniques. Some BMGs form crystals softer than the amorphous matrix, which is an interesting avenue to improve ductility, by forming a BMG composite (BMGC).

Mechanical properties such as hardness, fatigue performance, wear resistance, fracture and impact toughness, compressive, and tensile and bending strength were investigated. Hardness and compressive strengths were the most investigated ones because tests are easier and faster to perform, and can be done with small samples. Hardness and wear resistance of AM parts were comparable with the as-cast material. However, other mechanical properties were directly affected by AM defects and crystallization, which resulted in reduced properties with respect to the as-cast reference, while remaining higher than their crystalline counterparts. Low ductility in BMGs fabricated via AM techniques not only is affected by crystallization and the presence of defects, but also by the oxygen content, and by the annealing effect induced by the repeated heating cycles during processing.

**Author Contributions:** Conceptualization, N.S.; Validation, J.J. and R.E.L.; Writing—original draft preparation, N.S.; writing—review and editing, J.J. and R.E.L.; Visualization, N.S.; Supervision, J.J. and R.E.L.; Project administration, R.E.L.; Funding acquisition, R.E.L. All authors have read and agreed to the published version of the manuscript.

**Funding:** This work was supported by the "PREcision Additive Manufacturing of Precious metals Alloys (PREAMPA)" project. The PREAMPA project is funded by the Swiss ETH domain, within the Strategic Focus Area on Advanced Manufacturing.

**Institutional Review Board Statement:** Not applicable.

**Informed Consent Statement:** Not applicable.

**Data Availability Statement:** The data are available and can be shared upon a reasonable request.

**Acknowledgments:** The generous support of PX Group to the LMTM laboratory is also highly acknowledged. Finally, we thank the authors of many of the original articles cited in this review for providing high-resolution images of their artwork.

**Conflicts of Interest:** The authors declare no conflict of interest.

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
