# Peer review of "Additive Manufacturing of Bulk Metallic Glasses—Process, Challenges and Properties: A Review"

_metals, doi:10.3390/met11081279_

Round 1
Reviewer 1 Report
The work attempts to provide a comprehensive review of recent developments on additive manufacturing (AM) of bulk metallic glasses (BMGs). It considers different AM techniques that can be employed for fabricating BMGs, describes the crystallization behavior of BMGs, presents results for mechanical properties and deformation mechanisms. The manuscript also considers some challenges to be solved for further developing in this direction. Overall, this is a nice study on an interesting topic. The manuscript is clearly outlined, well arranged in a logical sequence and well presented. Only minor issues should be considered as follows:
-In addition to the review articles mentioned in the Introduction part by the authors, there are several important review articles published recently on additively manufacturing of BMGs (i.e., Mater Sci Eng R 145, 2021, 100625), which should be also mentioned. Considering the readily published review on this topic, the difference of the present review paper from the abovementioned review should be highlighted in the Introduction part.
-There are also several novel additive manufacturing techniques for fabricating BMGs, e.g., thermal spray and cold spray. See J Mater Chem A 6, 2018, 6800. Scr Mater 177, 2020, 112. Adv. Eng. Mater. 20, 2018, 1800433.
-Fig. 5 and Fig. 7 can be omitted.
-In Fig. 49 and Page 53, the authors use the wear-rate constant, which is defined as the ratio of wear rate to applied load. Normally, the wear rate is calculated using the equation of V/NS, wherein V is the wear volume (mm3), N and S means the applied load (N) and the total sliding distance (m). Accordingly, there is no need to normalize the wear-rate by dividing applied load.
Reviewer 2 Report
The paper "Additive manufacturing of bulk metallic glasses -Process, challenges and properties: A review" by Navid Sohrabi and co authors is a comprehensive report of recent works regarding additive manufacturing of metallic glasses.
The paper is a very complete and a well written work and the authors collected the main works in the field and resumed the main findings in an understandable manner.
The paper is publishable but I would invite the authors to consider the following point for the final revision
a) Please give a reference for the source of the data in figure 5 and 7.
b) please clarify what the authors mean by TTT upon cooling and heating - TTT diagrams are always measured in isothermal mode. Please also clarify the meaning and relevance in the filed of TTT (time temperature transition diagrams - measured in isothermal mode) , CHT ( continous heating transformation diagram - measured from the heating curve) and the CCT (continous cooling transformation diagram - measured from the cooling curve). CHT and CCT are the processes occuring in most industrial production processes. Please include the related definitions and consider them for the related chapters - especially in chapter 4 (i.e. section 4.6.8).
Round 2
Reviewer 1 Report
After revision the paper can be accepted now.